# Progressive Acyclicity Induction for Effective DAG Learning under Heteroscedastic Noise Models

## Abstract

This study focuses on learning causal directed acyclic graphs (DAGs) under a *heteroscedastic noise model* (HNM), wherein an effect is modeled as a function of its cause and a Gaussian noise term whose variance depends on the cause. Integrating HNMs into a continuous optimization framework allows us to learn a causal *directed acyclic graph* (DAG) under an acyclicity constraint by maximizing a likelihood objective parameterized by both mean and variance. However, DAG learning under HNM inherits the challenges of gradient-based likelihood optimization: the gradient is scaled by the predictive variance, which introduces a new optimization issue in DAG learning under an acyclicity constraint. In early training, because the gradient of reconstruction loss is scaled by the predicted variance, it becomes heavily attenuated; as a result, the DAG parameters are updated primarily by the acyclicity constraint, hindering effective structure learning. To address this, we propose a graduated optimization strategy with weighted loss scheduling. We introduce a scheduling coefficient into the loss, starting with a high weight for stable mean and variance learning, then *gradually* lowering the coefficient to transition to the standard likelihood objective and enforce acyclicity. This approach ensures that the learned DAG more faithfully reflects the data. Experimental results on both synthetic and real-world data verify the effectiveness of our approach.

## 1 Introduction

Learning a *Directed Acyclic Graph* (DAG) from observational data can yield important insights in fields such as biology (Opgen-Rhein & Strimmer, 2007), healthcare (Zhang et al., 2013), and economics (Neuberg, 2003). Structural Equation Models (SEMs) (Peters et al., 2011) provide a framework for causal discovery under certain conditions, generally modeling each variable as a function of its direct parents and an exogenous noise term. Recently, Zheng et al. (2018) have proposed to learn DAGs by parameterizing SEMs and enforcing acyclicity through a smooth differentiable constraint. Subsequently, several methods (Zheng et al., 2020; Yu et al., 2019; He et al., 2021) have adopted and extended this continuous optimization framework. These approaches are typically instantiated as SEMs with additive noise. For each variable, the associated noise is modeled as independent and identically distributed across observations with constant variance, which corresponds to homoscedastic noise.

Meanwhile, Mooij et al. (2016) and Immer et al. (2023) report that, under this (homoscedastic noise) assumption, models exhibit limitations in terms of direction identifiability and model fit when the noise variance can vary with the input (direct parents), and they address these limitations by explicitly modeling the noise variance as a function of the parents using *Heteroscedastic Noise Models* (HNMs).

Several prior studies have explored causal discovery under heteroscedasticity (Xu et al., 2022; Immer et al., 2023; Khemakhem et al., 2021; Strobl & Lasko, 2023), but mostly in simplified scenarios such as bivariate causal direction. To generalize to multivariate data, prior works had to first learn an undirected graph structure (e.g., using PC (Spirtes et al., 2000)) and then apply those heteroscedastic techniques to orient the edges. The authors of HOST (Duong & Nguyen, 2023) and ICDH (Yin et al., 2024) both establish identifiability of HNMs in a general multivariate setting. HOST achieves this by first estimating a causal ordering of the variables, and then reconstructing the DAG via

conditional independence tests. In contrast, ICDH performs gradient-based continuous optimization with a differentiable acyclicity constraint (Zheng et al., 2018) to directly recover the DAG, while modeling the noise distribution with neural networks. Appendix B summarizes additional research on noise variance assumptions and related work on continuous optimization-based methods.

Despite this progress, learning DAGs with HNMs presents an optimization challenge because both the mean and variance functions must be learned jointly. When training with a negative log-likelihood (NLL) loss, the gradient is often dominated by the variance term, to the detriment of accurate mean estimation. This issue is well documented in heteroscedastic regression (Stirn et al., 2023; Seitzer et al., 2022; Skafte et al., 2019), and various methods aim to mitigate it by decoupling the variance term's influence on the mean. Consequently, HNM-based DAG learning inherits these same pitfalls of NLL optimization. For example, Yin et al. (2024) improved training stability by updating the mean and variance networks separately in an alternating fashion, similar to the work of Skafte et al. (2019).

**Our contributions.** We identify a new challenge specific to structure learning with HNMs, which arises from the NLL optimization issue discussed above. In DAG learning under HNMs, the training objective consists of a reconstruction loss (likelihood) and an acyclicity constraint, which together encourage the learning of a DAG that accurately reflects the data. Under this objective, both components are expected to sufficiently contribute to the parameter updates. However, the consequences of the NLL optimization issue for DAG learning, where the reconstruction term and the acyclicity constraint coexist, have not yet been explored. We observe that, as the gradient of the reconstruction loss becomes heavily attenuated, the acyclicity constraint starts to dominate the parameter updates even from the early stages of training. When the acyclicity constraint dominates the updates from the beginning, the structure parameters are updated mainly in the direction that satisfies the acyclicity constraint, without sufficiently incorporating the data reconstruction signal, which hinders effective structure learning. This paper analyzes and formalizes this failure mode (see Figure 1).

To address this issue, we propose a graduated optimization strategy based on weighted loss scheduling to (i) start from a surrogate loss and smoothly transition to the standard NLL, and (ii) gradually activate the acyclicity constraint. In early training, we set the scheduling coefficient high initially to the surrogate loss that decomposes the standard NLL into an easier-to-optimize form. This, in turn, ensures that the updates of the DAG parameters are primarily driven by the data reconstruction signal. As training progresses, we gradually decrease the scheduling coefficient, so that the surrogate loss smoothly transitions to the NLL loss and the acyclicity constraint increasingly enforces an acyclic structure. In that it starts from a simple surrogate loss and then transitions to the standard NLL, our method is in the spirit of homotopy-based DAG learning (Deng et al., 2023), but extends it from the simple linear and bivariate settings to a nonlinear multivariate HNM regime, which is more realistic and challenging. In this case, global optimality is no longer guaranteed, but we formalize sample-dependent conditions under which the proposed optimization strategy is sufficient to improve DAG recovery, and verify them empirically. We also show that our method induces a much more stable optimization trajectory than standard NLL training, and empirically outperforms existing baselines in terms of structure learning performance on both synthetic and real datasets.

## 2 PRELIMINARIES

### 2.1 BACKGROUND

Let $X = [X_1, \ldots, X_d]$ be a $d$-dimensional random vector, and let $x^{(n)} = [x_1^{(n)}, \ldots, x_d^{(n)}] \in \mathbb{R}^d$ denote the $n$-th sample of $X$. Given observed data $\mathbf{X} \in \mathbb{R}^{N \times d}$ with $N$ samples, the underlying causal structure can be represented as a directed acyclic graph (DAG) $\mathcal{G} = (\mathcal{V}, \mathcal{E})$, where $\mathcal{V} = \{1, \ldots, d\}$ is a set of nodes corresponding to $d$ variables, and $\mathcal{E}$ is the set of directed edges with no cycles. For each node $j \in \mathcal{V}$, we define the parent set of $j$, $pa(j) = \{i \in \mathcal{V} \mid (i \rightarrow j) \in \mathcal{E}\}$, so that $X_{pa(j)}$ denotes parent variables and direct causes of $X_j$. Such relationships among variables can be encoded in a weighted matrix $W \in \mathbb{R}^{d \times d}$. We assume the data are generated by a structural equation model (SEM) with the following *heteroscedastic* additive noise (Duong & Nguyen, 2023):

**Definition 1** (Heteroscedastic noise model). *A data vector $X$ follows Heteroscedastic Noise Models (HNMs) if, for each node $j \in \mathcal{V}$,*

$$X_j = \mu_j(X_{pa(j)}) + \sigma_j(X_{pa(j)}) \cdot E_j, \quad \text{where } (E_1, \ldots, E_d)^\top \sim \mathcal{N}(\mathbf{0}, \mathbf{I}_d). \tag{1}$$

Here, $\mu_j : \mathbb{R}^{|pa(j)|} \to \mathbb{R}$ is a nonlinear deterministic mean function, and the scale function $\sigma_j : \mathbb{R}^{|pa(j)|} \to \mathbb{R}_{>0}$ adjusts the variance of the exogenous noise $E_j$ depending on the values of $X_{pa(j)}$. That is, we assume that $E_1, \ldots, E_d$ are mutually independent standard Gaussians, but their variances are scaled differently for each variable and observation by $\sigma_j(X_{pa(j)})^2$.

**Theorem 2.1** (Identifiability of HNMs (Yin et al. (2024), restated)). *Under HNMs, if $\mu(\cdot)$ is a nonlinear function, $\sigma(\cdot)$ is a piecewise function, and noise variables $E$ are independent and Gaussian, then the true causal graph $W^*$ is identifiable from the joint distribution of the observed variables $P_X$. Its proof is provided in Appendix F.*

**Definition 2** (Global solution equivalence class). *Let $P_{X^*}$ be the true data distribution generated by the ground-truth DAG $W^*$. The global solution equivalence class (Deng et al., 2024) $\mathcal{S}(P_{X^*})$ is the set of all model parameters $(\Theta, W)$ that perfectly reproduce this distribution, thereby achieving the minimum possible population risk.*

Under the conditions of Theorem 2.1, for any parameter set $(\Theta, W) \in \mathcal{S}(P_{X^*})$, the graph structure $W$ must be the true DAG $W^*$. This establishes the existence of a unique global solution, providing a clear theoretical target for our proposed method. Although $W^*$ may exist, it is generally difficult to find it when dealing with finite samples. We analyze how the algorithm actually behaves on finite samples under HNMs and focus on this issue.

Under HNM, the conditional distribution of $X_j$ given $X_{pa(j)}$ follows $p(X_j|X_{pa(j)}) \sim \mathcal{N}(\mu_j(X_{pa(j)}), \sigma_j^2(X_{pa(j)}))$ due to $\mathbb{E}[E_j|X_{pa(j)}] = 0$ and $Var[E_j|X_{pa(j)}] = 1$. Based on this conditional distribution, the Gaussian negative log-likelihood (NLL) for the entire data (over all $d$ variables) can be defined as the reconstruction loss with the mean function $\mu_j$ and the variance function $\sigma_j$ parameterized by $\Theta = \{\theta_\mu, \theta_\sigma\}$:

$$\mathcal{L}_{\text{NLL}}(\Theta, W) = \sum_{j=1}^{d} \left[ \frac{1}{2} \log \sigma_j^2(X; W_j, \theta_\sigma) + \frac{(X_j - \mu_j(X; W_j, \theta_\mu))^2}{2\sigma_j^2(X; W_j, \theta_\sigma)} \right], \quad (2)$$

where $W_j \in \mathbb{R}^d$ represents the directed connectivity from $X_{pa(j)}$ to $X_j$. Since the matrix $W$ must represent a DAG, the smooth acyclicity constraint $h(W) = \text{tr}(e^{W \circ W}) - d$, where $W \circ W$ denotes the Hadamard (element-wise) product of $W$, should be imposed (Zheng et al., 2018); enforcing $h(W) = 0$ yields the form of an equality-constrained optimization, and this can be solved via the Augmented Lagrangian Method (ALM) (Bertsekas, 2014; Zheng et al., 2018):

$$\min_{W, \Theta} \mathcal{L}_{\text{NLL}}(\Theta, W) + \frac{\rho}{2} h(W)^2 + \alpha h(W), \quad (3)$$

where $\rho > 0$ is the penalty coefficient and $\alpha$ is the Lagrange multiplier. As $h(W) \to 0$, the learned $W$ becomes an acyclic structure. The parameterization and use of $W$ in our work follow Zheng et al. (2020) and Yin et al. (2024), with further implementation details provided in Appendix H.5.

## 2.2 MOTIVATION

**Overestimated predictive variance due to joint optimization.** Skafte et al. (2019) and Seitzer et al. (2022) showed that directly optimizing a heteroscedastic model's NLL loss to jointly learn both the mean and variance functions with a neural network can cause the predicted variance to dominate the gradients, hindering accurate mean fitting. This can be understood by examining the gradients of the NLL loss with respect to the mean and variance:

$$\nabla_\mu \mathcal{L}_{\text{NLL}} = \frac{X_j - \mu_j(X; W_j, \theta_\mu)}{\sigma_j^2(X; W_j, \theta_\sigma)}, \quad \nabla_\sigma \mathcal{L}_{\text{NLL}} = \frac{\sigma_j^2(X; W_j, \theta_\sigma) - (X_j - \mu_j(X; W_j, \theta_\mu))^2}{2\sigma_j^4(X; W_j, \theta_\sigma)}. \quad (4)$$

Because the gradient $\nabla_\mu L_{\text{NLL}}$ is inversely proportional to the predictive variance, samples for which the model predicts a smaller variance have a disproportionately larger impact on the parameter updates. Consequently, instead of improving its accuracy on hard-to-predict regions, the model tends to reduce overall loss by increasing the predicted variance in regions that are already well-fit or inflating the variance for more difficult samples (Seitzer et al., 2022; Takahashi et al., 2018). To address this issue, several techniques have been proposed in the heteroscedastic learning literature. Skafte et al. (2019) introduced an alternating optimization scheme for the two (mean and variance)

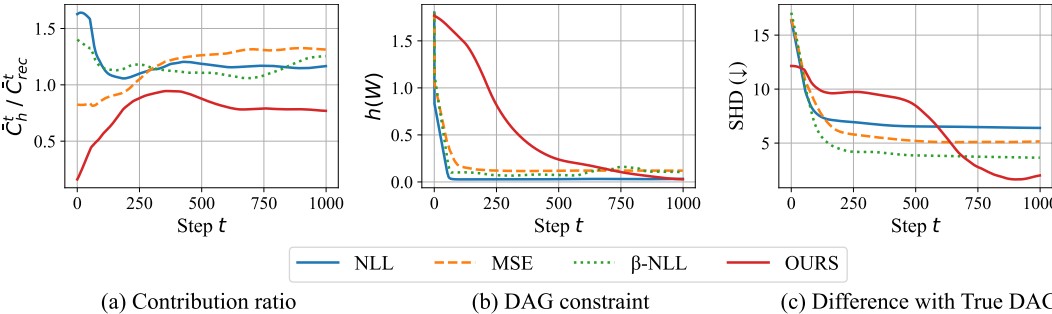

(a) Contribution ratio         (b) DAG constraint         (c) Difference with True DAG

Figure 1: (a) The cumulative contribution ratio $\bar{C}_h(t)/\bar{C}_{\mathrm{rec}}(t)$; a value above 1 indicates that $h(W)$ contributed more to the update of $W$ than the reconstruction loss. (b) Value of $h(W)$ during training steps. (c) Structural Hamming Distance between $W$ and the true DAG over training steps.

networks to effectively decouple the influence of variance estimation on the mean's learning. Seitzer et al. (2022) proposed a surrogate loss called $\beta$-NLL, which weights each data point's contribution by the $\beta$-th power of its predicted variance in order to reduce the effect of the variance term. Stirn et al. (2023) advocated for a decoupled architecture in which an encoder first produces a shared latent representation of the input, and then two separate networks use this representation to predict the mean and the covariance matrix—thereby isolating the mean estimation from direct variance interference.

**New challenge in DAG learning under HNM.** However, in the context of DAG learning under HNM, a new challenge arises when using NLL as the reconstruction loss. We focus on the balance between the NLL-based reconstruction term and the acyclicity constraint $h(W)$ that enforces a DAG structure. As Eq. (4) indicates, the predictive variance acts like an inverse weight on the gradient of the reconstruction loss with respect to the mean function—samples with higher predicted variance contribute much less to the mean updates. Consequently, during the initial phase of training, the influence of the NLL reconstruction loss on the update of $W$ is relatively weakened, and the acyclicity constraint $h(W)$ can come to dominate the learning. In other words, the learned structure $W$ ends up reflecting the acyclicity constraint more than the data reconstruction signal.

To empirically examine this phenomenon, we analyzed the contributions of the reconstruction loss and the acyclicity constraint to the updates of $W$ throughout training. Such influence can be quantified using the contribution score of Lan et al. (2019), which measures the dot product between the gradient of each loss and the corresponding parameter change at a given step. Let $\Delta W^{(s)} = W^{(s)} - W^{(s+1)}$ denote the change in $W$ at training step $s$, and let $g_{\mathrm{rec}}^{(s)}$ and $g_h^{(s)}$ be the gradients of the reconstruction loss and the acyclicity constraint at step $s$, respectively. We compute the dot product between them at every step, accumulate the values until $t$ steps, and then take an average. The resulting cumulative contributions ($\bar{C}_{\mathrm{rec}}(t)$ and $\bar{C}_h(t)$) can be expressed by:

$$\bar{C}_{\mathrm{rec}}(t) \;=\; \frac{1}{t}\sum_{s=0}^{t}\big|\langle-\Delta W^{(s)},\, g_{\mathrm{rec}}^{(s)}\rangle\big|, \qquad \bar{C}_h(t) \;=\; \frac{1}{t}\sum_{s=0}^{t}\big|\langle-\Delta W^{(s)},\, g_h^{(s)}\rangle\big|, \tag{5}$$

where $\langle\cdot\rangle$ denotes the dot product. Thus, for example, if $\bar{C}_h(t)$ exceeds $\bar{C}_{\mathrm{rec}}(t)$, it indicates that the acyclicity constraint has been the primary driver of the learning of $W$.

We examined NLL, MSE (Zheng et al., 2020), and $\beta$-NLL (Seitzer et al., 2022); for each reconstruction loss type, we measured three metrics over training steps: (a) the ratio $\bar{C}_h(t)/\bar{C}_{\mathrm{rec}}(t)$, (b) the value of the acyclicity constraint $h(W)$, and (c) the Structural Hamming Distance (SHD) between $W$ and the ground-truth DAG at each step (lower SHD is better). The experiments were conducted on the synthetic data from Erdős–Rényi (ER) graph under an HNM setting with $d=10$ (see Appendix H.6 for detailed settings). In Figure 1(a), for MSE, we observe $\bar{C}_h(t) > C_{\mathrm{rec}}(t)$ at around 250 steps; for NLL and $\beta$-NLL, the ratio exceeds 1, indicating that the acyclicity constraint contributed more to $W$ updates than the reconstruction loss even from the very beginning of training. This trend persisted throughout training. In Figure 1(b), they force $h(W)$ toward zero extremely quickly at the early phase of learning. Consistent with this, Figure 1(c) shows that their SHD curves quickly flattened out and remained nearly constant thereafter, implying that the structure has ceased to change because SHD is a metric highly sensitive to edge additions, deletions, and reversals.

**Remark.** As a result, if the acyclicity constraint dominates too early, the structure can get prematurely *locked-in* by the constraint instead of being learned from the data. This effect may become more pronounced as the penalty on the acyclicity constraint increases. Therefore, we propose a training strategy that ensures $W$ is guided by data reconstruction in the early stages, and later *progressively enforces* the acyclicity constraint. As shown in Figure 1, our method (a) pays more attention to the reconstruction loss in the early stages, guiding $W$ to prioritize data likelihood, (b) shows higher $h(W)$ in the early epochs but gradually pushes $h(W) \to 0$ over time, and thus (c) achieves the lowest SHD.

## 3 PROPOSED METHOD

The optimization issue observed in the previous section arises because the variance function is overestimated before the mean and variance functions are sufficiently trained, causing the contribution of the NLL loss to updates of $W$ to be diminished. We thus propose an optimization strategy tailored to this challenge specific to DAG learning with HNMs, mainly inspired by the homotopy-based approaches (Ko et al., 2023; Deng et al., 2023). In the early stages, our strategy induces stable learning of the mean and variance functions by focusing on a surrogate loss, and reduces the influence of the acyclicity constraint. Subsequently, we gradually transition from the surrogate loss to the standard NLL loss; the influence of the constraint gradually increases so that $W$ converges to a DAG while sufficiently reflecting the data likelihood.

As noted in the previous section, directly minimizing the NLL loss in a joint optimization presents challenges; therefore, we introduce a surrogate loss that decouples the mean and variance components of the NLL loss. Let $t \in [0, T]$ denote the current training step and $T$ be the time at which training terminates due to convergence of the penalty term. The overall loss function is defined over $t$ as:

$$\mathcal{L}_{\text{total}}(\Theta, W) = \begin{cases} \mathcal{L}_{\text{surrogate}}(\Theta, W, t) + \frac{\rho}{2}h(W)^2 + \alpha h(W), & \text{if } t < t^*, \\ \mathcal{L}_{\text{NLL}}(\Theta, W) + \frac{\rho}{2}h(W)^2 + \alpha h(W), & \text{otherwise}, \end{cases} \quad (6)$$

where $\mathcal{L}_{\text{surrogate}}(\Theta; t)$ denotes the surrogate loss for the initial training phase and $t^*$ denotes the pre-defined transition point. For $t < t^*$ (the early phase), we prioritize the surrogate loss to ensure stable convergence of the predictors, then smoothly transition to the NLL loss as $t$ approaches $t^*$. In what follows, Section 3.1 details how the surrogate loss facilitates stable initial convergence of the mean and variance functions and the process of transitioning to the NLL loss, and Section 3.2 discusses the rationale behind our surrogate loss and optimization strategy.

### 3.1 GRADUAL TRANSITION OF THE OBJECTIVE

Our surrogate loss serves as an initial training objective to prevent $W$ from being pushed in a wrong direction by unstable likelihood gradients, while also acting as a bridge toward the eventual NLL loss. It is defined as:

$$\mathcal{L}_{\text{surrogate}}(\Theta, W, t) = \lambda_{\text{reg}}(t) \cdot \left( (\mathcal{L}_{\text{MSE}}(\Theta_\mu, W) + \mathcal{L}_{\text{VarReg}}(\Theta, W)) + \mathcal{L}_{\text{StopNLL}}(\Theta, W) \right), \quad (7)$$

where $\lambda_{\text{reg}}(t)$ is a smooth scheduling coefficient that starts at a high value in early training, so that the loss terms aligning the mean and variance estimates are given greater weight, and gradually decays to ensure a stable transition. $\lambda_{\text{reg}}(t)$ is defined as:

$$\lambda_{\text{reg}}(t) = \lambda_{\text{reg}}(0) \cdot \exp\left(-\frac{t}{t^*/\tau}\right), \quad \lambda_{\text{reg}}(0) \gg 1, \quad (8)$$

where $\tau$ controls the rate of this decay.

**Decoupling the NLL components.** By design, when $t$ is small ($\lambda_{\text{reg}}(t)$ is large), the loss terms $\mathcal{L}_{\text{MSE}}$ and $\mathcal{L}_{\text{VarReg}}$ dominate the objective. This phase is intended to train the mean and variance functions independently, recognizing that joint optimization at the early steps can impede the convergence of each function. Prior work (Stirn et al., 2023; Skafte et al., 2019) showed that reliable learning of the variance function strongly depends on a good fit of the mean function. If the mean function has not yet sufficiently converged, the prediction residuals will not accurately reflect the data distribution and will be biased; a variance network that takes these biased residuals as input is at risk of learning a distorted signal. Therefore, in the early stage of training, we induce the mean function to converge independently via $\mathcal{L}_{\text{MSE}}$, and we accurately calibrate the variance function via residual-based regularization loss $\mathcal{L}_{\text{VarReg}}$. First, $\mathcal{L}_{\text{MSE}}$ is computed by:

$$\mathcal{L}_{\text{MSE}}(\Theta_\mu, W) = \sum_{j=1}^{d} (X_j - \mu_j(X; W_j, \theta_\mu))^2 \,. \tag{9}$$

Using the standard NLL loss too early (before the mean has converged) can cause the variance estimator to inflate its predictions to trivially reduce the loss (Seitzer et al., 2022). To mitigate this, we introduce $\mathcal{L}_{\text{VarReg}}$, which serves to regularize the variance in the initial phase such that the variance function reflects the actual magnitude of the prediction error:

$$\mathcal{L}_{\text{VarReg}}(\Theta, W) = \sum_{j=1}^{d} \left( \sigma_j^2(X; W_j, \theta_\sigma) - (X_j - \mu_j(X; W_j, \theta_\mu))^2 \right)^2 \,. \tag{10}$$

This regularization term encourages the predicted variance to align with the observed residual magnitude, thereby preventing the variance from growing excessively when the mean function has not yet converged. In the initial phase of training, $\mathcal{L}_{\text{MSE}}$ improves the accuracy of the mean prediction, and $\mathcal{L}_{\text{VarReg}}$ keeps the predicted variance from overestimating the mean error. Together, these two loss terms ensure that each function can converge stably. They are applied with a large initial weight $\lambda_{\text{reg}}(t)$, promoting better data reconstruction of the predictor networks. In Appendix C.2, we provide a justification and analysis of these two loss terms based on gradient consistency and an analysis on curvature and condition number.

**Stop-gradient NLL.**   Despite the stabilizing effect of $\mathcal{L}_{\text{VarReg}}$, this term alone does not guarantee that the predictive variance reflects the true conditional variance. For instance, if we simply carry over the variance learned by $\mathcal{L}_{\text{VarReg}}$ into the NLL loss, the overall loss can still remain high – indicating that the predicted variance is not satisfying the likelihood-optimal conditions. Since the ultimate goal of structure learning is to optimize both the NLL and the acyclicity constraint simultaneously, the variance network should eventually be oriented toward minimizing the NLL as well. To achieve this, we introduce a modified NLL loss with a stop-gradient operation as follows:

$$\mathcal{L}_{\text{StopNLL}}(\Theta, W) = \sum_{j=1}^{d} \left[ \frac{1}{2} \log \sigma_j^2(\lfloor X; W_j \rfloor, \theta_\sigma) + \frac{(X_j - \lfloor \mu_j(X; W_j, \theta_\mu) \rfloor)^2}{2\sigma_j^2(\lfloor X; W_j \rfloor, \theta_\sigma)} \right], \tag{11}$$

where $\lfloor \cdot \rfloor$ denotes the stop-gradient operation. Thus, $\mathcal{L}_{\text{StopNLL}}$ does not back-propagate through the mean function or the structural parameters (including $W$) and only updates the variance function parameters. By doing so, we design a scenario where the variance can be trained reliably according to the NLL criterion even if the mean function has not yet converged to its optimal estimate. This addresses the limitation of $\mathcal{L}_{\text{VarReg}}$: whereas $\mathcal{L}_{\text{VarReg}}$ adjusts the variance based on the current residual (which can be distorted if the mean is inaccurate), $\mathcal{L}_{\text{StopNLL}}$ blocks this indirect pathway. As a result, the variance estimation is not influenced by the mean function's biased predictions.

**Transition to the final objective.**   Once the variance estimates have been sufficiently stabilized via $\mathcal{L}_{\text{StopNLL}}$, we switch the overall loss back to the original NLL loss. At this point, we jointly optimize both the mean and variance functions, and the DAG constraint is now actively imposed in the loss, guiding the model to learn an acyclic structure based on the learned representations. This transition is handled smoothly by the scheduling coefficient (i.e. by the decay of $\lambda_{\text{reg}}(t)$). Appendix C.1 and I.8 analyzes the convergence characteristics and time complexity of our method, respectively. Appendix D provides the results of the contribution ratio analysis with respect to $t^*$ and $\lambda_{\text{reg}}(0)$ and the trajectory analysis of the training process. The overall training procedure is in Appendix G.

## 3.2   ANALYSIS OF WEIGHTED LOSS SCHEDULING

We define the standard training objective that combines the NLL loss and the ALM-based acyclicity constraint as follows:

$$r_{\rho,n}^{\text{NLL}}(W) := \inf_\Theta \left\{ \widehat{\mathbb{E}}\big[ \mathcal{L}_{\text{NLL}}(\Theta, W) \big] + \frac{\rho}{2} h(W)^2 + \alpha\, h(W) \right\}, \tag{12}$$

where $\widehat{\mathbb{E}}$ denotes the empirical expectation. The challenge is the inherent instability that arises when directly optimizing $r_{\rho,n}^{\text{NLL}}(W)$. As discussed in Section 2.2, in the early stage of training, the variance-attenuation phenomenon weakens the reconstruction gradient ($g_{\text{rec}}$). This weakened signal is easily overwhelmed by the constraint gradient ($g_h$) as the penalty coefficient $\rho$ in the ALM gradually

increases, leading to a situation where the DAG structure is determined prematurely by the acyclicity constraint rather than by the data. This failure mechanism can be quantitatively characterized by a *failure certificate* (using the cumulative contribution measure defined in Eq. (5)) as follows:

**Definition 3** (Failure certificate). *If there exist constants $T_0 > 0$, $\gamma > 1$, and $\varepsilon \geq 0$ such that*

$$\frac{C_h(t)}{C_{\text{rec}}(t)} \geq \gamma, \quad (0 \leq t \leq T_0), \qquad h(W_{T_0}) \leq \varepsilon, \tag{13}$$

*we call the interval $[0, T_0]$ a constraint-dominant early phase and the tuple $(T_0, \gamma, \varepsilon)$ a failure certificate.*

As empirically observed in Figure 1, optimizing the standard objective $r_{\rho,n}^{\text{NLL}}(W)$ entails the aforementioned failure certificate, which becomes a fundamental cause hindering the search for the optimal DAG structure. To address this issue, we introduce a surrogate loss and a weight scheduling scheme to enforce a *reconstruction-dominant* phase in the early stage of training. The condition required for our proposed method to achieve a reconstruction-dominant phase is provided in Appendix D.

**Lemma 3.1** (Sufficient condition for an early reconstruction-dominant phase). *If, during the initial phase $t \leq T_1$, the following holds:*

$$\lambda_{\text{reg}}(t) \left\| \nabla_W \big( \mathcal{L}_{\text{MSE}} + \mathcal{L}_{\text{VarReg}} \big) \right\| \geq \frac{\kappa}{1-\delta} \left\| \nabla_W \big( \tfrac{\rho}{2} h(W)^2 + \alpha h(W) \big) \right\| \quad (0 < \delta < 1), \tag{14}$$

*then the gradient contribution ratio is bounded as:*

$$\frac{C_h(t)}{C_{\text{rec}}(t)} \leq 1 - \delta \qquad (\forall\, t \leq T_1), \tag{15}$$

*where $\delta \in (0, 1)$ is the safety margin. This reverses the failure-certificate condition and guarantees a reconstruction-dominant phase in early training.*

Due to the limited space, its proof can be found in Appendix E.4, and we empirically verify the sufficient condition in Appendix D.1. Lemma 3.1 formalizes the conditions under which the early training phase is reconstruction-dominant and provides a theoretical handle for understanding how our method operates. Guaranteeing such a reconstruction-dominant phase at the beginning of training has a direct impact on the quality of the final solution. We denote by $\xi_{\text{NLL}}$ the minimal excess empirical risk achievable with the standard objective $r_{\rho,n}^{\text{NLL}}$, and by $\xi_{\text{ours}}$ the minimal excess empirical risk achievable with our method. Based on Lemma 3.1 and on the stability of optimization analysis in Appendix D.2, Claim 1 below shows that, for the same number of samples, $\xi_{\text{ours}} < \xi_{\text{NLL}}$ holds.

**Claim 1** (Sample-dependent sufficient condition). *Under the standard assumptions (i)–(iv) summarized in Appendix E.6 (namely, uniform convergence of $r_{\rho,n}$ to $r_\rho$, a local PL-type condition along the data-induced solution path, a slow schedule $\lambda_{\text{reg}}(t)$ that maintains a stable homotopy path, and the existence of an early reconstruction-dominant phase as defined in Lemma 3.1), the proposed optimization strategy achieves a smaller excess risk than standard NLL training:*

$$\xi_{\text{ours}} \leq C_1, \Delta_n + C_2, \frac{\kappa, |\nabla_W \big( \tfrac{\rho}{2} h(W)^2 + \alpha h(W) \big)|^2}{\lambda_{\text{reg}}(t_1), |\nabla_W (\mathcal{L}_{\text{MSE}} + \mathcal{L}_{\text{VarReg}})|^2} < \xi_{\text{NLL}}. \tag{16}$$

*Its assumptions and proof are discussed in detail in Appendix E.6. Claim 1 should be viewed not as a general guarantee but as an idealized result that provides intuition under these standard assumptions.*

## 4 EXPERIMENTAL RESULTS

We evaluated the performance of the proposed method on both synthetic datasets designed to include heteroscedastic noise and four real-world datasets. In all experiments, we consistently used the same values for $\lambda_{\text{reg}}(0)$, $t^*$, and $\tau$. Appendix H reports more detailed experimental settings.

### 4.1 RESULTS ON SYNTHETIC AND REAL-WORLD DATASETS

Figure 2 shows the results on the synthetic datasets generated under both homoscedastic and heteroscedastic settings. The proposed method achieves performance comparable to or better than

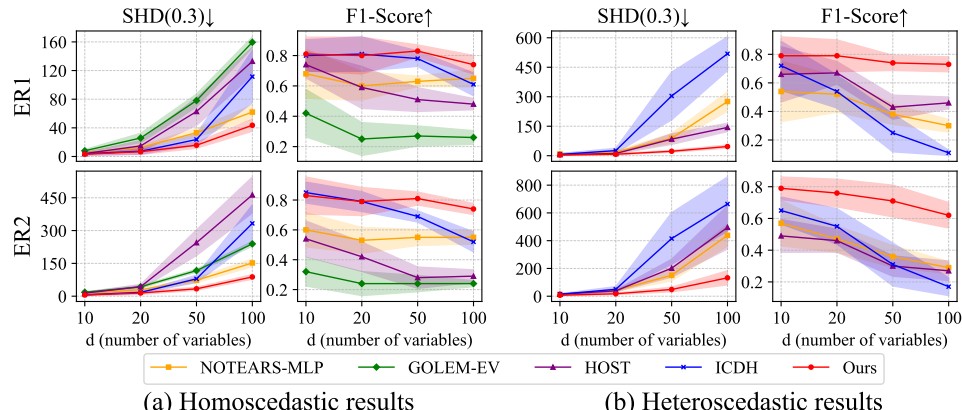

(a) Homoscedastic results          (b) Heteroscedastic results

Figure 2: Comparison results on the synthetic datasets. The colored lines show the mean value across 10 runs, and the shaded areas around each curve indicate the corresponding standard error.

Table 1: Results on the real-world datasets.

| Method | Sachs ($d = 11$) | | SynTReN ($d = 20$) | | CausalAssembly ($d = 40$) | |
|---|---|---|---|---|---|---|
| | SHD($\downarrow$) | F1-Score($\uparrow$) | SHD($\downarrow$) | F1-Score($\uparrow$) | SHD($\downarrow$) | F1-Score($\uparrow$) |
| NOTEARS-MLP | 15 | 0.26 | 114.6 | 0.13 | 139.4 | 0.06 |
| HOST | 13 | 0.33 | 114.5 | 0.15 | 183.5 | **0.17** |
| ICDH | 13 | 0.33 | 128.4 | 0.13 | 199.0 | 0.10 |
| **Ours** | **12** | **0.43** | **89.8** | **0.18** | **135.6** | 0.11 |

existing methods in the homoscedastic environment, and in the heteroscedastic environment it consistently achieves the best performance on all evaluation metrics for all tested values of $d$ (the numbers of variables). This suggests that our method attains high accuracy in structure learning while effectively suppressing spurious (false positive) edge detection. Notably, even as the number of variables $d$ increases, our method's performance degrades very gradually (e.g., its F1-score drops from 0.79 at $d = 10$ to 0.73 at $d = 100$ for ER1 (and to 0.62 for ER2)). In contrast, HOST's F1-score exhibits a sharp decline in F1-score as $d$ grows (under the heteroscedastic ER2 condition with $d = 100$); ICDH's F1-score also plunges to 0.17. We interpret these differences as follows: existing methods, due to the instability of variance estimation in heteroscedastic settings, tend to overestimate the variance in order to compensate for mean prediction errors, which in turn leads to many spurious edges. In contrast, our method effectively decouples the learning of the mean and variance functions through the surrogate loss and the graduated loss weighting strategy, preventing edge detection that is not grounded in the data during the early stages of training and enabling more reliable structure estimation. Appendix I.1 reports the result on ER3, ER4 and ER5 (dense graphs) and Appendix I.2 shows the results in an extreme case, $d = 1000$, where our method also achieves the best performance.

Table 1 presents the comparative results on the real-world datasets. The proposed method achieves the best performance in terms of both SHD and F1-score, with only one exceptional case in the CausalAssembly dataset. This shows that the proposed graduated optimization strategy can effectively estimate the DAG structure on real-world scenarios. Appendix I.3 reports the result on another real-world dataset, Cause-Effect Pairs (Sgouritsa et al., 2015); we also achieve the best here.

### 4.2 Further analysis

**Ablation study.** We analyzed the contribution of each component of the surrogate loss to the DAG learning performance. ER1 with $d \in \{10, 20, 50\}$ under HNM was used here. We compared the full proposed model (with all components of the surrogate loss) to ablated versions where we removed each of $\mathcal{L}_{\text{MSE}}$, $\mathcal{L}_{\text{VarReg}}$, or $\mathcal{L}_{\text{StopNLL}}$. Figure 3(a) indicates that removing $\mathcal{L}_{\text{MSE}}$ yields the highest SHD (worst performance). For the NLL loss to properly learn the variance, the mean function must be sufficiently converged first. If the MSE term is removed, due to a large discrepancy between the predictive mean and true values, the variance prediction becomes distorted, and overall performance degrades. The $\mathcal{L}_{\text{VarReg}}$ and $\mathcal{L}_{\text{StopNLL}}$ terms likewise play important roles in the initial stage by stabilizing variance estimation and preventing its overestimation; removing either of them also leads to performance deterioration. By contrast, with all three components, our method achieves the lowest

SHD across all conditions. Therefore, we can confirm that each component of our surrogate loss is crucial for effective structure learning. As an analysis of the effect of the graduated optimization, Appendix I.5 shows the result of applying it to NOTEARS-MLP; this helps improve performance, but our full framework that uses the surrogate loss achieves even higher performance. In addition, Appendix I.6 shows the influence of the scheduling parameters, showing that increasing $t^*$ and $\lambda_{\text{reg}}(0)$ stabilizes the optimization and improves structure learning performance, but if their values become too large, performance deteriorates. This result is consistent with our overall design premise.

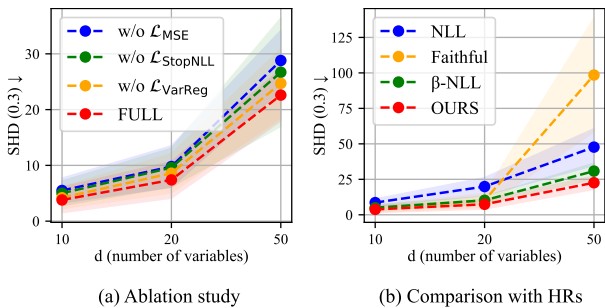

(a) Ablation study  (b) Comparison with HRs

Figure 3: (a) ablation study. (b) comparison with the strategies in heteroscedastic regression (HR).

**Application of the strategies in heteroscedastic regression.** In addition, we investigated whether existing optimization solutions in the context of heteroscedastic regression can be directly applied to our DAG learning problem. The first issue we identified is that naively optimizing the NLL is difficult from an optimization perspective under HNM. Faithful Heteroscedastic (Stirn et al., 2023) and $\beta$-NLL (Seitzer et al., 2022) are the existing approaches to address this issue. However, according to our hypothesis, these methods do not tackle problems specific to structure learning—such as the dominance of the acyclicity constraint over the NLL loss in learning $W$ or the initial stabilization of variance estimation—and thus their performance in DAG learning could be limited. We empirically validated this by comparing models trained with the standard NLL, with $\beta$-NLL, and with ours, under the same experimental setting.

Figure 3(b) shows the SHD results on ER1 with $d \in \{10, 20, 50\}$ under HNM. The plain NLL baseline has the highest SHD for $d = 10$ and 20, whereas at $d = 50$ Faithful Heteroscedastic performs the worst. $\beta$-NLL performs relatively well compared to the other two. While Faithful Heteroscedastic and $\beta$-NLL may stabilize variance learning, (as we expected) they do not consider the influence of the acyclicity constraint $h(W)$ during training, and consequently their DAG estimation performance is worse than that of our method. As a result, the proposed surrogate-loss-based graduated optimization strategy is specifically designed to address the challenges of DAG learning under heteroscedasticity, which sets it apart from existing heteroscedastic regression methods. Appendix I.4 shows an experiment where our optimization strategy is applied to the existing NLL losses.

## 5 LIMITATIONS AND CONCLUSIONS

Previous studies in machine learning optimization (Hazan et al., 2016; Gargiani et al., 2020) theoretically demonstrated the effectiveness of an approach that gradually transitions from a smooth approximation of a complex objective function, whose effectiveness was also shown in bivariate and linear cases of DAG learning (Deng et al., 2023). However, for the nonlinear neural network–based models addressed in this study, due to the non-convex structure, it is generally difficult to guarantee a global optimum, and only a stationary point can be guaranteed. In this case the parameter set for DAG learning includes all MLP weights, so parameter-level identifiability is not achieved. Nevertheless, under proper scheduling conditions we empirically demonstrated that our strategy reliably reaches a good stationary point and leads to more accurate DAG recovery.

In conclusion, we proposed a new optimization strategy for DAG learning under heteroscedastic noise. We analyzed the training instability and structure distortion issues of the existing NLL-based approach, and introduced a weighted loss scheduling-based graduated optimization scheme to mitigate these problems. In the early stage of training, the surrogate loss induces stable convergence of the mean and variance function estimators; as training progresses, we gradually transition to the standard NLL objective with the acyclicity constraint, thereby maximizing the structure learning performance. Our method achieved impressive performance on both synthetic and real datasets.

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

## A APPENDIX

In this **Appendix**, we provide supplementary details, additional experimental results, and further analyses to support the claims and findings presented in the main paper. Our code and data can be found at: https://anonymous.4open.science/r/GO-DAG-3BB1/

## B RELATED WORK

### B.1 CLASSICAL CAUSAL DISCOVERY METHODS

The gold standard for identifying causal factors is randomized controlled trials, but in practice these face constraints of time, cost, and ethics. A more realistic setting is to learn DAGs from observational data, and research in this area has proceeded along two strands: constraint-based methods and score-based methods (Spirtes & Zhang, 2016; Glymour et al., 2019). Constraint-based methods (e.g., PC (Spirtes et al., 2000), FCI (Spirtes et al., 1995; Colombo et al., 2012)) build the causal skeleton via independence tests and determine edge orientations. Score-based methods (e.g., GES (Chickering, 2002; Teyssier & Koller, 2005)) maximize a score that balances data fit and model complexity and

search the DAG space for an optimal graph. However these methods can identify only the Markov equivalence class (Guo et al., 2020). Subsequently, Functional Causal Models (FCMs), which assume data generation via structural equations with direct parents and noise, were introduced to identify the true causal structure within the equivalence class, including linear SEMs (Shimizu et al., 2006), Additive Noise Models (Peters et al., 2014), and Post-Nonlinear models (Zhang & Hyvarinen, 2012).

### B.2 Continuous optimization based methods.

However, classical methods rely on local heuristics to enforce acyclicity, which makes DAG discovery combinatorially challenging as the number of nodes increases. NOTEARS (Zheng et al., 2018) reformulated the original discrete DAG constraint into a continuous optimization problem solvable by gradient descent, becoming the standard optimization approach used in subsequent DAG learning methods. Follow-up work on linear SEMs, including GOLEM (Ng et al., 2020) which studies sparsity and acyclicity and directly minimizes likelihood, and DAGMA (Bello et al., 2022) which introduces a log-det acyclicity constraint, extended this line. Subsequent methods extended these approaches to nonlinear structures using neural function approximators, such as NOTEARS-MLP (Zheng et al., 2020) and GraN-DAG (Lachapelle et al., 2020). Other nonlinear methods include DAG-GNN (Yu et al., 2019) that deals with various data types using variational autoencoder parameterized by graph neural network and DAG-GAN (Gao et al., 2021) that considers DAG structure learning from the perspective of distributional optimization. DAG-NoCurl (Yu et al., 2021) proposes learning the DAG on the equivalent space of weighted gradients of graph potential functions.

### B.3 Different Noise assumptions

Representative methods such as ANMs with Gaussian noise assume homoscedastic noises, meaning equal noise variance across variables and observations. Ng et al. (2020) relax this by considering GOLEM-NV, where noise variances can differ by variable, while remaining constant across observations for a given variable; (Lachapelle et al., 2020) and (Park, 2020) make the same relaxation. When noise variance also differs across observations, one common approach in heteroscedasticity modeling is to relax independence between parent variables and additive noise. In the bivariate case, RECI (Blöbaum et al., 2018) assumes a joint distribution between cause and noise; HECI (Xu et al., 2022) explicitly assumes heteroscedastic noise with variance dependent on the cause, models it as a piece-wise function of the parent, and allows only a limited number of variance values. CAFEL (Khemakhem et al., 2021) and LOCI (Immer et al., 2023) introduce deterministic functions of the parent to adjust noise variance. To generalize to multivariate data, prior works first learn an undirected skeleton (e.g., PC (Spirtes et al., 2000)) then orient edges; HOST (Duong & Nguyen, 2023) estimates a causal order and then orient pair-wise directions. GraN-DAG++ (Lachapelle et al., 2020) allows noise variances to depend on parents in realistic settings. ICDH (Yin et al., 2024) is the first to incorporate heteroscedasticity into a continuous optimization framework for DAG learning and to model the noise distribution with neural networks.

**Detailed Comparison with ICDH.** Similar to Skafte et al. (2019), ICDH trains the mean network and the variance network in a two-phase alternating fashion. Specifically, it first trains the mean network and $W$ until convergence, then freezes $W$ and updates only the variance network, repeating this cycle. As a result, during the variance updates, the graph $W$ is kept fixed at the value learned in the previous phase, so the parameter updates of the variance function do not directly feed back into the choice of parent variables.

In contrast, our method uses the surrogate loss in Eq. (7) and the scheduling coefficient $\lambda_{\text{reg}}(t)$ in Eq. (8) to jointly optimize the mean, the variance, and $W$ throughout training. Since $W$ is shared by the mean and variance networks, the choice of parent variables is influenced by both the mean and variance estimates, which is consistent with the HNM formulation in Eq. (3), where the two networks share the same set of parents. In particular, to mitigate the failure mode in which the NLL loss and the DAG constraint interact harmfully in the early stages under the HNM assumption, the surrogate loss is designed to decouple the NLL into a mean term and a variance term, enabling stable joint optimization without significantly increasing computational cost.

### B.4 HETEROSCEDASTIC NOISE VS. HETEROGENEOUS DATA

Data observed in the real world often violate the equal-variance assumption, and various studies in causal discovery address data heterogeneity. To avoid confusion, we distinguish the heteroscedastic noise we focus on from heterogeneous data. Heteroscedastic noise means the noise variance depends on the parent variable (cause) (Xu et al., 2022). That is, the noise variance changes with the value of the input (cause), and consequently the variance differs across observations. By contrast, heterogeneous data refers to data in which the noise variables' distributions differ—typically distribution shifts across environments in multi-domain causal discovery (Ghassami et al., 2018; Wang et al., 2022; Zhang et al., 2023b; Li et al., 2024; Zhou et al., 2025). This includes cases where the noise variance is 1 in environment A and 4 in environment B, or where variances differ by variable within the same environment. In short, heteroscedastic noise means variance depends on the cause and thus also differs at the observation level, whereas heterogeneous data means different noise types and variances across variables and environments. This distinction also appears in prior work: DARING (He et al., 2021) explicitly considers variables with heterogeneous noises; ReScore (Zhang et al., 2023b) proposes a method for multi-domain and multi-group settings, consistent with MC (Ghassami et al., 2018), CD-NOD (Huang et al., 2020), DICD (Wang et al., 2022), and FedCDH (Li et al., 2024). MEE (Zhou et al., 2025) assumes noise distributions can differ both across and within environments, tackling a more challenging setting than prior work. We compare against these on HNM data; results appear in Appendix I.7.

## C DETAILS OF OUR SURROGATE LOSS

### C.1 CONVERGENCE ANALYSIS

To theoretically analyze the convergence characteristics of the surrogate loss employed as a substitute for the NLL in the early stage of training, we make the following assumption about the scheduling coefficient $\lambda_{\mathrm{reg}}(t)$ that controls the influence of the surrogate loss:

**Assumption 1** (smoothness of scheduling coefficient). *The scheduling function $\lambda_{\mathrm{reg}}(t)$, which is continuously differentiable over the interval $[0, t^*]$ is assumed to satisfy the following Lipschitz continuity condition:*

$$|\lambda_{\mathrm{reg}}(t + \Delta t) - \lambda_{\mathrm{reg}}(t)| \leq L_\lambda \Delta t, \quad \forall t, \Delta t > 0, \tag{17}$$

*where $L_\lambda > 0$ denotes a Lipschitz constant. We assume $\lambda_{\mathrm{reg}}(t) \to 0$ as $t \to t^*$, which implies that the influence of the surrogate loss diminishes as training progresses. Consequently, the overall training objective smoothly transitions from one based on the surrogate loss to the standard NLL objective.*

The gradual transition between losses induced by $\lambda_{\mathrm{reg}}(t)$ prevents abrupt changes in the optimization landscape, ensuring that the solution at each training step does not drastically deviate from the solution reached in the previous step. It thus helps the optimizer converge in a consistent direction. This behavior is similar to the locally linear tracking condition of the homotopy-SGD proposed in Gargiani et al. (2020).

**Monotonicity of loss for smooth objective transition.** We analyze the stability of the convergence path of the joint parameter sequence $\{(\Theta_t, W_t)\}$ depending on the objective transition. We use a quasi-Newton optimization algorithm such as L-BFGS, which chooses the step size at each iteration via a line search (Wright, 2006) satisfying the Wolfe conditions. As a result, the loss function decreases at every iteration, and if the loss is bounded below, the following monotonically decreasing condition holds:

$$\mathcal{L}_{\mathrm{surrogate}}(\Theta^{t+1}, W^{t+1}, t+1) < \mathcal{L}_{\mathrm{surrogate}}(\Theta^t, W^t, t). \tag{18}$$

The function $\mathcal{L}_{\mathrm{surrogate}}$ is continuous *w.r.t* the parameter sequence and has a lower bound, so it converges under the above condition. Additionally, since $\lambda_{\mathrm{reg}}(t)$ is Lipschitz continuous on the interval $[0, t^*]$ and $\lambda_{\mathrm{reg}}(t) \to 0$ as $t \to t^*$, the rate of change of the loss over time gradually decreases as training proceeds. As a result, the form of the overall objective function changes gradually: it starts from a simple form dominated by the surrogate loss and slowly transitions to the form of NLL with the acyclicity constraint. This gradual objective transition suppresses abrupt oscillations in the optimization trajectory and increases the stability of the convergence path. According to the analysis in Deng et al. (2023), such a gradual transition can lead to a stationary trajectory. Although our setting

is the general non-linear, multivariate setting (more complex than the bivariate-linear setting studied in Deng et al. (2023)), their observation that slowly decaying the coefficient leads to a smooth change in trajectory direction and preserves the continuity of the convergence trajectory can similarly apply to our proposed method.

**Convergence to a stationary point.** To show that the joint parameter sequence $\{(\Theta^t, W^t)\}$ converges to a stationary point set, following Yin et al. (2024), we apply the convergence theorem of Wu (1983). According to this theorem, convergence is guaranteed if the following two conditions are satisfied: (i) The parameter update mapping $\mathcal{M}$ is a closed point-to-set map on the complement of the stationary set $\mathcal{F}$, and (ii) If $(\Theta^{t-1}, W^{t-1}) \notin \mathcal{F}$, then the loss decreases at iteration $t$, *i.e.*, $\mathcal{L}_{\text{surrogate}}(\Theta^t, W^t, t) < \mathcal{L}_{\text{surrogate}}(\Theta^{t-1}, W^{t-1}, t-1)$, and thus the joint parameter sequence satisfies the monotonicity condition (Eq. (18)). Following our assumption, $\mathcal{L}_{\text{surrogate}}$ is continuous *w.r.t* the parameter sequence, and $\mathcal{M}$ (defined by the optimization trajectory) is a continuous point-to-set map, thus satisfying the closedness condition. Therefore, the joint parameter sequence $\{(\Theta^t, W^t)\}$ converges to the stationary set $F$. Once training has progressed sufficiently, $(\Theta^t, W^t)$ will lie in a region where the gradient norm of the loss is below $\varepsilon$, *i.e.*, within some $\varepsilon$-stationary point.

**Remark.** Once the early phase of training is over, the mean and variance networks have converged stably and $W$ is optimized within a representation that explains the data well. In the later stage, as the influence of the scheduling coefficient diminishes, the NLL loss and the acyclicity constraint fully come into effect, thereby facilitating effective DAG learning. This indicates that the graduated optimization, which gradually adjusts the influence of each loss term, alleviates instabilities that may occur in the early stage of training and helps the overall training trajectory converge stably to a stationary point. To maintain the continuity of such a stationary trajectory, the stationary point at each step must be isolated and the corresponding Jacobian must be invertible. Deng et al. (2023) theoretically showed that under this condition, the homotopy trajectory is preserved. Although a complete guarantee is difficult in our nonlinear and multivariate setting, the design of the scheduling coefficient $\lambda_{\text{reg}}(t)$ helps ensure that the trajectory continues smoothly without discontinuities or abrupt changes in the loss landscape.

## C.2 JUSTIFICATION FOR OUR SURROGATE

The proposed surrogate loss $\mathcal{L}_{\text{surrogate}}$ is designed to decouple the components of the NLL loss: the mean function is trained with an MSE loss and the variance function is trained with a residual-based regularization loss. In this subsection, we aim to demonstrate that the proposed loss function indeed serves as a surrogate for the NLL by comparing the gradients of these two loss terms with the gradient of the NLL loss.

**Gradient consistency.** We decouple the mean and variance components of the NLL and compute the gradient for each. To this end, we first define the NLL loss for a single sample as follows:

$$\mathcal{L}_{\text{NLL}}^{(j)} = \frac{1}{2} \log \sigma_j^2 - \frac{(X_j - \mu_j)^2}{2\sigma_j^2}, \tag{19}$$

where $\mu_j = \mu_j(X_j; W_j; \theta_\mu)$ and $\sigma_j^2 = \sigma_j^2(X_j; W_j; \theta_\sigma)$ denote the predictive mean and variance, respectively. The gradients of $\mathcal{L}_{\text{NLL}}^{(j)}$ with respect to $\mu_j$ and $\sigma_j$ are given by:

$$\nabla_{\mu_j} \mathcal{L}_{\text{NLL}}^{(j)} = -\frac{X_j - \mu_j}{\sigma_j^2}, \quad \nabla_{\sigma_j^2} \mathcal{L}_{\text{NLL}}^{(j)} = \frac{1}{2\sigma_j^2} - \frac{(X_j - \mu_j)^2}{2\sigma_j^4}. \tag{20}$$

The stationary point at which the gradients with respect to the mean and variance are zero is given by:

$$\nabla_\mu \mathcal{L}_{\text{NLL}} = -\frac{(X_j - \mu_j)}{\sigma_j^2} = 0 \quad \Rightarrow \quad \mu_j = X_j, \tag{21}$$

$$\nabla_\sigma \mathcal{L}_{\text{NLL}} = \frac{1}{2\sigma_j^2} - \frac{(X_j - \mu_j)^2}{2\sigma_j^4} = 0 \quad \Rightarrow \quad \sigma_j^2 = (X_j - \mu_j)^2.$$

Based on the above stationary point, we construct the surrogate loss with the following two terms:

$$\mathcal{L}_{\text{MSE}}^{(j)} = (X_j - \mu_j)^2, \quad \mathcal{L}_{\text{VarReg}}^{(j)} = \left( \sigma_j^2 - (X_j - \mu_j)^2 \right)^2. \tag{22}$$

The gradient of each loss term is as follows:

$$\nabla_{\mu_j}\mathcal{L}_{\text{MSE}}^{(j)} = -2(X_j - \mu_j), \quad \nabla_{\sigma_j^2}\mathcal{L}_{\text{VarReg}}^{(j)} = 2\left(\sigma_j^2 - (X_j - \mu_j)^2\right). \tag{23}$$

Now, assuming the surrogate loss has reached a stationary point, the corresponding predictive mean and variance are given by:

$$\nabla_{\mu_j}\mathcal{L}_{\text{MSE}}^{(j)} = 0 \quad \Rightarrow \quad \mu_j = X_j, \tag{24}$$

$$\nabla_{\sigma_j^2}\mathcal{L}_{\text{VarReg}}^{(j)} = 0 \quad \Rightarrow \quad \sigma_j^2 = (X_j - \mu_j)^2.$$

Substituting the above results into the gradient expressions of the NLL, we obtain:

$$\nabla_{\mu_j}\mathcal{L}_{\text{NLL}}^{(j)} = 0, \quad \nabla_{\sigma_j^2}\mathcal{L}_{\text{NLL}}^{(j)} = 0. \tag{25}$$

Thus, if the surrogate loss reaches a stationary point, the NLL loss is also at a stationary point. Meanwhile, as $\lambda_{\text{reg}}(t)$ decays over time and approaches zero, the influence of $\lambda_{\text{reg}}(t)$ in the surrogate loss vanishes and $\mathcal{L}_{\text{StopNLL}}$ becomes the dominant term. This term blocks gradients from propagating to the mean network, whereas for the variance network it operates in the same form as the NLL loss. Therefore, the gradient of the overall surrogate loss asymptotically aligns with the gradient of the NLL:

$$\|\nabla\mathcal{L}_{\text{surrogate}} - \nabla\mathcal{L}_{\text{NLL}}\| \to 0. \tag{26}$$

**Analysis of curvature and condition number.** We now analyze whether the surrogate loss is a better-conditioned function than the NLL loss by comparing the Hessians of each loss. First, for the NLL loss, the diagonal elements of the Hessian (i.e. the second derivatives) with respect to the mean and variance are as follows:

$$\nabla_{\mu_j}^2\mathcal{L}_{\text{NLL}}^{(j)} = \frac{1}{\sigma_j^2}, \quad \nabla_{\sigma_j^2}^2\mathcal{L}_{\text{NLL}}^{(j)} = -\frac{1}{2\sigma_j^4} + \frac{(X_j - \mu_j)^2}{\sigma_j^6}. \tag{27}$$

The above expressions indicate that as $\sigma_j^2$ becomes smaller, the magnitude of the second derivative may increase sharply, which can cause ill-conditioning; on the other hand, as $\sigma_j^2$ becomes larger, the magnitude of the second derivative can become very small, potentially causing the training to stall. This issue is particularly problematic in joint optimization, where there is a tendency for the variance to increase in order to reduce the NLL in the early stage of training.

In contrast, the second derivatives of each term composing the surrogate loss are as follows:

$$\nabla_{\mu_j}^2\mathcal{L}_{\text{MSE}}^{(j)} = 2, \tag{28}$$

$$\nabla_{\mu_j}^2\mathcal{L}_{\text{VarReg}}^{(j)} = 4\left(\sigma_j^2 - 3(X_j - \mu_j)^2\right),$$

$$\nabla_{\sigma_j^2}^2\mathcal{L}_{\text{VarReg}}^{(j)} = 2.$$

- For $\mu_j$, $\mathcal{L}_{\text{MSE}}^{(j)}$ is a quadratic function and is locally more convex than the NLL, with its second derivative always constant at 2. This means that the loss landscape in the $\mu_j$ direction has uniform curvature, enabling more stable gradient updates during the optimization process.

- Meanwhile, although the second derivative of $\mathcal{L}_{\text{VarReg}}^{(j)}$ does vary depending on the difference between $X_j$ and $\mu_j$, it exhibits relatively mild curvature with respect to changes in $\sigma_j^2$. In particular, even as $\sigma_j^2 \to 0$, this curvature does not diverge; and even if $\sigma_j^2$ becomes very large, the curvature does not approach 0, leading to a smooth training in both extremes.

- For $\sigma_j^2$, the second derivative of $\mathcal{L}_{\text{VarReg}}^{(j)}$ in the direction of $\sigma_j^2$ is constant at 2. Since the curvature in the $\sigma_j^2$ direction is constant, the loss landscape remains stable without excessive oscillation even when the predictive variance is very small in the early training stage. Moreover, the gradually applied $\mathcal{L}_{\text{StopNLL}}$ term prevents the unstable variance gradient in the early stages from affecting the mean trajectory, as the gradient with respect to $\mu_j$ does not propagate.

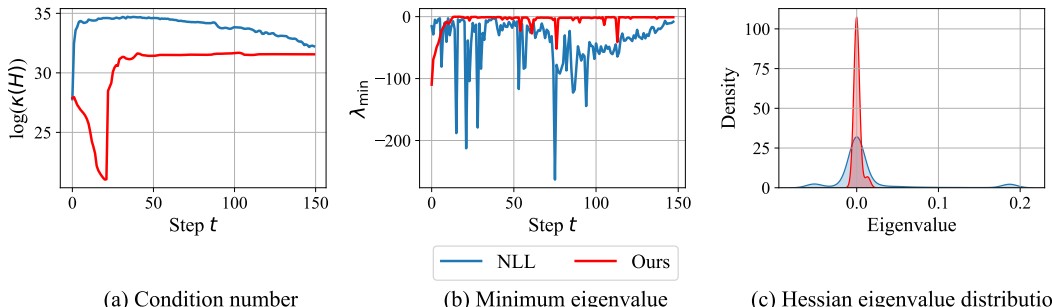

(a) Condition number      (b) Minimum eigenvalue      (c) Hessian eigenvalue distribution

Figure 4: Hessian matrix analysis with respect to our surrogate and the NLL loss. (a) shows the condition number at each step, where a log scale is applied. (b) compares the minimum eigenvalue ($\lambda_{\min}$) of the Hessian matrix at each step. (c) draws the distribution of eigenvalues of the Hessian matrix calculated after training.

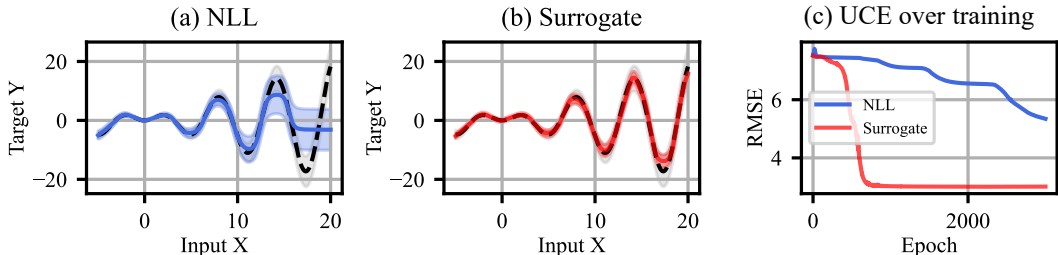

Figure 5: Fits for the heteroscedastic sine example from (Seitzer et al., 2022). A sine curve with increasing amplitude and noise: $y(x) = x\sin(x) + x\xi_1 + \xi_2$ where $\xi_1$ and $\xi_2$ are Gaussian noise with standard deviation $\sigma = 0.3$. Uncertainty Calibration Error (UCE) is used as an evaluation metric.

As a result, compared to the NLL loss, the surrogate loss avoids sudden curvature changes and numerical instabilities caused by high-order variance terms (e.g. $1/\sigma_j^4$, $1/\sigma_j^6$), and yields an optimization landscape with a lower condition number. This can be summarized by the inequality:

$$\kappa(\nabla^2 \mathcal{L}_{\text{surrogate}}) < \kappa(\nabla^2 \mathcal{L}_{\text{NLL}}), \tag{29}$$

where $\kappa(\cdot)$ denotes the condition number of the corresponding Hessian.

We now empirically compare our surrogate loss and the NLL loss. We computed their Hessian matrices and analyzed their condition numbers, minimum eigenvalues ($\lambda_{\min}$), and eigenvalue distributions. We generated synthetic data ten times with $d = 10$ under the ER1 setting, and performed DAG learning under the heteroscedastic conditions described in Section 4.1 of the main paper. Figure 4 summarizes our results. (a) The surrogate loss exhibited a lower condition number compared to the NLL loss, which indicates that it has a more well-conditioned curvature landscape. (b) The minimum eigenvalue of the NLL loss was significantly skewed towards negative values, which means the presence of saddle points in the loss surface and potentially unstable optimization behavior. Conversely, our surrogate loss maintained a minimum eigenvalue $\lambda_{\min}$ close to zero, indicating a generally flatter loss surface. (c) The eigenvalue distribution of the NLL loss exhibited a long tail, whereas the surrogate loss showed eigenvalues concentrated near zero. This implies that the surrogate loss landscape is flatter, which can lead to convergence to flatter minima and potentially better generalization performance. In conclusion, this empirical analysis supports the theoretical perspective, demonstrating that the surrogate loss indeed possesses a lower condition number compared to the NLL loss. This indicates that the surrogate loss is a more well-conditioned function with the potential for more stable convergence behavior.

**Toy example: Heteroscedastic regression.** Stirn et al. (2023) demonstrated the potential for stable convergence of nonlinear models by using the stop-gradient operation to decouple the updates of the mean and variance networks so that they proceed independently. Similarly, our proposed $\mathcal{L}_{\text{surrogate}}$ decouples the mean and variance contributions of the NLL loss, minimizing their interference with

each other and enabling a balanced optimization. In addition, the scheduling coefficient $\lambda_{\text{reg}}(t)$ assigns a large weight in the early stage of training to induce fast convergence, and gradually allows the objective function to transition smoothly to the NLL loss; this operates on a principle similar to the *temperature annealing* proposed in Upadhyay et al. (2023).

To verify that optimization under our $\mathcal{L}_{\text{surrogate}}$ is more stable than under the NLL loss, we conducted an experiment following Seitzer et al. (2022). In this experiment, we consider a heteroscedastic regression problem of simultaneously predicting the mean and variance for data generated from a sine function with heteroscedastic noise. We trained a two-layer MLP with sigmoid activations on this data. Figure 5 visualizes the results of this experiment. It shows that when using $\mathcal{L}_{\text{surrogate}}$, the predictive mean aligns with the true curve of the input data more closely than when using the NLL. For a quantitative evaluation metric we used the uncertainty calibration error (UCE), which measures how accurately the predictive variance explains the residuals. The results showed that the proposed method achieved a lower UCE compared to NLL, indicating that it effectively estimates the uncertainty (variance) of data while also fitting the mean well. These findings empirically support that $\mathcal{L}_{\text{surrogate}}$ can converge more stably than the NLL.

## D    DETAILS OF GRADUATED OPTIMIZATION

Our proposed objective transition approach based on the weighted loss scheduling guides the model parameters to converge stably in the early stage of training. Until the predictive mean fits the data well, it suppresses the influence of the complex NLL loss and ensures that $W$ is updated in a direction that explains the data effectively. This in turn effectively mitigates the phenomenon in which an incorrect mean estimate distorts both the predictive variance and the DAG learning process.

Conceptually, our optimization strategy is closely related to homotopy optimization and graduated non-convexity (GNC) techniques. Homotopy optimization (Gargiani et al., 2020; Deng et al., 2023) sets a simple, easy-to-optimize surrogate objective at the beginning and then gradually transitions to the original complex objective function to find the optimal solution. GNC is likewise an approach that, rather than directly optimizing a nonconvex objective function, gradually transforms a convex or otherwise smooth surrogate loss into a progressively less flat form during training (Hazan et al., 2016; Le & Zach, 2020). These methods share the following characteristics: (i) they do not directly minimize a nonconvex function with many local minima; (ii) they begin with a smooth and well-conditioned loss function; and (iii) by gradually transitioning the objective function to the nonconvex form, they maintain the prior optimization trajectory. Our proposed weighted loss scheduling optimization follows the same principles. Initially, training is performed using loss functions that are smoother than the NLL (such as $\mathcal{L}_{\text{MSE}}$ and $\mathcal{L}_{\text{VarReg}}$), and subsequently the objective function is gradually transitioned to the NLL loss and the acyclicity constraint. This process alleviates instabilities caused by abrupt changes in the gradient surface. Therefore, this work can be regarded as the first example of applying the classical concepts of GNC and homotopy to DAG learning under HNMs.

We state the assumptions used to derive conditions on the hyperparameters.

**Assumption 2** (Initial Constraint Suppression). *The penalty coefficient $\rho$ for the ALM is increased gradually, remaining small during the initial training phase ($t \leq T_1$). This approach prevents the acyclicity constraint from prematurely dominating the learning process, allowing the data reconstruction signal to guide the initial updates of the DAG structure.*

**Assumption 3** (Variance Alignment). *During the initial phase where the scheduling coefficient $\lambda_{reg}(t)$ is large, the surrogate loss terms, particularly $L_{VarReg}$ and $L_{StopNLL}$, are designed to align the predicted variance $\sigma^2$ with the magnitude of the empirical residuals. This mechanism calibrates the variance predictor and suppresses its tendency to overestimate, which is a common issue in joint optimization with NLL loss.*

**Assumption 4** (Slow Transition). *The scheduling coefficient $\lambda_{reg}(t)$ must decay slowly. A gradual transition from the surrogate objective to the standard NLL is critical for maintaining the continuity of the homotopy path, ensuring that the optimization trajectory does not experience abrupt shifts that could lead to suboptimal solutions.*

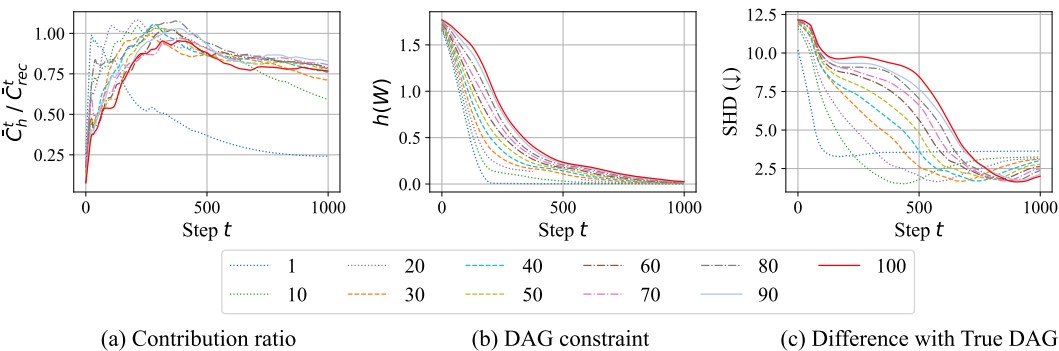

(a) Contribution ratio        (b) DAG constraint        (c) Difference with True DAG

Figure 6: Analysis on initial training dynamics using small $\lambda_{\mathrm{reg}}(0)$ values (within $\{1, 10, ..., 100\}$).

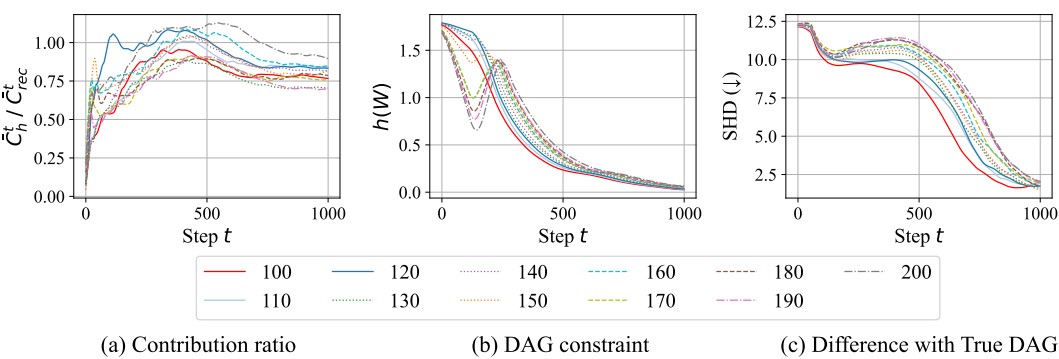

(a) Contribution ratio        (b) DAG constraint        (c) Difference with True DAG

Figure 7: Analysis on initial training dynamics using large $\lambda_{\mathrm{reg}}(0)$ values (within $\{100, 110, ..., 200\}$).

### D.1 ANALYSIS OF CONDITIONS FOR SCHEDULE PARAMETERS

As suggested in (Deng et al., 2023), homotopy optimization requires three conditions to maintain a continuous stationary trajectory toward the global optimum; we especially focused on the second condition, which stipulates that the coefficient $\lambda_{\mathrm{reg}}(t)$ must change only very slowly. The remaining conditions (the stationary point must be isolated and the Jacobian around the solution must be invertible) depend largely on the model architecture and the optimizer's characteristics, so we do not consider those two here. In relation to this second condition, the adjustable factors in our method are the schedule parameters $\lambda_{\mathrm{reg}}(0)$ and $t^*$. $\lambda_{\mathrm{reg}}(0)$ determines the relative weighting between the reconstruction loss and the acyclicity constraint $h(W)$ in the initial objective function. If $\lambda_{\mathrm{reg}}(0)$ is small, both terms are incorporated early on in a balanced manner, whereas if it is large, the reconstruction loss dominates the initial optimization. $t^*$ represents the transition point (but it is also related to the decay rate $\tau$. Section I.6 shows the experiments on $\tau$). If $t^*$ is very small, the penalty term increases abruptly and the loss landscape changes quickly, potentially violating the condition that the coefficient $\lambda_{\mathrm{reg}}(t)$ should decay slowly. Conversely, if $t^*$ is excessively large, the acyclicity constraint may take effect too late, causing the optimization to remain in a suboptimal regime.

We conducted DAG learning experiments under various settings to examine how changes in the schedule parameters affect training dynamics and performance. The experimental setup and procedure were identical to those described in Section 2.2 and Figure 2 of the main paper. Specifically, we generated 10 different synthetic datasets with $d = 10$ under the ER1 condition, and for each we evaluated the contribution ratio of the two loss functions, the value of $h(W)$, and the SHD (Structural Hamming Distance). Figures 6 and 7 show the results depending on varying $\lambda_{\mathrm{reg}}(0)$. When $\lambda_{\mathrm{reg}}(0)$ is small, the contribution ratio quickly approached 1 from the very beginning of training, and $h(W)$ also decreased rapidly; SHD initially decreased and then tended to stagnate at a somewhat high value. In contrast, as $\lambda_{\mathrm{reg}}(0)$ becomes larger (e.g., more than 150), $h(W)$ decreases slowly at first and then exhibits abnormal behavior such as rising again or oscillating, and in some cases the contribution ratio exceeded 1. The SHD values also showed an unstable pattern of decreasing, then increasing, then decreasing again. Overall, the range $100 \le \lambda_{\mathrm{reg}}(0) \le 130$ results in relatively stable training dynamics.

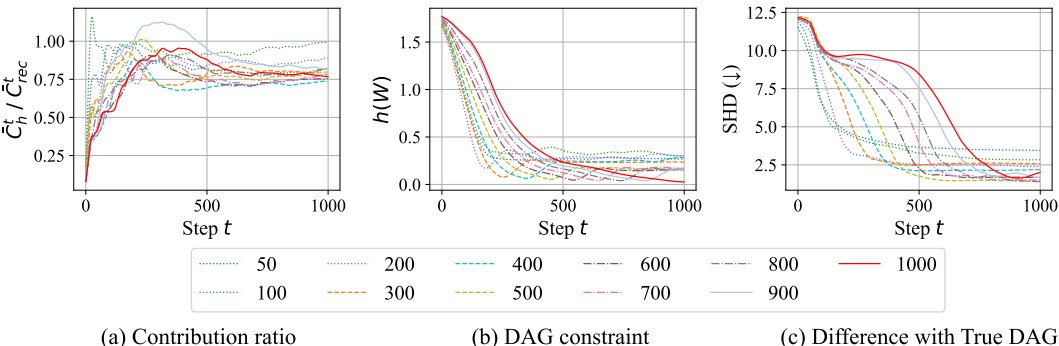

(a) Contribution ratio      (b) DAG constraint      (c) Difference with True DAG

Figure 8: Analysis on initial training dynamics using small $t^*$ values (within $\{50, 100, 200, ..., 1000\}$).

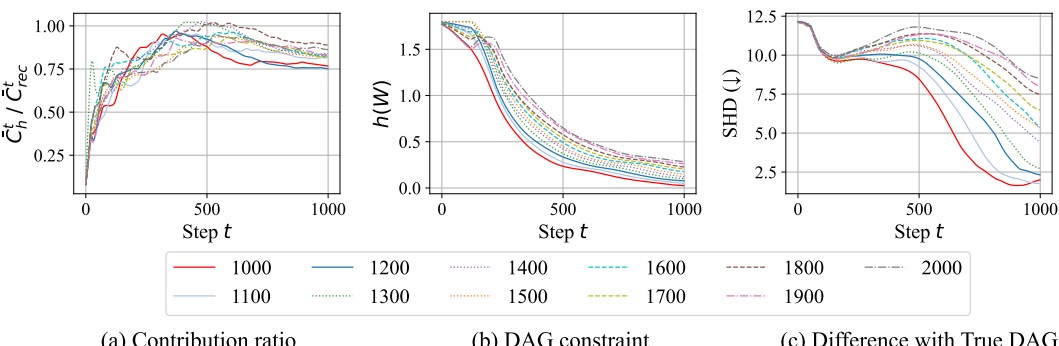

(a) Contribution ratio      (b) DAG constraint      (c) Difference with True DAG

Figure 9: Analysis on initial training dynamics using large $t^*$ values (within $\{1000, 1100, ..., 2000\}$).

Figures 8 and 9 present the results depending on different $t^*$ settings. When $t^*$ is very small (e.g., 50 to 300), the contribution ratio fluctuated irregularly and $h(W)$ dropped quickly at first but then its value oscillated. On the other hand, for values $t^* \geq 1000$, the contribution ratio remained stable and $h(W)$ decreased in a smooth, natural manner. Notably, beyond about $t^* \geq 1300$, we observed in some cases that early in training the contribution ratio would spike or $h(W)$ would actually increase. In conclusion, the range $1000 \leq t^* \leq 1200$ appears to be the most effective for DAG learning, as the SHD was gradually reduced within this range.

These results suggest that by appropriately tuning the schedule parameters, one can empirically satisfy the condition from Deng et al. (2023) that the coefficient changes only slowly. In other words, even in a complex nonlinear DAG learning setting, choosing suitable schedule parameters can maintain stable optimization, demonstrating that the theoretical condition does have a certain degree of effectiveness in practical optimization.

## D.2 TRAJECTORY ANALYSIS

We summarize the theoretical context of the proposed methodology through visualizations. The proposed method shares a similar objective with the spirit of homotopy optimization (Allgower & Georg, 2003; Gargiani et al., 2020) and the graduated non-convexity (GNC) method (Hazan et al., 2016; Ichikawa, 2024). Both methods share the characteristic that, in early training stages, they use an objective function that is easy to optimize and then gradually transform it continuously into the original nonlinear, nonconvex problem. The homotopy method proceeds along an "easy-to-solve" path, and it is classically known that following this path yields a stationary path (Allgower & Georg, 2003). GNC similarly operates on the same principle: it starts with smoothing a nonconvex function using a smoothing parameter, and then gradually reduces the level of flatness, allowing the optimizer to avoid poor local minima and gradually converge to better solutions (Hazan et al., 2016). Our method can be connected to homotopy's continuous path-tracking effect by gradually adjusting the relative influence between loss terms via $\lambda_{\text{reg}}(t)$ and by using a surrogate loss for the NLL in the early stage.

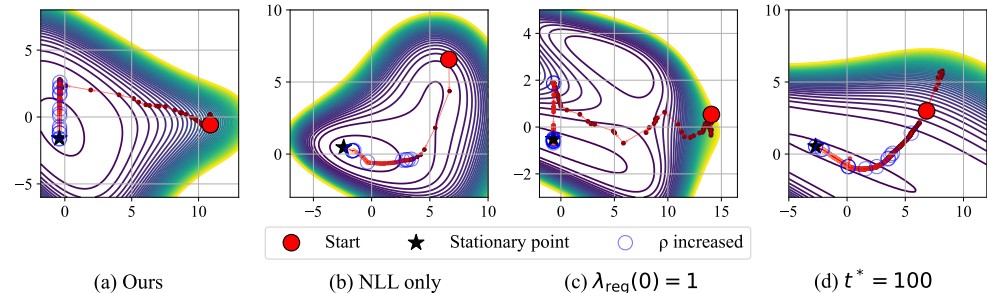

(a) Ours  (b) NLL only  (c) $\lambda_{\text{reg}}(0) = 1$  (d) $t^* = 100$

Figure 10: Trajectory analysis of the optimization. Contours closer to concentric circles indicate a more stable loss landscape, and trajectories closer to straight lines imply more stable convergence.

In this connection, Deng et al. (2023) theoretically presented conditions under which, for a linear bivariate DAG learning problem, a stationary path can be structurally maintained if the penalty is increased gradually. This explains that if the stationary point is isolated, the Jacobian around the solution is invertible, and the penalty is increased sufficiently slowly, then the entire optimization path can continue without breaks. However, such conditions generally do not hold in nonlinear multivariate structure learning involving MLPs, and in particular, due to the nonconvex nature of the loss landscape, even if a stationary path exists, there is no guarantee that it will be continuous. Nonetheless, we empirically observed the influence of the proposed optimization strategy on the actual training trajectory.

Figure 10 compares the loss contours and optimization trajectories visualized via PCA under different conditions for the ER1 graph with $d = 10$ (see Appendix E.1 for its detailed settings). The proposed method's trajectory is very smooth overall, and the loss contours near the convergence point form nearly concentric circles. Compared to the proposed method (a), all alternative settings (b–d) exhibit more unstable optimization behaviors. In particular, the highly reduced surrogate influence (c) produces a zigzag trajectory, while a rapid transition to NLL (d) results in a distorted loss landscape with sharp or elongated contours. Meanwhile, abrupt directional changes in the trajectory are observed across all settings (a–d), which aligns with prior findings that such instability can naturally occur in ALM-based L-BFGS optimization (Deng et al., 2023; Ng et al., 2022). Importantly, in our method, even after this change in direction, the trajectory continues smoothly. This suggests that the loss scheduling coefficient was designed to be neither too fast nor too slow, allowing the optimization at each stage to stay well within the attraction basin of the next stationary path. Even though our experimental setting involves a multivariate structure and nonlinear models that do not strictly satisfy the mathematical assumptions of Deng et al. (2023), the fact that the overall path continues without interruption and that increasing the penalty does not cause discontinuous changes in the trajectory provides empirical support for the theory.

To intuitively compare the influence of the schedule parameters on the optimization trajectory, we projected each trajectory obtained from different parameter values into the same two-dimensional space. Figure 11 shows the visualization, where both trajectories in each of (a) and (b) were projected onto the same PCA space, aligning their starting points to the same location.

Figure 11(a) shows the differences in trajectory when varying $\lambda_{\text{reg}}(0)$. When $\lambda_{\text{reg}}(0)$ is small (shown in green), the effect of the penalty term is manifested early, causing $\rho$ to increase rapidly and inducing premature convergence. In this case, the trajectory explores only a narrow region of the loss landscape, which may increase the risk of getting trapped in local minima or saddle points. In contrast, when $\lambda_{\text{reg}}(0)$ is large (red), the reconstruction loss dominates the initial optimization and $\rho$ increases only later. As a result, the trajectory covers a substantially broader region of the loss landscape, and convergence begins after sufficient exploration once $\rho$ eventually starts to rise. This tendency allows for a better understanding of the geometry of the loss landscape and offers a high potential for convergence to a better stationary point.

Figure 11(b) illustrates how the trajectory changes with different values of $t^*$. When it is large (red), the influence of the penalty term grows gradually, leading the optimization to proceed stably; a clear convergence trend appears when $\rho$ eventually increases at the final stage. Although the trajectory exhibits some wobbling initially, the optimization followed a consistent desired direction until the

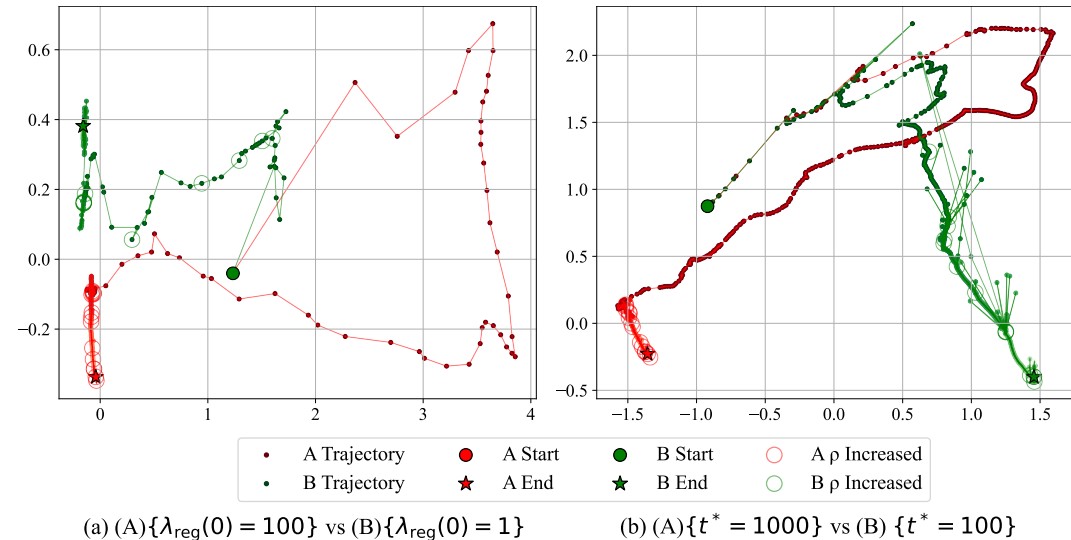

(a) (A)$\{\lambda_{\mathrm{reg}}(0) = 100\}$ vs (B)$\{\lambda_{\mathrm{reg}}(0) = 1\}$      (b) (A)$\{t^* = 1000\}$ vs (B) $\{t^* = 100\}$

Figure 11: Visualization of optimization trajectories projected onto the same two-dimensional space. For fair comparison, we did not use all training steps, but instead adjusted the sampling interval of trajectory points to match the number of steps in the shorter trajectory of the two methods. Through this analysis, we aim to assess how wide the space exploration is.

penalty was strengthened. In contrast, when $t^*$ is small (green), the penalty's influence comes into play early, causing the loss landscape to shift abruptly; consequently, the trajectory made discontinuous jumps or sharp turns and showed very unstable behavior.

As a result, we can see how the schedule parameters affect the dynamics and stability of the optimization trajectory; changes in $\lambda_{\mathrm{reg}}(0)$ and $t^*$ led to clear differences in the traversed region of the loss landscape, the point of convergence, and the consistency of the update direction. This suggests that when and how the penalty term dominates plays a critical role in maintaining a stable optimization trajectory. A gradual decrease of the scheduling coefficient directly impacts the continuity of the optimization trajectory and the accuracy of the learned DAG.

## E  DETAILS FOR SECTION 3.2

In this section, we define the key notation required to substantiate the theoretical analysis developed in Section 3.2, in particular to prove how the weighted loss scheduling strategy regulates the balance between the reconstruction loss and the DAG constraint to prevent training failures.

### E.1  NOTATIONS

First, we define the gradients representing the two key factors we aim to analyze. The update of $W$ is determined by the interaction between the reconstruction loss and the DAG constraint. The reconstruction gradient $g_{\mathrm{rec}}(t) = \nabla_W L_{\mathrm{rec}}(t)$ is the loss signal indicating how well the current model explains the data. The DAG constraint gradient $g_h(t) = \nabla_W\left(\frac{\rho}{2} h(W_t)^2 + \alpha\, h(W_t)\right)$ is the loss signal indicating how well the weight matrix $W_t$ satisfies the DAG structure.

In the optimization process (using an L-BFGS optimizer), the change $\Delta W_t$ in the parameter $W$ at each step $t$ can be approximated as:

$$\Delta W_t = -\eta_t H_t(g_{rec}(t) + g_h(t)), \tag{30}$$

where $\eta_t$ is the step size and $H_t \succeq 0$ is the L-BFGS inverse-Hessian approximation (a positive semi-definite matrix). This equation shows that the update of $W$ is a linear combination of the two gradients. As discussed in Section 3.2, it is important to quantitatively determine which gradient dominates the update of $W$ in the early stage of training. The per-step contribution ratio $\mathcal{Q}_t$ indicates

how much more strongly $g_h(t)$ influences the update of $W$ compared to $g_{\mathrm{rec}}(t)$ in a single step.

$$
\mathcal{Q}_t \;:=\; \frac{\left|\langle H_t\big(g_{\mathrm{rec}}(t) + g_h(t)\big),\, g_h(t)\rangle\right|}{\left|\langle H_t\big(g_{\mathrm{rec}}(t) + g_h(t)\big),\, g_{\mathrm{rec}}(t)\rangle\right|}. \tag{31}
$$

The cumulative contribution ratio is defined as the average of the per-step contribution ratios up to time $T_0$ (this is the metric used in Definition 3).

$$
\frac{C_h(T_0)}{C_{\mathrm{rec}}(T_0)} \;=\; \frac{1}{T_0} \sum_{t=0}^{T_0-1} \mathcal{Q}_t. \tag{32}
$$

If this ratio is greater than 1, it means that over that period the constraint term dominated the learning more than the reconstruction loss did. Finally, let $\kappa$ denote the condition number of the Hessian $\nabla_W^2 \mathcal{L}_t$ of the loss function, which indicates the stability of the optimization problem. A large value of $\kappa$ implies that the loss surface is unstable and optimization can become difficult.

$$
\kappa \;:=\; \mathrm{cond}\big(\nabla_W^2 \mathcal{L}_t\big) \;=\; \frac{\lambda_{\max}(\nabla_W^2 \mathcal{L}_t)}{\lambda_{\min}(\nabla_W^2 \mathcal{L}_t)} \in [1, \infty). \tag{33}
$$

### E.2 FAILURE CERTIFICATE

When optimizing the standard objective (Eq. (3)) under an HNM, the training can sometimes converge prematurely in the wrong direction. The failure certificate is a concept that formally defines and diagnoses this optimization failure mechanism. It quantitatively captures the phenomenon where, under certain conditions, the learned DAG structure is determined too hastily by the acyclicity constraint rather than by the true data relationships. A failure certificate is defined as the scenario in the early training interval $[0, T_0]$ where the following two conditions hold simultaneously: constraint-dominant and premature acyclicity. Here, constraint-dominant means that the cumulative contribution of the constraint relative to the reconstruction loss stays above some constant $\gamma > 1$ (indicating that the optimization is overly focused on enforcing acyclicity — essentially, the model is forming an acyclic graph without sufficiently learning the data's patterns). Premature acyclicity means that by the early time $T_0$, the value of $h(W)$ has converged below an extremely small threshold $\epsilon$, indicating a premature lock-in phenomenon wherein the graph structure has been finalized before the model has adequately learned from the data.

$$
\frac{C_h(t)}{C_{\mathrm{rec}}(t)} \;\geq\; \gamma \quad (0 \leq t \leq T_0), \qquad h(W_{T_0}) \;\leq\; \varepsilon. \tag{34}
$$

This failure arises from the inherent instability of the NLL loss under an HNM. Early in training, the variance tends to be overestimated, and because the reconstruction component of the NLL loss is inversely proportional to this variance, its influence is greatly attenuated. The DAG constraint, which grows increasingly stronger due to the ALM, can then easily overwhelm the reconstruction loss, thereby satisfying the conditions of the failure certificate described above. Moreover, this phenomenon occurred even with $\beta-$NLL (which was designed to mitigate the NLL loss's instability), and we regard this issue as stemming from the model architecture under HNM — that is, a network that must output both a mean and a variance. This demonstration of failure serves as a key motivation for our proposed progressive optimization strategy based on weighted loss scheduling. Our aim is to deliberately strengthen the influence of the reconstruction loss in the early stage of training so as to prevent these failure conditions from occurring, and to gradually induce the formation of the DAG structure only after the model has sufficiently learned from the data.

### E.3 AUXILIARY INEQUALITIES

In the next section, we present two key inequalities needed to complete the proof of Lemma 3.1. These auxiliary results yield an upper bound on the per-step contribution ratio $\mathcal{Q}_t$ and analyze conditions under which this bound can be controlled.

**Lemma E.1** (Quadratic–form bound for $\mathcal{Q}_t$). *First, to obtain a more tractable expression for $\mathcal{Q}_t$ at each step, we define an upper bound as a function of $u$, which encodes the relative magnitude of the reconstruction gradient $g_{\mathrm{rec}}$ and the constraint gradient $g_h$.*

*Let $a := \langle H_t g_{\text{rec}}, g_{\text{rec}} \rangle$, $b := \langle H_t g_h, g_h \rangle$, $c := \langle H_t g_{\text{rec}}, g_h \rangle$, and $u := \sqrt{b/a} \in [0, 1)$. Then*

$$\mathcal{Q}_t \leq \frac{b + \sqrt{ab}}{a - \sqrt{ab}} = u\frac{u+1}{1-u} =: \phi(u). \tag{35}$$

*Proof.* Triangle inequality and Cauchy–Schwarz in the $H_t$–inner product yield $|\langle H_t(g_{\text{rec}} + g_h), g_h \rangle| \leq b + \sqrt{ab}$ and $|\langle H_t(g_{\text{rec}} + g_h), g_{\text{rec}} \rangle| \geq a - \sqrt{ab}$, giving Eq. (35). $\square$

**Lemma E.2** (Monotonicity and threshold). *Next, we analyze properties of the upper-bound function $\phi(u)$ defined in Lemma 3.1. By exploiting its monotonicity, we identify a clear threshold $u_\delta$ such as $\mathcal{Q}_t$ remains below a target value $1 - \delta$. On $[0, 1)$, $\phi(u) = u(u + 1)/(1 - u)$ is strictly increasing with $\phi(0) = 0$. Let $u_\delta \in (0, 1)$ solve $\phi(u_\delta) = 1 - \delta$, i.e.,*

$$\frac{u_\delta(u_\delta + 1)}{1 - u_\delta} = 1 - \delta \quad \implies \quad u_\delta = \frac{-(2 - \delta) + \sqrt{(2 - \delta)^2 + 4(1 - \delta)}}{2}. \tag{36}$$

*Then $u \leq u_\delta \Rightarrow \mathcal{Q}_t \leq 1 - \delta$.*

*Proof.* Direct calculus and inversion of a quadratic. $\square$

Using these two auxiliary lemmas, we show that if the relative gradient size $u$ stays below the threshold $u_\delta$ the per-step contribution ratio $\mathcal{Q}_t$ can be controlled within the desired range—this plays a central role in the final proof presented next.

E.4 PROOF OF LEMMA 3.1

Fix $\delta \in (0, 1)$ and $T_1 \in \mathbb{N}$. If for all $t \leq T_0$,

$$\lambda_{\text{reg}}(t) \left\| \nabla_W(\mathcal{L}_{\text{MSE}} + \mathcal{L}_{\text{VarReg}}) \right\| \geq \frac{\kappa}{1 - \delta} \left\| \nabla_W(\tfrac{\rho}{2} h(W)^2 + \alpha h(W)) \right\|, \tag{37}$$

then $\mathcal{Q}_t \leq 1 - \delta$ for all $t \leq T_0$, hence $\frac{C_h(T_0)}{C_{\text{rec}}(T_0)} \leq 1 - \delta$.

*Proof.* By spectral bounds of $H_t$,

$$u = \sqrt{\frac{\langle H_t g_h, g_h \rangle}{\langle H_t g_{\text{rec}}, g_{\text{rec}} \rangle}} \leq \sqrt{\kappa}\frac{\|g_h\|}{\|g_{\text{rec}}\|}. \tag{38}$$

In the early phase with stop–gradient,

$$g_{\text{rec}}(t) = \lambda_{\text{reg}}(t) \nabla_W(\mathcal{L}_{\text{MSE}} + \mathcal{L}_{\text{VarReg}}), \qquad \nabla_W \mathcal{L}_{\text{StopNLL}}(t) = 0. \tag{39}$$

With stop–gradient Eq. (39), assumption Eq. (37) implies $\|g_{\text{rec}}\| \geq \frac{\kappa}{1-\delta}\|g_h\|$, hence

$$u \leq \frac{1 - \delta}{\sqrt{\kappa}} < 1. \tag{40}$$

Combining Lemma E.1 and E.2, $u \leq u_\delta$ yields $\mathcal{Q}_t \leq 1 - \delta$. A sharp sufficient condition is

$$\|g_{\text{rec}}\| \geq \frac{\sqrt{\kappa}}{u_\delta} \|g_h\|, \tag{41}$$

which is implied by Eq. (37) whenever $\kappa \geq \kappa_\delta := \left(\frac{1-\delta}{u_\delta}\right)^2$. Averaging over $t \leq T_0$ and using Eq. (32) gives the claim. $\square$

### E.5 PATH STABILITY UNDER SLOW SCHEDULING

This section provides the theoretical rationale for why, in the proposed weighted-loss scheduling, it is important to decrease the scheduling coefficient $\lambda_{\text{reg}}(t)$ slowly. This principle ensures that the optimization does not drift toward unstable regions but instead follows a stable homotopy path to a convergent solution.

To guarantee stable path tracking, we first assume that the scheduling coefficient $\lambda_{\text{reg}}(t)$ changes only slightly at each step (i.e., *decays slowly*):

$$\left| \lambda_{\text{reg}}(t+1) - \lambda_{\text{reg}}(t) \right| \leq L_\lambda / t^*, \tag{42}$$

We also assume that the gradient of the loss does not change abruptly ($(W, \lambda) \mapsto \nabla_W \mathcal{L}_{\text{sur}}(W; \lambda)$ is $L$–Lipschitz), and that around the solution path induced by the data the loss has a locally unique minimizer (local strong convexity (or PL–type) holds along the data-aligned stationary path with smallest Hessian eigenvalue $\mu > 0$). When these conditions are satisfied, we can apply the implicit function theorem, which guarantees that if the problem parameter ($\lambda_{\text{reg}}$) changes slightly, then the solution changes only slightly as well. Then by the implicit function theorem (Hazan et al., 2016; Allgower & Georg, 2003),

$$\left\| W^\star(\lambda') - W^\star(\lambda) \right\| \leq \frac{L}{\mu} |\lambda' - \lambda|, \tag{43}$$

In conclusion, this bound implies that if the scheduling coefficient $\lambda_{\text{reg}}(t)$ changes slowly enough, the optimal solution $W^\star$ at each step also moves only gradually. Therefore, our optimization algorithm can stably track this slowly moving target ($W^\star$). This prevents the optimization path from suddenly jumping in a different direction or reverting to the previously discussed constraint-dominant state, ultimately helping it converge to a better solution.

### E.6 MORE DETAILS OF CLAIM 1

This section provides a detailed proof of Claim 1. The core of Claim 1 is that the proposed scheduling strategy avoids the failure certificate into which standard NLL optimization falls, thereby learning a more accurate DAG structure with the same number of samples ($\xi_{\text{ours}} < \xi_{\text{NLL}}$).

**Assumptions**    We impose the following four standard assumptions for the proof.

(i) **Uniform convergence**: The empirical risk computed from a finite sample converges to the population risk when the sample size is sufficiently large.

$$\sup_W |r_{\rho,n}(W) - r_\rho(W)| \leq \Delta_n \tag{44}$$

(ii) **Local PL condition**: In a neighborhood of the data-induced optimal solution path ($\mathcal{N}$), the loss is locally strongly convex (PL-type), implying that the gradient norm is proportional to the distance to the optimum and thus ensuring stable convergence.

$$\frac{1}{2}\|\nabla_W r_{\rho,n}(W)\|^2 \geq \mu(r_{\rho,n}(W) - r_{\rho,n}(W_n^*)) \quad (\forall W \in \mathcal{N}, \mu > 0). \tag{45}$$

(iii) **Slow schedule**: As shown in the previous section, the scheduling coefficient varies sufficiently slowly so that the optimization path remains stable (homotopy tracking).

(iv) **Reconstruction-dominant phase**: Lemma 3.1 holds, and the early stage of training $[0, T_1]$ is dominated by the reconstruction loss.

*Proof.*

Step 1: Per-step dominance carried to the handoff We analyze the handoff time $t_1$ when the objective transitions smoothly from the surrogate loss to the NLL loss after the early training phase. By Assumption (iv), reconstruction remains dominant up to this time, so the magnitude of the reconstruction gradient $g_{\text{rec}}$ is much larger than that of the constraint gradient $g_h$:

$$\|g_{\text{rec}}(t_1)\| = \lambda_{\text{reg}}(t_1)\|\nabla_W(\mathcal{L}_{\text{MSE}} + \mathcal{L}_{\text{VarReg}})\| \geq \frac{\kappa}{1-\delta}\|g_h(t_1)\|. \tag{46}$$

Here $\kappa \geq 1$ denotes the condition number of the Hessian $\nabla_W^2 L_t$ (see Eq. (33)), and $\delta \in (0,1)$ is the safety margin introduced in Lemma 3.1.

**Step 2: Gradient decomposition and bound at $t_1$** Using the triangle inequality, we bound the full gradient $\nabla_W r_{\rho,n}(W_{t_1})$ by the sum of the component gradients. Combining this with Eq. (46), we derive an upper bound on the squared norm of the full gradient:

$$\|\nabla_W r_{\rho,n}(W_{t_1})\|^2 \leq C' \frac{\kappa \|g_h(t_1)\|^2}{\lambda_{\text{reg}}(t_1)\|\nabla_W(\mathcal{L}_{\text{MSE}} + \mathcal{L}_{\text{VarReg}})\|^2} \ , \tag{47}$$

for some constant $C' > 0$ that aggregates the smoothness constants of $r_{\rho,n}$ and the factor $(1-\delta)^{-2}$ introduced in Lemma 3.1. The key point is that the denominator's reconstruction signal is very large, so the overall gradient at the handoff is very small. This indicates that the optimization starts near a good solution.

**Step 3: Empirical gap via PL condition** Applying the PL condition in Assumption (ii), we show that the empirical error of the converged solution, $r_{\rho,n}(\hat{W}) - r_{\rho,n}(W_n^*)$, is proportional to the squared gradient norm. The small gradient magnitude from Step 2 directly yields a small empirical error:

$$r_{\rho,n}(\hat{W}) - r_{\rho,n}(W_n^*) \leq \frac{1}{2\mu}\|\nabla_W r_{\rho,n}(\hat{W})\|^2 \leq C_2 \frac{\kappa \|\nabla_W \mathcal{L}_{\text{DAG}}(\hat{W})\|^2}{\lambda_{\text{reg}}(t_1)\|\nabla_W(\mathcal{L}_{\text{MSE}} + \mathcal{L}_{\text{VarReg}})(\hat{W})\|^2}. \tag{48}$$

Here $C_2 > 0$ is a constant that depends only on $C'$ from Eq. (47) and on the PL parameter $\mu$ in Eq. (45) (in particular, one may take $C_2 = C'/(2\mu)$).

**Step 4: From empirical to population and the $\xi$-bound** Finally, using the uniform convergence in Assumption (i), we transfer the small empirical error to a small population error. In particular, combining Eq. (44) with the empirical bound in Eq. (48) yields

$$r_\rho(\hat{W}) - r_\rho(W_\rho^*) \leq C_1 \Delta_n + C_2 \frac{\kappa \|\nabla_W \mathcal{L}_{\text{DAG}}(\hat{W})\|^2}{\lambda_{\text{reg}}(t_1)\|\nabla_W(\mathcal{L}_{\text{MSE}} + \mathcal{L}_{\text{VarReg}})(\hat{W})\|^2} \ , \tag{49}$$

for some numerical constant $C_1 \geq 1$ derived from the standard empirical-to-population decomposition; in our setting we can take $C_1 = 2$. This proves the conclusion of Claim 1 that the error bound attainable by the proposed method ($\xi_{\text{ours}}$) is smaller than that of the standard NLL approach ($\xi_{\text{NLL}}$). $\square$

**Remarks** In conclusion, the proposed optimization strategy stabilizes and amplifies the reconstruction signal in the early phase of training, ensuring that optimization starts from a favorable initialization. Consequently, it avoids constraint-induced early lock-in and enables the learning of more accurate DAG structures. The purpose of Claim 1, however, is not to provide a fully general non-asymptotic guarantee that holds for arbitrary network architectures, optimization algorithms, or data distributions. Rather, it should be interpreted as an idealized, sample-dependent sufficient condition derived under the standard assumptions (i)–(iv) above—uniform convergence, a local PL-type condition along the data-induced solution path, a slow schedule for $\lambda_{\text{reg}}(t)$, and an early reconstruction-dominant phase. Within this setting, the bound in Eq. (16) is meant to formalize the qualitative mechanism suggested by our experiments: by keeping the optimization trajectory in a reconstruction-dominant regime and ensuring a smooth transition to the NLL loss, the scheduling scheme can reduce the attainable error $\xi_{\text{ours}}$ relative to $\xi_{\text{NLL}}$.

Regarding the assumptions themselves, Assumptions (iii) and (iv) are formalized using Assumption (i) and Lemma 3.1 (Eqs. (14)–(15)), and are further supported empirically by the analyses of the contribution ratio, the DAG constraint, and the optimization trajectory in Appendix D (Figures 1 and 6–10). These experiments show that our choice of schedule $\lambda_{\text{reg}}(t)$ indeed creates an early training phase in which the reconstruction loss dominates the DAG constraint, and that the transition from the surrogate loss to the NLL loss is sufficiently gradual in practice.

Assumptions (i) and (ii) follow the standard practice in the literature on homotopy and graduated non-convexity methods and PL-type convergence analyses (Hazan et al., 2016; Gargiani et al., 2020). We use them as idealized conditions that allow us to relate empirical and population risks and to derive a formal sample-dependent bound. Verifying global uniform convergence and PL-type conditions for general nonlinear neural networks is intractable in full generality; instead, we adopt relaxed assumptions in the spirit of the above works and complement Claim 1 with empirical evidence for the behavior of our scheduling and optimization trajectory.

# F    IDENTIFIABILITY OF MULTIVARIATE HNM

This section discusses the identifiability of HNM in the multivariate setting. As mentioned in Section 2 of the main paper, the identifiability condition of HNM in the multivariate setting follows previous studies (Duong & Nguyen, 2023; Yin et al., 2024). Therefore, the identifiability presented in this section is consistent with the identifiability theory established in prior work. We first analyze the identifiability of HNM in the bivariate setting and then extend it to the multivariate setting. HNM is identifiable if it satisfies the following conditions: (i) $\mu(\cdot)$ is a nonlinear function, (ii) $\sigma(\cdot)$ is a piecewise function, and (iii) the noise variables $E$ are independent and Gaussian.

Assume that the variables X and Y follow the model in Eq (1),

$$Y = \mu_Y(X) + \sigma_Y(X)E_Y. \tag{50}$$

If a backward model exists, i.e.

$$X = \mu_X(Y) + \sigma_X(Y)E_X, \tag{51}$$

where $E_X$ and $E_Y$ are the independent exogenous noise variables following standard Gaussian distributions, $\mu_j$ and $\sigma_j$ are twice-differentiable scalar functions with $\sigma_j(X), \sigma_j(Y) > 0$. For the forward and backward models to generate the same data distribution $p(X_1, X_2)$ simultaneously (i.e. for the model to be *unidentifiable*), one of the following scenarios must hold (Khemakhem et al., 2021; Immer et al., 2023):

1. $(\sigma_Y, \mu_Y) = (\frac{1}{Q}, \frac{P}{Q})$ and $(\sigma_X, \mu_X) = (\frac{1}{Q'}, \frac{P'}{Q'})$ where $Q, Q'$ are polynomials of degree two, $Q, Q' > 0$, $P, P'$ are polynomials of degree two or less, and $p(Y), p(X)$ are strictly log-mix-rational-log.

2. $\sigma_j(X), \sigma_j(Y)$ are constant, $\mu_Y, \mu_X$ are linear and $p(Y), p(X)$ are Gaussian densities.

In the HNM formulation we adopt in Eq. (1) of the main paper, $\mu_j$ is a nonlinear function, so the second scenario above (which requires linear $\mu$) does not apply. Likewise, $\sigma_j$ is piecewise (not of the form $\frac{1}{Q}$), so the first scenario does not apply. Therefore, there is no valid backward model for any distribution that satisfies Eq. (1) for bivariate cases; in other words, the model is identifiable.

Building on *Proposition 28* and *Lemmas 35* and *36* of Peters et al. (2014), we next examine identifiability in the multivariate setting. Assume that there exists another HNM with graph $\mathcal{G}'$ that $\mathcal{G} \neq \mathcal{G}'$. According to *Proposition 28* in Peters et al. (2014): let $\mathcal{G}$ and $\mathcal{G}'$ be two different DAGs over a set of variables $\boldsymbol{X}$. Assume $p(\boldsymbol{X})$ is generated by HNM in Eq. (1) and satisfies the Markov condition and causal minimality with respect to $\mathcal{G}$ and $\mathcal{G}'$. Then there are variables $L, Y \in \boldsymbol{X}$ such that for the set $\boldsymbol{Q} := \mathbf{PA}_Y^{\mathcal{G}} \setminus \{L\}$, $\boldsymbol{R} := \mathbf{PA}_L^{\mathcal{G}'} \setminus \{L\}$ and $\boldsymbol{S} := \boldsymbol{Q} \cup \boldsymbol{R}$, we have: (i) $L \to Y$ in $\mathcal{G}$ and $Y \to L$ in $\mathcal{G}'$. (ii) $\boldsymbol{S} \subseteq \mathbf{ND}_Y^{\mathcal{G}} \setminus \{L\}$ and $\boldsymbol{S} \subseteq \mathbf{ND}_L^{\mathcal{G}} \setminus \{Y\}$. The set $\mathbf{PA}_Y^{\mathcal{G}}$ consists of the parent variables of $Y$ in $\mathcal{G}$ and $\mathbf{ND}_Y^{\mathcal{G}}$ includes the non-descendants of $Y$ in $\mathcal{G}'$.

Let $\boldsymbol{S} = \boldsymbol{s}$ for some value $\boldsymbol{s}$ such that $p(\boldsymbol{s}) > 0$. Denote $L^* := L \mid \boldsymbol{S} = \boldsymbol{s}$ and $Y^* := Y \mid \boldsymbol{S} = \boldsymbol{s}$. According to Lemma 36 in (Peters et al., 2014), if the distribution $p(\boldsymbol{X})$ is generated by the following SEM:

$$X_j = f_j(X_{pa(j)}, E_j), \quad j = 1, 2, \ldots, d, X_{pa(j)} \in \boldsymbol{X}, \tag{52}$$

with corresponding DAG $\mathcal{G}$, then for any subset $\boldsymbol{K} \subseteq \mathbf{ND}_{X_j}^{\mathcal{G}}$, it holds that $E_{X_{pa(j)}} \perp\!\!\!\perp \boldsymbol{K}$. *Lemma 36* from Peters et al. (2014) is applicable to HNM because an HNM can be viewed as one specific class of the SEM in Eq. (52) (Yin et al., 2024), and based on this, Yin et al. (2024) showed the following holds: $E_Y \perp\!\!\!\perp (L, \boldsymbol{S})$ and $E_L \perp\!\!\!\perp (Y, \boldsymbol{S})$. We also follow these findings.

*Lemma 35* from Peters et al. (2014) indicates that if $E_Y \perp\!\!\!\perp (Y, \boldsymbol{Q}, \boldsymbol{R})$ then this condition holds for all $\boldsymbol{q}, \boldsymbol{r}$ with $p(\boldsymbol{q}, \boldsymbol{r}) > 0$ and $f(Y, \boldsymbol{Q}, E_Y) \mid_{\boldsymbol{Q}=\boldsymbol{q}, \boldsymbol{R}=\boldsymbol{r}} = f(Y \mid_{\boldsymbol{Q}=\boldsymbol{q}, \boldsymbol{R}=\boldsymbol{r}}, \boldsymbol{q}, E_Y)$. Building on this, Yin et al. (2024) derives the following expressions:

$$f(L, \boldsymbol{Q}, E_Y)|_{\boldsymbol{S}=\boldsymbol{s}} = f(L|_{\boldsymbol{S}=\boldsymbol{s}}, E_Y) = f(L^*, \boldsymbol{q}, E_Y), \tag{53}$$

$$f(Y, \boldsymbol{R}, E_L)|_{\boldsymbol{S}=\boldsymbol{s}} = f(Y|_{\boldsymbol{S}=\boldsymbol{r}}, E_L) = f(Y^*, \boldsymbol{r}, E_L). \tag{54}$$

From Eq. (1) of the main paper, we then obtain the following result:

$$Y^* = \mu_Y(\boldsymbol{q}, L^*) + \sigma_Y(\boldsymbol{q}, L^*)E_Y, \ E_Y \perp\!\!\!\perp L^* \text{ in } \mathcal{G}, \tag{55}$$

---

**Algorithm 1** GRADUATEDOPTIMIZATION: a weighted loss scheduling based L-BFGS optimization

---

**Require:** Input data $X$, parameters of the prediction network $\Theta$, DAG parameter $W$, initial scheduling coefficient $\lambda_{\text{reg}}(0)$, transition point $t^*$, decay rate $\tau$, current iteration $t$, Lagrange multiplier $\alpha$, penalty coefficient $\rho$

**Ensure:** Learned parameters $\hat{\Theta}, \hat{W}$

1: **repeat**
2:     **if** $t < t^*$ **then**                                   ▷ condition for transition
3:         $\lambda_{\text{reg}}(t) \leftarrow \lambda_{\text{reg}}(0) \cdot \exp\left(-\frac{t}{t^*/\tau}\right)$             ▷ scheduling (Eq. (8))
4:         $\mathcal{L}_{\text{surrogate}}(\Theta^{(t)}, W^{(t)}, t) \leftarrow \lambda_{\text{reg}}(t) \cdot \left((\mathcal{L}_{\text{MSE}}(\Theta_\mu^{(t)}, W^{(t)}) + \mathcal{L}_{\text{VarReg}}(\Theta^{(t)}, W^{(t)})\right) +$
    $\mathcal{L}_{\text{StopNLL}}(\Theta^{(t)}, W^{(t)})$                         ▷ surrogate loss (Eq. (7))
5:         $\mathcal{L}_{\text{total}} \leftarrow \mathcal{L}_{\text{surrogate}}(\Theta^{(t)}, W^{(t)}, t) + \alpha \cdot h(W^{(t)}) + \frac{\rho}{2}h(W^{(t)})^2$ ▷ early-phase objective
6:     **else**
7:         $\mathcal{L}_{\text{total}} \leftarrow \mathcal{L}_{\text{NLL}}(\Theta^{(t)}) + \alpha \cdot h(W^{(t)}) + \frac{\rho}{2}h(W^{(t)})^2$      ▷ original objective (Eq. (3))
8:     **end if**
9:     $\Theta^{(t+1)}, W^{(t+1)} \leftarrow$ L-BFGS($\mathcal{L}_{\text{total}}$)   ▷ update parameters via L-BFGS (with $W_{ij} \geq 0$ and $W_{ii} = 0$) (Zheng et al., 2020; Yin et al., 2024)
10:     $t \leftarrow t + 1$
11: **until** Convergence
12: $\hat{\Theta} \leftarrow \Theta^{(t)}$
13: $\hat{W} \leftarrow W^{(t)}$
14: **return** $\hat{\Theta}, \hat{W}$

---

$$L^* = \mu_L(\boldsymbol{r}, Y^*) + \sigma_L(\boldsymbol{r}, Y^*)E_L, \ E_L \perp\!\!\!\perp Y^* \text{ in } \mathcal{G}'. \tag{56}$$

Whereas no backward model satisfies the same assumptions as the forward model, Eqs. (55) and (56) holding simultaneously is inconsistent with this. Therefore, the assumption that there exists another HNM with $\mathcal{G} = \mathcal{G}'$ does not hold. As a result, the DAG $\mathcal{G}$ identifiable from $p(\boldsymbol{X})$ is unique.

**Remark.** Our method does not change the model class or the population objective itself. The surrogate loss together with its schedule serves as an auxiliary optimization mechanism introduced to improve the initial optimization trajectory: they guide the parameters toward a reconstruction-friendly regime and then, in the later stages of training, converge to the same Eq. (3) as in Yin et al. (2024). The identifiability discussion under idealized conditions is built precisely on this standard NLL-based objective, and our method uses Eq. (3) as is. Therefore, the proposed optimization strategy does not alter the identifiability of the HNM itself; at the same time, it converges to a better solution than Yin et al. (2024).

## G ALGORITHM

Algorithm 1 illustrates our proposed graduated optimization method.

Algorithm 2 describes the overall training procedure based on augmented Lagrangian method (ALM). At the end of each L-BFGS inner loop, the penalty coefficients $\rho$ and $\alpha$ are increased to enforce acyclicity on $W$, which is identical to the approach of Zheng et al. (2020). The training process terminates when $\rho$ reaches $\rho_{\max}$ or $h(W)$ falls below the tolerance $\epsilon$.

## H EXPERIMENT SETTINGS

For optimization, we used the NOTEARS (Zheng et al., 2018) continuous optimization framework as the backbone, and all baseline methods were run with their default settings as given in the original papers. We adopted L-BFGS (Byrd et al., 1995) for gradient-based training. All networks were 2-layer MLPs with 10 hidden units, and the learned weight matrix $W$ was binarized using the standard threshold of 0.3. In all experiments, we set $\tau = 5$, $\lambda_{\text{reg}}(0) = 100$, and $t^* = 1000$. We employed Structural Hamming distance (SHD), which measures edge differences between the estimated DAG and the ground-truth DAG, and F1-score of the edge detection, as our evaluation metrics.

---

**Algorithm 2** Augmented Lagrangian method-based DAG learning

---

**Require:** Initialized parameters for the prediction networks $\Theta^0 = \{\theta_\mu^0, \theta_\sigma^0\}$, Initialized DAG parameter $W^0$, input data $X$, max outer iterations $S$, initial scheduling coefficient $\lambda_{\text{reg}}(0)$, transition point $t^*$, decay rate $\tau$, tolerance $\epsilon$, maximum penalty coefficient $\rho_{\max}$

**Ensure:** Learned parameters $\Theta^*, W^*$

1: $h \leftarrow \infty, \alpha \leftarrow 0, \rho \leftarrow 1$        ▷ initialize Dual Ascent coefficients
2: **global** $t \leftarrow 0$        ▷ $t$ counts inner-loop iterations
3: **for** $s = 0$ to $S$ **do**        ▷ start outer loop (outer iteration index $s$)
4:      **while** $\rho < \rho_{\max}$ **do**        ▷ inner loop: check penalty increase condition
5:          $\Theta^{(t)}, W^{(t)} \leftarrow$ GRADUATEDOPTIMIZATION$(X, \Theta^{(t)}, W^{(t)}, \lambda_{\text{reg}}(0), t^*, \tau, t, \alpha, \rho)$    ▷
        inner loop optimization
6:          $h' \leftarrow h(W)$        ▷ compute DAG violations
7:          **if** $h' > 0.25 \cdot h$ **then**
8:             $\rho \leftarrow \min(10 \cdot \rho, \rho_{\max})$        ▷ increase penalty coefficient
9:          **else**
10:            **break**        ▷ stop inner loop
11:          **end if**
12:      **end while**
13:      $\alpha \leftarrow \alpha + \rho \cdot h'$        ▷ update Lagrange multiplier
14:      $h \leftarrow h'$
15:      **if** $h \leq \epsilon$ or $\rho \geq \rho_{\max}$ **then**
16:          **break**        ▷ stop
17:      **end if**
18: **end for**
19: $\Theta^* \leftarrow \Theta^{(t)}$
20: $W^* \leftarrow W^{(t)}$
21: **return** $\Theta^*, W^*$

---

We followed the settings of prior studies (Lachapelle et al., 2020; Yu et al., 2019; Zheng et al., 2020; Yin et al., 2024) in generating synthetic data. The ground-truth causal graph was sampled from an Erdős–Rényi (ER) model with parameter $k \in \{1, 2\}$, which controls the average number of edges. We considered numbers of variables $d \in \{10, 20, 50, 100\}$ and generated 10 random graphs for each $d$. For each graph, we then generated $N = 1000$ observations, repeating this process 10 times.

## H.1 SYNTHETIC DATASET DESCRIPTION

To evaluate the performance of the proposed method, we generated nonlinear synthetic data with two types of noise (homoscedastic and heteroscedastic settings). The data generation process follows the standard setup used in Zheng et al. (2020); Lachapelle et al. (2020); Yin et al. (2024). Ground-truth DAGs were generated based on Erdős–Rényi (ER) random graphs (ERDdS & R&wi, 1959). Specifically, we randomly sampled a topological order of $d$ variables and then added directed edges with probability $p = \frac{2k}{d^2-d}$, where $d$ is the number of variables and $k$ is the expected number of edges in the DAG. We used ER graphs with $k \in \{1, 2\}$, and for each setting of $d \in \{10, 20, 50, 100\}$, we generated 10 random graphs (using 10 different random seeds); each graph yielded $N = 1000$ observations.

**Synthetic data with homoscedastic noise.** Homoscedastic noise data is generated under the assumption that all variables have equal noise variance across all observations. In other words, each variable's noise has the same variance regardless of the observation. We generated such data by simulating a random DAG $G$ and then, following its topological order, by drawing observations from the following nonlinear SEM:

$$X_j = \mu_j(X_{pa(j)}) + E_j, \quad j = 1, \ldots, d, \tag{57}$$

where $\mu_j(\cdot)$ indicates a randomly initialized MLP with one hidden layer of size 100 and sigmoid activation. $E_j$ are standard Gaussian noises, i.e. $E_j \sim \mathcal{N}(0, 1)$.

**Synthetic data with heteroscedastic noise.** Heteroscedastic noise data is generated under the assumption that noise variances differ across variables and across observations. In particular, we assume each variable's noise variance depends on its parent variables (as in Eq. (1) of the main paper). Using randomly generated DAGs $G$, we followed the topological order of $G$ and drew observations from the following nonlinear SEM:

$$X_j = \mu_j(X_{pa(j)}) + \sigma_j(X_{pa(j)})E_j, \quad j = 1, \ldots, d, \tag{58}$$

where $\sigma_j(\cdot) = \sqrt{\log(1 + \exp(g_j(\cdot)))}$, and $\mu_j(\cdot)$ and $g_j(\cdot)$ are randomly initialized MLPs with one hidden layer of size 100 and sigmoid activation. $E_j$ are standard Gaussian noises, i.e. $E_j \sim \mathcal{N}(0, 1)$. Each variable's distribution in these simulations is Gaussian, and observations are sampled from these distributions. Because of this, inferring the exact structure from a single sample is challenging. Therefore, to perform effective inference when noise variances differ across the data, probabilistic modeling is essential, and using the NLL as the reconstruction loss is appropriate.

## H.2 REAL-WORLD DATASETS

To evaluate structure-learning performance in real settings, we use Sachs (Sachs et al., 2005), SynTReN (Van den Bulcke et al., 2006), and Causal Assembly (Göbler et al., 2024). The Sachs dataset consists of human cell protein expression data and includes 11 variables, 17 causal edges, and 853 observations. In this study we used only the observational data collected without intervention. The SynTReN dataset is a simulator-based pseudo-real dataset that generates synthetic transcriptional regulatory networks and gene expression data; for 20 nodes, we generated networks and data using 10 random seeds. The Causal Assembly Station dataset is based on sensor and actuator measurements collected from a real industrial assembly line; in this study we used the first two stations to construct a causal graph with 40 nodes and 86 directed edges, and similarly generated data with 10 random seeds.

## H.3 BASELINES

We compared our proposed method with several representative DAG learning methods from prior work to evaluate its performance. The baselines were selected to cover methods that assume either homoscedastic or heteroscedastic noise and that support the multivariate setting:

1. **NOTEARS-MLP** (Zheng et al., 2020) is a continuous optimization-based method that learns a DAG, assuming nonlinear SEMs with homoscedastic noise.

2. **GOLEM** (Ng et al., 2020) is a continuous optimization-based method for linear Gaussian and non-Gaussian settings. We use GOLEM-EV for the synthetic homoscedastic experiments.

3. **HOST** (Duong & Nguyen, 2023) is a combinatorial optimization-based method that assumes HNM and first identifies a causal ordering based on each variable's conditional normality, then recovers the DAG via conditional independence tests.

4. **ICDH** (Yin et al., 2024) is a continuous optimization-based method that assumes HNM and introduces a two-phase iterative learning algorithm that separates the training of the mean and variance functions to overcome optimization difficulties.

## H.4 EVALUATION METRICS

To assess the accuracy of DAG learning, we use the following metrics. Let the ground-truth DAG be $\mathcal{G}^* = (\mathcal{V}, \mathcal{E}^*)$ and the estimated DAG be $\hat{\mathcal{G}} = (\mathcal{V}, \hat{\mathcal{E}})$, where $\mathcal{V} = \{1, \ldots, d\}$ denotes the node set corresponding to the $d$ variables of $X = [X_1, \ldots, X_d]$. For a directed edge set $E$, define its skeleton (undirected edge set) by

$$\text{sk}(E) := \big\{\{i, j\} : (i \to j) \in E \text{ or } (j \to i) \in E\big\}.$$

Also define

$$\text{TP} = |\hat{\mathcal{E}} \cap \mathcal{E}^*|, \quad \text{FP} = |\hat{\mathcal{E}} \setminus \mathcal{E}^*|, \quad \text{FN} = |\mathcal{E}^* \setminus \hat{\mathcal{E}}|.$$

Here, TP (true positives) counts correctly predicted directed edges, FP (false positives) counts predicted directed edges not in the ground truth, and FN (false negatives) counts ground-truth directed edges missed by the predictor.

- **Structural Hamming Distance (SHD).** The number of edge additions, deletions, and reversals needed to transform $\hat{\mathcal{G}}$ into $\mathcal{G}^*$:

$$\mathrm{SHD}(\hat{\mathcal{G}}, \mathcal{G}^*) \;=\; \left| \mathrm{sk}(\hat{\mathcal{E}}) \triangle \mathrm{sk}(\mathcal{E}^*) \right| \;+\; R,$$

where the reversal count

$$R \;=\; \left| \left\{ \{i,j\} \in \mathrm{sk}(\hat{\mathcal{E}}) \cap \mathrm{sk}(\mathcal{E}^*) : (i \to j) \in \hat{\mathcal{E}},\ (j \to i) \in \mathcal{E}^* \text{ or vice versa} \right\} \right|.$$

- **False discovery rate (FDR).** Proportion of false positives among predicted positives:

$$\mathrm{FDR} \;=\; \frac{\mathrm{FP}}{\mathrm{TP} + \mathrm{FP}} \quad \text{(defined as 0 if TP + FP = 0)}.$$

- **True positive rate (TPR).** Proportion of true positives among actual positives:

$$\mathrm{TPR} \;=\; \frac{\mathrm{TP}}{\mathrm{TP} + \mathrm{FN}}.$$

- **F1-score.** Harmonic mean of precision and recall:

$$\text{Precision} = \frac{\mathrm{TP}}{\mathrm{TP} + \mathrm{FP}}, \quad \text{Recall} = \mathrm{TPR} = \frac{\mathrm{TP}}{\mathrm{TP} + \mathrm{FN}}, \qquad \mathrm{F1} \;=\; \frac{2\,\mathrm{TP}}{2\,\mathrm{TP} + \mathrm{FP} + \mathrm{FN}}.$$

### H.5 IMPLEMENTATION DETAILS

We implemented our DAG learning algorithm following the pseudocode given in Algorithms 1 and 2. Following prior work (Zheng et al., 2020; Yin et al., 2024), we set $\epsilon$, the tolerance for the acyclicity constraint in ALM, to $10^{-8}$; the maximum penalty coefficient to $\rho_{\max} = 10^{16}$. We increased $\rho$ by a factor of 10 at each update. We applied an $\ell_1$ regularization on $W$ (with coefficient 0.05) to encourage minimality of the learned graph, and an $\ell_2$ regularization (with coefficient 0.005) to reduce model complexity. Following Zheng et al. (2020), we extended the DAG parameter from $W \in \mathbb{R}^{d \times d}$ to $W \in \mathbb{R}^{d \times d \times m \times 2}$, where $m$ is the dimension of a hidden representation $Z$ (we set $m = 10$). This $W$ is split into two parameters $W^+ \in \mathbb{R}^{d \times d \times m}$ and $W^- \in \mathbb{R}^{d \times d \times m}$, and we compute $Z$ as:

$$Z = \mathrm{sigmoid}(XW^+ - XW^-). \tag{59}$$

The continuous graph structure represented by $W^+$ and $W^-$ is given by:

$$\text{matrix } \mathcal{G} = ||(W^+ - W^-)_{ij\cdot}||_2 = \left( \sum_{r=1}^{m} \left( W_{ijr}^+ - W_{ijr}^- \right)^2 \right)^{1/2}. \tag{60}$$

Finally, we obtain a discrete adjacency matrix by thresholding $G$; following Zheng et al. (2020), we fix this threshold to 0.3. All experiments were conducted on a computer equipped with an Intel i9-11900K CPU, 128GB RAM, and an NVIDIA GeForce RTX 3090 GPU, with the operating system of Ubuntu 20.04.

### H.6 MORE DETAILS FOR SECTION 2.2

This section provides the detailed settings for the experiments shown in Section 2.2 of the main paper. The results reported in Section 2.2 and Figure 1 of the main paper empirically demonstrate the phenomenon that can occur in DAG learning under an HNM assumption. We used an ER1 random graph setting with $d = 10$ and heteroscedastic noise, generating 10 different datasets (with 10 different random seeds). For a fair comparison, we fixed $\rho$ and $\alpha$ for the duration of 1000 training steps, since otherwise each method would start increasing $\rho$ and $\alpha$ at different times to enforce acyclicity, making it difficult to analyze the compared methods in the same environment. When computing SHD, motivated by Yin et al. (2024), we did not use a single fixed threshold of 0.3; instead, we averaged the SHD over 100 threshold values evenly spaced between 0.1 and 0.75, in order to capture edges with very weak connections during the early training phase. To observe the general trend of each metric over training, we applied Gaussian smoothing to remove sharp spikes in the metric values; we note that this smoothing does not affect the overall results but makes the trends easier to interpret.

Table 2: Comparison of SHD using denser graphs (ER3, ER4, ER5). Results are reported as mean ± standard deviation over 10 trials.

| Methods | SHD (↓) | | |
| | ER3 $d = 20$ | ER4 $d = 20$ | ER5 $d = 30$ |
| --- | --- | --- | --- |
| NOTEARS-MLP | 57.4 ± 7.405 | 65.0 ± 5.899 | 140.2 ± 17.910 |
| HOST | 58.4 ± 3.442 | 77.5 ± 7.406 | 195.6 ± 17.568 |
| ICDH | 59.6 ± 17.414 | 74.6 ± 14.705 | 185.5 ± 44.592 |
| **Ours** | **30.6 ± 6.735** | **39.5 ± 5.937** | **88.1 ± 14.356** |

Table 3: Comparison of F1-score using denser graphs (ER3, ER4, ER5). Results are reported as mean ± standard deviation over 10 trials.

| Methods | F1-score (↑) | | |
| | ER3 $d = 20$ | ER4 $d = 20$ | ER5 $d = 30$ |
| --- | --- | --- | --- |
| NOTEARS-MLP | 0.44 ± 0.074 | 0.45 ± 0.070 | 0.41 ± 0.046 |
| HOST | 0.40 ± 0.047 | 0.35 ± 0.068 | 0.27 ± 0.036 |
| ICDH | 0.58 ± 0.065 | 0.61 ± 0.062 | 0.55 ± 0.046 |
| **Ours** | **0.68 ± 0.101** | **0.68 ± 0.076** | **0.59 ± 0.093** |

For the experimental results reported in Figure 1 of the main paper, we also used an ER1 graph with $d = 10$ under heteroscedastic noise. This experiment visualizes the trajectory of $W$ and the loss landscape over training. Because $W \in \mathbb{R}^{d \times d}$, direct visualization was challenging; thus, we reduced the dimensionality of $W$ via PCA. We added small perturbations (noise) to $W$ in the reduced space, projected the perturbed parameters back to the original space, and evaluated the loss to plot landscape[1].

# I ADDITIONAL EXPERIMENTS

## I.1 PERFORMANCE ON DENSER GRAPHS

Following prior work (Duong & Nguyen, 2023; Yin et al., 2024), we have presented experiments on nonlinear synthetic data for ER1 and ER2, which correspond to relatively sparse graphs. However, in real-world scenarios the data's causal structure can be represented by denser graphs, so it is necessary to evaluate DAG learning performance in such cases as well. This section evaluates the performance of our method on denser graphs (ER graphs with $k \in \{3, 4, 5\}$) under heteroscedastic noise.

Tables 2 and 3 show the results for the SHD and F1-score metrics, respectively. Consistent with the experiments on the ER1 and ER2 data, the proposed method achieved the lowest SHD on the these denser graphs compared to the baselines. Notably, in the ER5 setting, the proposed method obtained an SHD of 88.1, which is much lower than the value of 140.2 achieved by NOTEARS-MLP (the next best baseline). In terms of F1-score, the proposed method also consistently achieved the highest scores compared to the baselines, demonstrating that even in the denser graph setting it accurately identifies the true edges while effectively excluding spurious edges.

## I.2 PERFORMANCE ON A LARGER GRAPH (UP TO 1,000)

In the main text, we performed experiments on graphs with up to 100 nodes. To test whether performance remains stable even when the graph size becomes extremely large, we now evaluate our approach on a graph with 1,000 nodes. The results are shown in Table 4, 5. We observe that ICDH suffers from an explosive increase in SHD when the number of variables is very large. In contrast, our proposed method still performs much better than ICDH. Specifically, ICDH adds too many edges in an attempt to reconstruct the data, leading to a high SHD and a high FDR (false discovery rate), which in turn yields a low F1-score. NOTEARS-MLP, since it does not suffer from variance-related

---

[1]Meanwhile, the resulting trajectory and loss landscape shown in the main paper would be difficult to interpret due to the dimensionality reduction. Therefore, in Figure 11, we projected the trajectories into the same two-dimensional space (obtained via PCA) to enable a direct comparison.

Table 4: Comparison of SHD and F1-score using a larger graph (ER2). Results are reported as mean ± standard deviation over 10 trials.

| Methods | SHD ($\downarrow$) | F1-score ($\uparrow$) |
|---|---|---|
| NOTEARS-MLP | 1592.3 ± 219.8 | 0.18 ± 0.127 |
| ICDH | 5158.3 ± 598.1 | 0.09 ± 0.066 |
| **Ours** | **1130.5 ± 153.5** | **0.54 ± 0.093** |

Table 5: Comparison of FDR, TPR, and the number of edges using a larger graph (ER2). Results are reported as mean ± standard deviation over 10 trials.

| Methods | FDR ($\downarrow$) | TPR ($\uparrow$) | # of edges |
|---|---|---|---|
| NOTEARS-MLP | 0.89 ± 0.085 | 0.43 ± 0.142 | 2150 ± 237.61 |
| ICDH | 0.95 ± 0.033 | **0.64 ± 0.128** | 6404 ± 672.31 |
| **Ours** | **0.42 ± 0.087** | 0.51 ± 0.109 | 1552 ± 75.66 |

errors, produces fewer spurious edges than ICDH, but it likewise fails to properly reconstruct the data, resulting in a low F1 score. By contrast, our method is able to successfully reconstruct the data by placing a high weight on the reconstruction loss in the initial stage, achieving the highest F1-score and the lowest SHD among all methods. Of course, as the scale becomes this extreme, performance drops for all methods (including ours). However, even at this scale, our method maintains a significantly higher F1-score compared to the others.

### I.3 ADDITIONAL RESULT ON REAL-WORLD DATA

Table 6: Results on the Cause-Effect Pairs dataset.

| Method | Accuracy $\uparrow$ | Weighted Accuracy $\uparrow$ |
|---|---|---|
| NOTEARS-MLP | 39/99 | 0.49 |
| NOTEARS | 36/99 | 0.47 |
| GOLEM-EV | 33/99 | 0.40 |
| GOLEM-NV | 33/99 | 0.40 |
| ICDH | 52/99 | 0.58 |
| Ours | **58/99** | **0.66** |

We evaluated the proposed method on another real-world dataset, Cause-Effect Pairs (Sgouritsa et al., 2015). This dataset consists of 99 bivariate datasets collected from different domains, and for each pair the task is to infer the causal direction. The evaluation criterion is whether the inferred direction between two variables is correct ($X \leftarrow Y$ or $X \rightarrow Y$), and we use accuracy as the metric. To correct for potential bias caused by multiple pairs originating from the same source dataset (Mooij et al., 2016), we also report a weighted accuracy. In computing the weighted accuracy, all pairs from the same source dataset are given equal weight such that the total weight for each dataset is 1.

Table 6 presents the experimental results. The proposed method correctly inferred 58 out of 99 causal directions. Most baselines inferred only 33 or 36 directions correctly; while the recently proposed ICDH method achieved 52, our method surpasses it. These results, together with those on the Sachs dataset, indicate that the proposed method achieves superior performance on real-world data compared to existing methods.

### I.4 FURTHER ANALYSIS OF OUR STRATEGIES

In this section, we further analyze the effect of the proposed surrogate loss and weighted loss scheduling. To this end, we compare the accuracy of learned DAGs when using NLL (Nix & Weigend, 1994), $\beta$-NLL (Seitzer et al., 2022), Faithful Heteroscedastic (Stirn et al., 2023), and using only the surrogate loss. Then, we apply the weighted loss scheduling to each loss function and

Table 7: Additional analysis on our surrogate loss and weighted loss scheduling. Comparison of SHD for all baseline algorithms on ER1 (mean ± standard deviation over 10 trials). Here, 'w †' indicates that weighted loss scheduling is applied.

| Graph | Methods | SHD (↓) | | |
|---|---|---|---|---|
| | | $d = 10$ | $d = 20$ | $d = 50$ |
| ER1 | NLL | 8.6 ± 2.498 | 19.9 ± 6.220 | 47.7 ± 12.884 |
| | $\beta$-NLL | 5.0 ± 1.844 | **10.2 ± 2.676** | **30.8 ± 5.510** |
| | Faithful | **4.4 ± 2.289** | 10.2 ± 4.686 | 98.5 ± 40.038 |
| | Ours w/o † | 4.5 ± 1.857 | 11.8 ± 4.556 | 32.5 ± 9.770 |
| | NLL w † | 8.4 ± 3.323 | 16.8 ± 4.059 | 38.3 ± 5.352 |
| | $\beta$-NLL w † | 4.0 ± 1.949 | 9.3 ± 3.743 | 28.0 ± 6.633 |
| | Faithful w † | 4.3 ± 2.610 | 11.7 ± 4.900 | 55.5 ± 20.422 |
| | **Ours** | **3.8 ± 2.272** | **7.4 ± 3.262** | **22.6 ± 4.609** |

compare performance. This experiment is intended to analyze how each component influences DAG learning performance and to evaluate the interaction between the surrogate loss and the weighted loss scheduling.

Table 7 shows the results in terms of SHD. First, comparing the loss functions without weighted loss scheduling, 'Ours w/o †' (surrogate loss without transition and coefficient) achieved an SHD on par with $\beta$-NLL. This suggests that using the surrogate loss alone produces a similar effect on DAG learning as the loss functions proposed in the heteroscedastic regression literature (Nix & Weigend, 1994; Seitzer et al., 2022; Stirn et al., 2023). Nevertheless, since its SHD is lower than that of NLL, such result implies that our surrogate loss is a more advantageous loss function for DAG learning than NLL. In other words, the surrogate loss can alleviate the difficulties arising from the joint optimization of NLL.

When applying the weighted loss scheduling strategy, the proposed method achieved the lowest SHD in all settings. For example, at $d = 50$, the proposed method attained an SHD of 22.6, whereas $\beta$-NLL obtained 28.0 and Faithful Heteroscedastic 55.5. Although all loss functions saw some performance improvement with weighted loss scheduling (denoted by 'w †'), applying this to the surrogate loss yielded an overwhelming performance gain. This suggests that the proposed surrogate loss is designed to effectively guide the optimization trajectory under the weighted loss scheduling strategy.

As a result, the surrogate loss is not simply aimed at stable likelihood optimization; beyond this, it is designed in anticipation of the training objective's gradual transition to the NLL with using weighted loss scheduling. This design is a distinguishing feature compared to the loss functions from the heteroscedastic regression literature, and ultimately it enables more accurate DAG learning.

## I.5 FURTHER ANALYSIS OF WEIGHTED LOSS SCHEDULING WITH THE CONSTANT VARIANCE MODEL

Table 8: Comparison of SHD for NOTEARS-MLP (which does not predict variance), with and without the proposed weighted loss scheduling, on nonlinear synthetic heteroscedastic-noise datasets (mean ± standard deviation over 10 trials). † denotes weighted loss scheduling.

| Graph | Methods | SHD (↓) | | | |
|---|---|---|---|---|---|
| | | $d = 10$ | $d = 20$ | $d = 50$ | $d = 100$ |
| ER1 | NOTEARS-MLP | 9.1 ± 4.323 | 19.1 ± 4.636 | 88.9 ± 23.851 | 275.6 ± 49.364 |
| | NOTEARS-MLP w † | 5.1 ± 2.835 | 12.5 ± 3.923 | 64.5 ± 16.832 | 112.0 ± 19.955 |
| | **Ours** | **3.8 ± 2.272** | **7.4 ± 3.262** | **22.6 ± 4.609** | **46.7 ± 7.988** |
| ER2 | NOTEARS-MLP | 12.8 ± 3.682 | 42.0 ± 8.136 | 150.8 ± 40.506 | 438.5 ± 77.740 |
| | NOTEARS-MLP w † | 10.0 ± 3.184 | 32.3 ± 7.095 | 103.7 ± 29.947 | 308.5 ± 66.395 |
| | **Ours** | **7.0 ± 2.236** | **17.5 ± 5.408** | **48.5 ± 16.585** | **132.38 ± 50.838** |

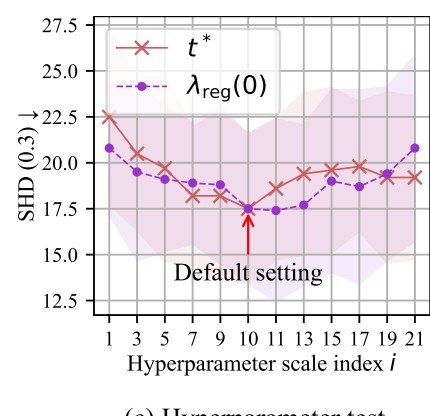

(c) Hyperparameter test

Figure 12: Results on the hyperparameter analysis.

The central motivation of this paper is the optimization instability that occurs when using the NLL as the reconstruction loss — an issue that does not arise when using MSE. We empirically verified this phenomenon through the experiment in Section 2.2 (Figure 1). In addition, we analyze the performance of applying our weighted loss scheduling to NOTEARS-MLP (which uses a constant variance). This entails multiplying the MSE term of NOTEARS-MLP by our weight factor that varies with each training step. We use the same hyperparameters for the scheduling, and the results are presented in Table 8. Applying the proposed weighted loss scheduling to NOTEARS-MLP consistently improved its performance. This indicates that training the model with a strongly emphasized reconstruction loss in the early stage can steer $W$ closer to the true DAG, demonstrating how crucial the influence of the reconstruction loss is. Nonetheless, our proposed approach (using a surrogate loss with weighted loss scheduling) achieved even higher performance. This suggests that in an HNM setting a variance function is indeed necessary, and that one must not only predict the variance but also ensure that the reconstruction loss has a strong influence in the initial phase.

## I.6 HYPERPARAMETER ANALYSIS

Here, we analyzed the influence of the scheduling parameters: $t^*$ and $\lambda_{\text{reg}}(0)$. Figure 12 plots SHD, where the $x$-axis corresponds to a scaling index $i$ such that $t^* = 100i$ and $\lambda_{\text{reg}}(0) = 10i$ so that 20 linearly spaced values for each hyperparameter are tested (each value is repeated 10 times). We observed that as $t^*$ increases, performance gradually improves, then beyond approximately $t^* \geq 1100$ we observed SHD tending to rise (low performance) again. This indicates that when $t^*$ is set above a certain level, the optimization process becomes smoother and converges more stably, which aligns with the design premise of our strategy. A similar trend was observed for $\lambda_{\text{reg}}(0)$. We interpret this as follows: when $\lambda_{\text{reg}}(0)$ is sufficiently large, the influence of the acyclicity constraint term $\mathcal{L}_{\text{DAG}}$ is relatively diminished and the surrogate loss dominates the optimization, leading to better structure learning performance. However, if both schedule parameters are set too high, performance starts to deteriorate again, indicating that simply increasing these values is not always beneficial. Our weighted loss scheduling is defined in Eq. (8) of the main paper, where we adopted an exponential decay schedule with a decay rate of $\tau = 5$. To validate this choice, we varied $\tau$ from 0.1 to 100 and also replaced the exponential schedule with a linear schedule to compare performance. The experiments were conducted on ER2 graph with $d = 20$, repeated 10 times. Figure 13 illustrates the change of $\lambda_{\text{reg}}(t)$ under each strategy, and Table 9 presents the corresponding performance.

The results show that using $\tau = 5$ yields the best performance (SHD = 17.5, F1-score = 0.76). In contrast, linear scheduling results in a noticeable performance drop (SHD = 23.6, F1-score = 0.69). The cases of $\tau = 0.1$ and $\tau = 1$ also exhibit significantly worse performance; in those cases, $\lambda_{\text{reg}}(t)$ decays more slowly than with linear scheduling and remains very high at the transition point ($t^* = 1000$). This implies that if the objective's gradual transition is too slow, or if $\lambda_{\text{reg}}(t)$ is not sufficiently low by the transition point, performance degrades. Meanwhile, for $\tau = 10$ and $\tau = 100$, $\lambda_{\text{reg}}(t)$ decays faster than for $\tau = 5$, and such faster decay leads to a larger performance drop. This suggests that $\lambda_{\text{reg}}(t)$ must decrease at an appropriate rate—neither too fast nor too slow. Consequently,

Table 9: Performance comparison under different decay strategies for weighted loss scheduling.

| Method | SHD ↓ | F1-score ↑ |
|---|---|---|
| linear | 23.6 ± 11.842 | 0.69 ± 0.135 |
| $\tau = 0.1$ | 26.6 ± 13.381 | 0.66 ± 0.093 |
| $\tau = 1$ | 25.4 ± 12.052 | 0.68 ± 0.109 |
| $\tau = 5$ | **17.5 ± 5.408** | **0.76 ± 0.087** |
| $\tau = 10$ | 21.5 ± 8.846 | 0.66 ± 0.181 |
| $\tau = 100$ | 28.7 ± 8.112 | 0.47 ± 0.266 |

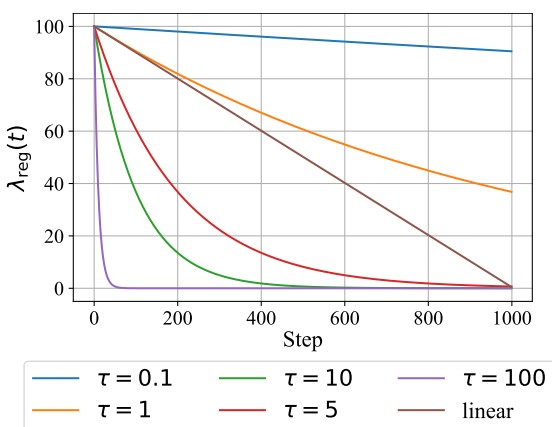

Figure 13: $\lambda_{\text{reg}}(t)$ trends with different scheduling.

we conclude that $\tau$ should be greater than 1 but less than 10, and setting $\tau = 5$ generally delivers the best performance.

### I.7 COMPARISON WITH HETEROGENEOUS DATA METHODS

We discussed the differences between methods for handling heterogeneous data and HNM-based methods in Section B.4. Despite differences in the underlying assumptions, here we aim to clarify the distinctions by directly comparing their performance. We do not treat these results as our primary observations, and we do not interpret performance differences with these baselines as superiority, since they correspond to a different problem setting.

For this comparison, we selected DARING (He et al., 2021) and ReScore (Zhang et al., 2023a) as representative methods designed for heterogeneous data. We conducted experiments on synthetic data corresponding to ER1 and ER2 with $d = \{10, 20, 50, 100\}$ under the HNM setting. Table 10 shows the results in terms of SHD, and Table 11 shows the results in terms of F1-score. DARING estimates a DAG by assuming independence among residuals; however, under an HNM this assumption can be violated because the residual(noise) variance depends on the input, causing DARING's performance to deteriorate as the number of variables increases. Additionally, ReScore assigns weights based solely on the sample loss, which under HNM amplifies the score estimation bias caused by input-dependent variance, leading to degraded performance.

For the ER1 case with $d = 10$, the HNM-based methods HOST and ICDH show better performance than DARING and ReScore. This performance difference can be interpreted as a consequence of the different assumptions used by the methods. On the other hand, as the graph becomes more complex, the performance of HOST and ICDH worsens. This is exactly the limitation of those methods as defined in our paper. Because of the need to estimate variance information in HNMs, the model's reconstruction capability is unstable in the early stage. We analyzed this problem and proposed a methodology to address it. As a result, our method is able to maintain the best performance even as the graph structure becomes more complex.

Table 10: Comparison of SHD with heterogeneous-data methods in the heteroscedastic setting (mean ± standard deviation over 10 trials).

| Graph | Methods | SHD ($\downarrow$) | | | |
| | | $d = 10$ | $d = 20$ | $d = 50$ | $d = 100$ |
|---|---|---|---|---|---|
| ER1 | DARING | 7.3 ± 3.164 | 14.5 ± 4.433 | 59.6 ± 15.351 | 134.6 ± 17.500 |
| | ReScore | 7.4 ± 3.929 | 25.6 ± 10.707 | 311.7 ± 58.905 | 1044.1 ± 165.83 |
| | HOST | 5.4 ± 3.105 | 11.4 ± 2.458 | 84.4 ± 26.740 | 143.8 ± 21.085 |
| | ICDH | 6.7 ± 4.473 | 26.0 ± 11.645 | 303.9 ± 121.223 | 518.8 ± 85.588 |
| | **Ours** | **3.8 ± 2.272** | **7.4 ± 3.262** | **22.6 ± 4.609** | **46.7 ± 7.988** |
| ER2 | DARING | 13.2 ± 3.970 | 37.0 ± 7.416 | 107.0 ± 17.748 | 271.8 ± 29.624 |
| | ReScore | 13.9 ± 3.330 | 57.1 ± 16.748 | 371.6 ± 104.348 | 1360.1 ± 298.263 |
| | HOST | 14.5 ± 3.138 | 38.8 ± 5.689 | 201.8 ± 63.871 | 495.7 ± 155.535 |
| | ICDH | 13.7 ± 4.001 | 50.0 ± 19.037 | 415.3 ± 185.616 | 664.0 ± 194.376 |
| | **Ours** | **7.0 ± 2.236** | **17.5 ± 5.408** | **48.5 ± 16.585** | **132.38 ± 50.838** |

Table 11: Comparison of F1-score with heterogeneous-data methods in the heteroscedastic setting (mean ± standard deviation over 10 trials).

| Graph | Methods | F1-score ($\uparrow$) | | | |
| | | $d = 10$ | $d = 20$ | $d = 50$ | $d = 100$ |
|---|---|---|---|---|---|
| ER1 | DARING | 0.58 ± 0.189 | 0.57 ± 0.145 | 0.46 ± 0.064 | 0.45 ± 0.057 |
| | ReScore | 0.66 ± 0.184 | 0.53 ± 0.088 | 0.20 ± 0.047 | 0.13 ± 0.023 |
| | HOST | 0.66 ± 0.193 | 0.67 ± 0.079 | 0.43 ± 0.084 | 0.46 ± 0.041 |
| | ICDH | 0.72 ± 0.171 | 0.54 ± 0.116 | 0.25 ± 0.133 | 0.11 ± 0.018 |
| | **Ours** | **0.79 ± 0.132** | **0.79 ± 0.111** | **0.74 ± 0.053** | **0.73 ± 0.052** |
| ER2 | DARING | 0.54 ± 0.157 | 0.49 ± 0.088 | 0.42 ± 0.064 | 0.37 ± 0.033 |
| | ReScore | 0.65 ± 0.085 | 0.51 ± 0.099 | 0.29 ± 0.075 | 0.18 ± 0.031 |
| | HOST | 0.49 ± 0.103 | 0.46 ± 0.072 | 0.30 ± 0.064 | 0.27 ± 0.061 |
| | ICDH | 0.65 ± 0.085 | 0.55 ± 0.111 | 0.31 ± 0.135 | 0.17 ± 0.056 |
| | **Ours** | **0.79 ± 0.071** | **0.76 ± 0.087** | **0.71 ± 0.100** | **0.62 ± 0.080** |

## I.8 COMPUTATIONAL COST

Table 12: Computational cost on the ER2 synthetic graph with $d = 20$ under the heteroscedastic noise setting. We report wall-clock training time (in seconds), the total number of optimization iterations (in units of $10^3$), and peak GPU memory usage (in MB) for each method.

| Method | Time (s) | Iterations ($\times 10^3$) | Memory (MB) |
|---|---|---|---|
| NOTEARS-MLP | 19.67 | 25.07 | 540.93 |
| ICDH | 389.85 | 131.88 | 542.87 |
| NLL | 1463.85 | 163.07 | 550.05 |
| Ours | 651.47 | 28.03 | 549.11 |

**Time Complexity.** From Zheng et al. (2020), NOTEARS-MLP requires $O\left(Nd^2m + d^2m + d^3\right)$ floating-point operations per L-BFGS iteration, where $N$ denotes the number of samples, $d$ denotes the number of variables, and $m$ denotes the hidden-layer width. When focusing on the dependence with respect to $d$, the total time complexity simplifies to $O(d^3)$. ICDH is implemented under the same augmented Lagrangian framework and uses the same MLP backbone, so the per-iteration cost of evaluating its objective and gradient has the same order $O\left(Nd^2m + d^2m + d^3\right)$. However, ICDH employs a two-phase alternating optimization strategy in which the mean and variance networks are

updated in separate phases. Let $S \geq 1$ denote the constant that summarizes the additional number of such two-phase alternations required to reach convergence compared to a single-phase update. Then the total complexity of ICDH can be expressed as $O(Sd^3)$.

Our method shares the same augmented Lagrangian backbone and 2-layer MLP architecture as NOTEARS-MLP and ICDH, so that per L-BFGS iteration incurs a cost of $O(Nd^2m + d^2m + d^3)$, and the additional surrogate loss terms involve only simple pointwise operations and a scalar update of $\lambda_{\text{reg}}(t)$, which do not change this asymptotic order. Unlike ICDH, however, our approach jointly updates the entire mean and variance networks in a single phase at every iteration, so there is no extra alternating factor. Consequently, the overall complexity of our method over total optimization iterations is $O(d^3)$ with respect to $d$.

**Empirical Analysis.** To quantify the extra computational overhead incurred by the surrogate loss and the weighted loss scheduling, we compare our method against NOTEARS-MLP and ICDH in terms of wall-clock training time, total number of optimization iterations, and peak GPU memory usage on the ER2 synthetic graph with $d = 20$ under the HNM. All methods are implemented within the same augmented Lagrangian framework with L-BFGS updates and share identical two-layer MLP backbones and hardware configurations, so that the per-iteration computational complexity scales identically with the number of samples and variables. The surrogate loss and the scheduling mechanism only add simple pointwise computations ($\mathcal{L}_{\text{MSE}}$, $\mathcal{L}_{\text{VarReg}}$, $\mathcal{L}_{\text{StopNLL}}$) and a scalar update for $\lambda_{\text{reg}}(t)$, and thus do not change the asymptotic per-iteration complexity.

Table 12 reports the result. In terms of wall-clock time, our method takes 651.47 seconds, which is slower than NOTEARS-MLP (19.67 seconds). ICDH takes 389.85 seconds and is therefore faster than our method. However, while NOTEARS-MLP and our method require a comparable number of iterations, ICDH needs about 4.7 times more iterations, reaching 131.88M. This larger number of iterations for ICDH is consistent with our time-complexity analysis, according to which ICDH requires $S$ times more iterations. Nevertheless, ICDH finishes training faster than our method because, in its two-phase training procedure, the gradients for the mean and variance are computed and updated separately, which is substantially cheaper than jointly updating all parameters. In contrast, our method updates the full mean and variance networks at every L-BFGS step and evaluates additional surrogate-loss terms, leading to a higher per-iteration cost but a more efficient trajectory in terms of the number of iterations required for convergence. We also measured the training time when we train from the beginning with the NLL objective; in that case, the training took much longer and required many more iterations than our method. This highlights the difficulty of modeling heteroscedastic noise while jointly learning the mean and variance networks. Because our method optimizes an easier objective than NLL and follows a smoother optimization trajectory, it appears to converge faster than directly optimizing the NLL.

### I.9 COMPREHENSIVE COMPARISON WITH VARIANCE ESTIMATION METHODS

To further strengthen the empirical evaluation, we additionally compare with several DAG learning methods that include variance- or likelihood-based objectives: GraN-DAG (Lachapelle et al., 2020), NPVAR (Gao et al., 2020), NOTEARS-ENT (Chen et al., 2023), DAGMA (Bello et al., 2022), and MCP (Deng et al., 2024). These methods implicitly estimate variance or uncertainty through likelihood- or entropy-based losses and therefore use a negative log-likelihood reconstruction objective, which makes them natural baselines in our setting. However, unlike our method, they do not explicitly output an observation-wise variance via a separate network, nor are they derived under the HNM assumption—that is, they do not estimate variance as a function of the parent variables. Instead, variance (or scale) information is absorbed into their loss or regularization terms, and their optimization procedures are not designed to account for the interaction between variance estimation and the acyclicity constraint.

We evaluate all methods on nonlinear synthetic datasets in the HNM setting with ER1 and ER2 graphs and $d \in \{10, 20, 50, 100\}$, using the same two-layer MLP architecture and sigmoid activation, and keeping the default optimization settings of each method. Tables 13-16 report the results in terms of SHD, F1-score, TPR, and FDR.

As can be seen from Table 13, our method consistently achieves the lowest SHD across all graph types and dimensional settings. For ER1 graphs, the advantage of our method over the strongest variance-estimation baselines becomes more pronounced as the number of variables increases. For

example, when $d = 100$, our method attains an SHD of 46.7, whereas the strongest baselines in this setting, DAGMA and NPV AR, achieve 81.4 and 66.1, respectively. A similar trend is observed for ER2 graphs: our method again yields the smallest SHD for all $d$, with particularly large gaps on larger graphs. In contrast, GraN-DAG attains very large SHD values, indicating a tendency to produce severely misspecified structures under HNM.

These results show that merely using a DAG learner with a likelihood- or entropy-based objective that implicitly handles variance is not sufficient to guarantee robust structure learning under HNM. Although GraN-DAG, NPV AR, NOTEARS-ENT, DAGMA, and MCP exploit variance-related information through their objectives, they do not explicitly parameterize observation-wise variance and are not designed to account for the interaction between NLL and the acyclicity constraint in the early stage of training. As the graphs grow larger, these methods tend to overfit, leading to high FDR and SHD, or, conversely, to miss many true edges. In contrast, our method explicitly models the HNM and uses the proposed surrogate loss and graduated loss scheduling to stabilize the joint learning of the mean and variance in the early phase, and then slowly transitions to the standard NLL objective. This design contributes to consistently lower SHD and higher F1-scores than the variance-estimation baselines, especially in higher-dimensional and more complex graph settings.

Table 13: Comparison of SHD results across variance estimation methods in the heteroscedastic setting (mean ± standard deviation over 10 trials).

| Graph | Methods | SHD ($\downarrow$) | | | |
|---|---|---|---|---|---|
| | | $d = 10$ | $d = 20$ | $d = 50$ | $d = 100$ |
| ER1 | GraN-DAG | 22.7 ± 8.179 | 91.6 ± 26.563 | 572.5 ± 74.911 | 2319.2 ± 316.701 |
| | NPVAR | 5.9 ± 3.635 | 14.2 ± 6.143 | 32.3 ± 3.592 | 66.1 ± 5.859 |
| | NOTEARS-ENT | 7.5 ± 3.240 | 17.5 ± 7.153 | 41.4 ± 6.415 | 103.5 ± 6.721 |
| | DAGMA | 5.0 ± 2.309 | 11.0 ± 4.619 | 31.7 ± 7.088 | 81.4 ± 19.478 |
| | MCP | 4.4 ± 2.591 | 12.3 ± 4.644 | 42.1 ± 10.671 | 139.9 ± 61.952 |
| | **Ours** | **3.8 ± 2.272** | **7.4 ± 3.262** | **22.6 ± 4.609** | **46.7 ± 7.988** |
| ER2 | GraN-DAG | 20.4 ± 6.346 | 77.6 ± 20.866 | 438.3 ± 77.379 | 1459.5 ± 215.857 |
| | NPVAR | 14.2 ± 4.158 | 32.5 ± 5.642 | 80.8 ± 8.025 | 169.6 ± 8.444 |
| | NOTEARS-ENT | 12.0 ± 3.590 | 29.1 ± 7.909 | 83.9 ± 6.540 | 200.2 ± 14.266 |
| | DAGMA | 11.5 ± 2.799 | 22.1 ± 6.967 | 63.5 ± 9.301 | 146.9 ± 64.851 |
| | MCP | 9.8 ± 3.882 | 23.5 ± 5.855 | 72.1 ± 17.785 | 167.7 ± 23.386 |
| | **Ours** | **7.0 ± 2.236** | **17.5 ± 5.408** | **48.5 ± 16.585** | **132.38 ± 50.838** |

Table 14: Comparison of F1-score results across variance estimation methods in the heteroscedastic setting (mean ± standard deviation over 10 trials).

| Graph | Methods | F1-score ($\uparrow$) | | | |
|---|---|---|---|---|---|
| | | $d = 10$ | $d = 20$ | $d = 50$ | $d = 100$ |
| ER1 | GraN-DAG | 0.37 ± 0.133 | 0.23 ± 0.093 | 0.11 ± 0.023 | 0.06 ± 0.010 |
| | NPVAR | 0.62 ± 0.229 | 0.55 ± 0.181 | 0.57 ± 0.082 | 0.55 ± 0.050 |
| | NOTEARS-ENT | 0.65 ± 0.156 | 0.55 ± 0.146 | 0.48 ± 0.079 | 0.19 ± 0.052 |
| | DAGMA | 0.67 ± 0.192 | 0.66 ± 0.168 | 0.63 ± 0.080 | 0.60 ± 0.072 |
| | MCP | 0.73 ± 0.201 | 0.66 ± 0.136 | 0.59 ± 0.057 | 0.51 ± 0.094 |
| | **Ours** | **0.79 ± 0.132** | **0.79 ± 0.111** | **0.74 ± 0.053** | **0.73 ± 0.052** |
| ER2 | GraN-DAG | 0.54 ± 0.115 | 0.38 ± 0.067 | 0.24 ± 0.046 | 0.16 ± 0.022 |
| | NPVAR | 0.47 ± 0.171 | 0.48 ± 0.105 | 0.45 ± 0.071 | 0.41 ± 0.029 |
| | NOTEARS-ENT | 0.65 ± 0.111 | 0.59 ± 0.132 | 0.40 ± 0.073 | 0.18 ± 0.068 |
| | DAGMA | 0.59 ± 0.133 | 0.64 ± 0.164 | 0.58 ± 0.075 | 0.60 ± 0.043 |
| | MCP | 0.68 ± 0.143 | 0.67 ± 0.111 | 0.59 ± 0.070 | 0.56 ± 0.036 |
| | **Ours** | **0.79 ± 0.071** | **0.76 ± 0.087** | **0.71 ± 0.100** | **0.62 ± 0.080** |

Table 15: Comparison of TPR results across variance estimation methods in the heteroscedastic setting (mean ± standard deviation over 10 trials).

| Graph | Methods | TPR (↑) | | | |
|-------|---------|---------|---------|---------|----------|
| | | $d = 10$ | $d = 20$ | $d = 50$ | $d = 100$ |
| ER1 | GraN-DAG | 0.70 ± 0.200 | 0.67 ± 0.103 | 0.69 ± 0.076 | **0.69 ± 0.072** |
| | NPVAR | 0.56 ± 0.196 | 0.51 ± 0.158 | 0.53 ± 0.102 | 0.49 ± 0.046 |
| | NOTEARS-ENT | 0.76 ± 0.196 | 0.59 ± 0.160 | 0.41 ± 0.086 | 0.13 ± 0.041 |
| | DAGMA | 0.59 ± 0.223 | 0.59 ± 0.205 | 0.57 ± 0.105 | 0.62 ± 0.091 |
| | MCP | 0.69 ± 0.233 | 0.68 ± 0.190 | 0.64 ± 0.099 | **0.69 ± 0.088** |
| | **Ours** | **0.77 ± 0.127** | **0.76 ± 0.157** | **0.71 ± 0.069** | 0.66 ± 0.073 |
| ER2 | GraN-DAG | 0.69 ± 0.116 | 0.63 ± 0.071 | **0.69 ± 0.078** | **0.69 ± 0.031** |
| | NPVAR | 0.41 ± 0.178 | 0.46 ± 0.134 | 0.40 ± 0.071 | 0.35 ± 0.028 |
| | NOTEARS-ENT | 0.67 ± 0.092 | 0.59 ± 0.159 | 0.30 ± 0.071 | 0.11 ± 0.047 |
| | DAGMA | 0.45 ± 0.127 | 0.54 ± 0.188 | 0.48 ± 0.097 | 0.53 ± 0.053 |
| | MCP | 0.59 ± 0.133 | 0.66 ± 0.139 | 0.57 ± 0.094 | 0.57 ± 0.054 |
| | **Ours** | **0.71 ± 0.092** | **0.72 ± 0.109** | 0.63 ± 0.092 | 0.54 ± 0.033 |

Table 16: Comparison of FDR results across variance estimation methods in the heteroscedastic setting (mean ± standard deviation over 10 trials).

| Graph | Methods | FDR (↓) | | | |
|-------|---------|---------|---------|---------|----------|
| | | $d = 10$ | $d = 20$ | $d = 50$ | $d = 100$ |
| ER1 | GraN-DAG | 0.74 ± 0.106 | 0.86 ± 0.070 | 0.94 ± 0.013 | 0.97 ± 0.005 |
| | NPVAR | 0.29 ± 0.308 | 0.38 ± 0.234 | 0.37 ± 0.071 | 0.35 ± 0.071 |
| | NOTEARS-ENT | 0.43 ± 0.143 | 0.47 ± 0.149 | 0.43 ± 0.094 | 0.63 ± 0.071 |
| | DAGMA | **0.17 ± 0.124** | 0.20 ± 0.173 | 0.29 ± 0.117 | 0.42 ± 0.083 |
| | MCP | 0.20 ± 0.140 | 0.31 ± 0.178 | 0.44 ± 0.098 | 0.58 ± 0.132 |
| | **Ours** | 0.18 ± 0.154 | **0.16 ± 0.092** | **0.22 ± 0.043** | **0.18 ± 0.070** |
| ER2 | GraN-DAG | 0.54 ± 0.128 | 0.72 ± 0.064 | 0.85 ± 0.030 | 0.91 ± 0.014 |
| | NPVAR | 0.42 ± 0.204 | 0.48 ± 0.117 | 0.48 ± 0.075 | 0.50 ± 0.039 |
| | NOTEARS-ENT | 0.36 ± 0.143 | 0.41 ± 0.124 | 0.39 ± 0.072 | 0.58 ± 0.132 |
| | DAGMA | **0.12 ± 0.147** | **0.20 ± 0.117** | 0.25 ± 0.079 | **0.27 ± 0.062** |
| | MCP | 0.20 ± 0.183 | 0.31 ± 0.116 | 0.37 ± 0.088 | 0.43 ± 0.072 |
| | **Ours** | **0.12 ± 0.080** | **0.20 ± 0.070** | **0.19 ± 0.125** | 0.29 ± 0.190 |

## J  SUPPLEMENTARY OF FIGURE 2

In Figure 2 of the main paper, we compared the performance of our method with the baselines on the synthetic datasets. This section provides more detailed results related to this.

### J.1  HETEROSCEDASTIC NOISE DATASETS

Tables 17, 18, 19 and 20 present the experimental results on nonlinear heteroscedastic noise synthetic datasets in terms of the SHD, F1-score, TPR, and FDR metrics, respectively. The proposed method consistently achieved the best performance in SHD, F1-score, and FDR. Although ICDH attained the highest TPR, but this is because ICDH produced more edges than the ground-truth DAG; this is evidenced by ICDH's extremely high FDR, indicating that it generates a graph with many spurious edges. In contrast, the proposed method achieved the lowest FDR, meaning it effectively filtered out spurious edges. Moreover, by achieving the highest F1-score, our method overall recovered a structure most similar to the true DAG.

Table 17: SHD results in the heteroscedastic setting (mean ± standard deviation over 10 trials).

| Graph | Methods | SHD (↓) | | | |
| | | $d = 10$ | $d = 20$ | $d = 50$ | $d = 100$ |
|---|---|---|---|---|---|
| ER1 | NOTEARS-MLP | 9.1 ± 4.323 | 19.1 ± 4.636 | 88.9 ± 23.851 | 275.6 ± 49.364 |
| | HOST | 5.4 ± 3.105 | 11.4 ± 2.458 | 84.4 ± 26.740 | 143.8 ± 21.085 |
| | ICDH | 6.7 ± 4.473 | 26.0 ± 11.645 | 303.9 ± 121.223 | 518.8 ± 85.588 |
| | **Ours** | **3.8 ± 2.272** | **7.4 ± 3.262** | **22.6 ± 4.609** | **46.7 ± 7.988** |
| ER2 | NOTEARS-MLP | 12.8 ± 3.682 | 42.0 ± 8.136 | 150.8 ± 40.506 | 438.5 ± 77.740 |
| | HOST | 14.5 ± 3.138 | 38.8 ± 5.689 | 201.8 ± 63.871 | 495.7 ± 155.535 |
| | ICDH | 13.7 ± 4.001 | 50.0 ± 19.037 | 415.3 ± 185.616 | 664.0 ± 194.376 |
| | **Ours** | **7.0 ± 2.236** | **17.5 ± 5.408** | **48.5 ± 16.585** | **132.38 ± 50.838** |

Table 18: F1-score results in the heteroscedastic setting (mean ± standard deviation over 10 trials).

| Graph | Methods | F1-score (↑) | | | |
| | | $d = 10$ | $d = 20$ | $d = 50$ | $d = 100$ |
|---|---|---|---|---|---|
| ER1 | NOTEARS-MLP | 0.54 ± 0.211 | 0.52 ± 0.125 | 0.38 ± 0.070 | 0.30 ± 0.044 |
| | HOST | 0.66 ± 0.193 | 0.67 ± 0.079 | 0.43 ± 0.084 | 0.46 ± 0.041 |
| | ICDH | 0.72 ± 0.171 | 0.54 ± 0.116 | 0.25 ± 0.133 | 0.11 ± 0.018 |
| | **Ours** | **0.79 ± 0.132** | **0.79 ± 0.111** | **0.74 ± 0.053** | **0.73 ± 0.052** |
| ER2 | NOTEARS-MLP | 0.57 ± 0.143 | 0.47 ± 0.083 | 0.36 ± 0.068 | 0.29 ± 0.038 |
| | HOST | 0.49 ± 0.103 | 0.46 ± 0.072 | 0.30 ± 0.064 | 0.27 ± 0.061 |
| | ICDH | 0.65 ± 0.085 | 0.55 ± 0.111 | 0.31 ± 0.135 | 0.17 ± 0.056 |
| | **Ours** | **0.79 ± 0.071** | **0.76 ± 0.087** | **0.71 ± 0.100** | **0.62 ± 0.080** |

Table 19: TPR results in the heteroscedastic setting (mean ± standard deviation over 10 trials).

| Graph | Methods | TPR (↑) | | | |
| | | $d = 10$ | $d = 20$ | $d = 50$ | $d = 100$ |
|---|---|---|---|---|---|
| ER1 | NOTEARS-MLP | 0.57 ± 0.219 | 0.58 ± 0.153 | 0.57 ± 0.059 | 0.61 ± 0.079 |
| | HOST | 0.60 ± 0.190 | 0.66 ± 0.092 | 0.65 ± 0.081 | 0.66 ± 0.065 |
| | ICDH | **0.83 ± 0.149** | **0.78 ± 0.174** | **0.81 ± 0.068** | **0.82 ± 0.045** |
| | **Ours** | 0.77 ± 0.127 | 0.76 ± 0.157 | 0.71 ± 0.069 | 0.66 ± 0.073 |
| ER2 | NOTEARS-MLP | 0.50 ± 0.151 | 0.51 ± 0.066 | 0.44 ± 0.066 | 0.45 ± 0.020 |
| | HOST | 0.43 ± 0.112 | 0.48 ± 0.069 | 0.46 ± 0.056 | 0.47 ± 0.042 |
| | ICDH | 0.70 ± 0.100 | **0.78 ± 0.148** | **0.78 ± 0.067** | **0.77 ± 0.087** |
| | **Ours** | **0.71 ± 0.092** | 0.72 ± 0.109 | 0.63 ± 0.092 | 0.54 ± 0.033 |

Table 20: FDR results in the heteroscedastic setting (mean ± standard deviation over 10 trials).

| Graph | Methods | FDR (↓) | | | |
| | | $d = 10$ | $d = 20$ | $d = 50$ | $d = 100$ |
|---|---|---|---|---|---|
| ER1 | NOTEARS-MLP | 0.47 ± 0.248 | 0.54 ± 0.112 | 0.71 ± 0.070 | 0.80 ± 0.035 |
| | HOST | 0.27 ± 0.210 | 0.31 ± 0.077 | 0.67 ± 0.100 | 0.65 ± 0.041 |
| | ICDH | 0.35 ± 0.201 | 0.56 ± 0.122 | 0.84 ± 0.111 | 0.94 ± 0.010 |
| | **Ours** | **0.18 ± 0.154** | **0.16 ± 0.092** | **0.22 ± 0.043** | **0.18 ± 0.070** |
| ER2 | NOTEARS-MLP | 0.33 ± 0.156 | 0.56 ± 0.106 | 0.69 ± 0.081 | 0.79 ± 0.037 |
| | HOST | 0.42 ± 0.141 | 0.55 ± 0.089 | 0.77 ± 0.061 | 0.81 ± 0.055 |
| | ICDH | 0.39 ± 0.108 | 0.55 ± 0.154 | 0.79 ± 0.136 | 0.90 ± 0.048 |
| | **Ours** | **0.12 ± 0.080** | **0.20 ± 0.070** | **0.19 ± 0.125** | **0.29 ± 0.190** |

## J.2 HOMOSCEDASTIC NOISE DATASETS

Tables 21, 22, 23 and 24 present the results in terms of the SHD, F1-score, TPR, and FDR metrics in the homoscedastic noise setting. The evaluation results show a similar trend in the homoscedastic noise environment. The proposed method achieved the best performance in terms of SHD, F1-score, and FDR, whereas ICDH recorded a high TPR but also an extremely high FDR, indicating that its prediction includes many spurious edges. In contrast, the proposed method recovered a structure that is overall the closest to the true DAG while excluding unnecessary edges.

Table 21: SHD results in the homoscedastic setting (mean ± standard deviation over 10 trials).

| Graph | Methods | SHD ($\downarrow$) | | | |
| | | $d = 10$ | $d = 20$ | $d = 50$ | $d = 100$ |
|---|---|---|---|---|---|
| ER1 | NOTEARS-MLP | 4.9 ± 2.071 | 12.6 ± 2.764 | 33.2 ± 5.758 | 62.0 ± 9.726 |
| | GOLEM-EV | 8.0 ± 1.732 | 25.8 ± 6.508 | 78.1 ± 8.994 | 159.5 ± 7.487 |
| | HOST | 4.1 ± 1.221 | 14.8 ± 7.373 | 62.8 ± 19.302 | 132.9 ± 19.429 |
| | ICDH | 3.8 ± 1.778 | 7.3 ± 4.051 | 23.7 ± 7.403 | 111.5 ± 36.456 |
| | **Ours** | **3.3 ± 1.792** | **6.7 ± 3.716** | **15.7 ± 4.196** | **43.6 ± 8.273** |
| ER2 | NOTEARS-MLP | 12.6 ± 3.800 | 29.4 ± 5.024 | 72.7 ± 10.696 | 151.9 ± 17.524 |
| | GOLEM-EV | 18.0 ± 2.049 | 42.8 ± 5.154 | 117.0 ± 6.856 | 239.2 ± 10.971 |
| | HOST | 13.8 ± 4.142 | 43.4 ± 10.200 | 244.6 ± 58.563 | 462.8 ± 81.906 |
| | ICDH | 6.0 ± 2.608 | 17.4 ± 4.652 | 79.3 ± 16.273 | 332.7 ± 83.568 |
| | **Ours** | **5.7 ± 2.795** | **14.5 ± 6.136** | **33.7 ± 6.709** | **88.0 ± 10.392** |

Table 22: F1-score results in the homoscedastic setting (mean ± standard deviation over 10 trials).

| Graph | Methods | F1-score ($\uparrow$) | | | |
| | | $d = 10$ | $d = 20$ | $d = 50$ | $d = 100$ |
|---|---|---|---|---|---|
| ER1 | NOTEARS-MLP | 0.68 ± 0.161 | 0.60 ± 0.094 | 0.63 ± 0.034 | 0.65 ± 0.056 |
| | GOLEM-EV | 0.42 ± 0.156 | 0.25 ± 0.108 | 0.27 ± 0.066 | 0.26 ± 0.045 |
| | HOST | 0.74 ± 0.087 | 0.59 ± 0.138 | 0.51 ± 0.078 | 0.48 ± 0.037 |
| | ICDH | 0.80 ± 0.104 | 0.81 ± 0.113 | 0.78 ± 0.051 | 0.61 ± 0.075 |
| | **Ours** | **0.81 ± 0.111** | **0.80 ± 0.121** | **0.83 ± 0.040** | **0.74 ± 0.061** |
| ER2 | NOTEARS-MLP | 0.60 ± 0.114 | 0.53 ± 0.086 | 0.55 ± 0.054 | 0.55 ± 0.045 |
| | GOLEM-EV | 0.32 ± 0.096 | 0.24 ± 0.080 | 0.24 ± 0.054 | 0.24 ± 0.020 |
| | HOST | 0.54 ± 0.119 | 0.42 ± 0.098 | 0.28 ± 0.071 | 0.29 ± 0.042 |
| | ICDH | 0.85 ± 0.070 | 0.79 ± 0.066 | 0.69 ± 0.037 | 0.52 ± 0.067 |
| | **Ours** | **0.83 ± 0.124** | **0.79 ± 0.094** | **0.81 ± 0.043** | **0.74 ± 0.037** |

Table 23: TPR results in the homoscedastic setting (mean ± standard deviation over 10 trials).

| Graph | Methods | TPR ($\uparrow$) | | | |
| | | $d = 10$ | $d = 20$ | $d = 50$ | $d = 100$ |
|---|---|---|---|---|---|
| ER1 | NOTEARS-MLP | 0.59 ± 0.170 | 0.52 ± 0.123 | 0.61 ± 0.059 | 0.62 ± 0.062 |
| | GOLEM-EV | 0.32 ± 0.147 | 0.22 ± 0.092 | 0.30 ± 0.071 | 0.29 ± 0.057 |
| | HOST | 0.68 ± 0.117 | 0.58 ± 0.119 | 0.69 ± 0.055 | 0.66 ± 0.050 |
| | ICDH | **0.83 ± 0.090** | **0.83 ± 0.119** | **0.88 ± 0.062** | **0.85 ± 0.037** |
| | **Ours** | 0.77 ± 0.135 | 0.76 ± 0.132 | 0.82 ± 0.074 | 0.70 ± 0.064 |
| ER2 | NOTEARS-MLP | 0.51 ± 0.104 | 0.44 ± 0.092 | 0.48 ± 0.051 | 0.49 ± 0.040 |
| | GOLEM-EV | 0.23 ± 0.071 | 0.18 ± 0.064 | 0.20 ± 0.046 | 0.19 ± 0.015 |
| | HOST | 0.48 ± 0.117 | 0.45 ± 0.085 | 0.49 ± 0.068 | 0.49 ± 0.048 |
| | ICDH | **0.90 ± 0.114** | **0.84 ± 0.102** | **0.90 ± 0.026** | **0.88 ± 0.032** |
| | **Ours** | 0.78 ± 0.163 | 0.72 ± 0.115 | 0.76 ± 0.061 | 0.66 ± 0.039 |

Table 24: FDR results in the homoscedastic setting (mean ± standard deviation over 10 trials).

| Graph | Methods | FDR ($\downarrow$) $d = 10$ | $d = 20$ | $d = 50$ | $d = 100$ |
|-------|---------|----------|----------|----------|-----------|
| ER1 | NOTEARS-MLP | 0.16 ± 0.160 | 0.26 ± 0.095 | 0.33 ± 0.083 | 0.31 ± 0.069 |
| | GOLEM-EV | 0.36 ± 0.134 | 0.67 ± 0.185 | 0.75 ± 0.064 | 0.76 ± 0.037 |
| | HOST | 0.16 ± 0.119 | 0.38 ± 0.173 | 0.59 ± 0.094 | 0.62 ± 0.036 |
| | ICDH | 0.23 ± 0.127 | 0.21 ± 0.124 | 0.28 ± 0.097 | 0.52 ± 0.084 |
| | **Ours** | **0.14 ± 0.101** | **0.14 ± 0.117** | **0.16 ± 0.045** | **0.21 ± 0.069** |
| ER2 | NOTEARS-MLP | 0.21 ± 0.219 | 0.30 ± 0.125 | 0.34 ± 0.088 | 0.37 ± 0.065 |
| | GOLEM-EV | 0.40 ± 0.155 | 0.59 ± 0.120 | 0.68 ± 0.065 | 0.68 ± 0.038 |
| | HOST | 0.38 ± 0.146 | 0.59 ± 0.123 | 0.80 ± 0.066 | 0.80 ± 0.036 |
| | ICDH | 0.18 ± 0.071 | 0.25 ± 0.057 | 0.43 ± 0.053 | 0.63 ± 0.072 |
| | **Ours** | **0.12 ± 0.076** | **0.12 ± 0.077** | **0.13 ± 0.030** | **0.16 ± 0.036** |

## K  SUPPLEMENTARY OF TABLE 1

In the main text Table 1, we performed experiments on real-world datasets in terms of the SHD and F1-score. In addition to this, Table 25 reports TPR and FDR. On the Sachs and SynTReN datasets, our method consistently achieves low FDR and high TPR, which is consistent with the SHD and F1-score results in Table 1. On the Causal Assembly dataset, HOST attains the lowest FDR and highest TPR. This suggests that first learning a causal ordering of the variables and then reconstructing the DAG via conditional independence tests is effective in complex manufacturing systems. Among the methods that perform continuous DAG learning without separately learning an ordering, our method achieves the second-lowest FDR after HOST.

Table 25: FDR and TPR results on the real-world datasets.

| Method | Sachs ($d = 11$) FDR($\downarrow$) | TPR($\uparrow$) | SynTReN ($d = 20$) FDR($\downarrow$) | TPR($\uparrow$) | CausalAssembly ($d = 40$) FDR($\downarrow$) | TPR($\uparrow$) |
|--------|------------|----------|------------|----------|------------|----------|
| NOTEARS-MLP | 0.72 ± 0.027 | 0.25 ± 0.017 | 0.92 ± 0.043 | 0.37 ± 0.191 | 0.92 ± 0.049 | 0.06 ± 0.046 |
| HOST | 0.68 ± 0.024 | 0.35 ± 0.015 | 0.91 ± 0.043 | 0.39 ± 0.176 | **0.76 ± 0.028** | **0.13 ± 0.042** |
| ICDH | 0.71 ± 0.028 | 0.38 ± 0.018 | 0.92 ± 0.045 | 0.38 ± 0.166 | 0.91 ± 0.029 | **0.13 ± 0.050** |
| **Ours** | **0.55 ± 0.031** | **0.41 ± 0.015** | **0.88 ± 0.062** | **0.40 ± 0.209** | 0.87 ± 0.065 | 0.11 ± 0.036 |

