# OpenReview forum: "Progressive Acyclicity Induction for Effective DAG Learning under Heteroscedastic Noise Models"
_ICLR.cc/2026/Conference — Submitted to ICLR 2026_

### Official Review · Reviewer_fhFt · 2025-10-28

**Soundness:** 3
**Presentation:** 2
**Contribution:** 2
**Rating:** 4
**Confidence:** 4

**Summary:**

This paper addresses the problem of learning causal directed acyclic graphs (DAGs) under heteroscedastic noise models (HNMs), where noise variance depends on parent variables. The authors identify that when using negative log-likelihood (NLL) loss with gradient-based optimization, the acyclicity constraint can dominate early training due to variance-scaled gradients, hindering effective structure learning. To address this, they propose a graduated optimization strategy with weighted loss scheduling that uses a surrogate loss (combining MSE, variance regularization, and stop-gradient NLL) in early training, then gradually transitions to standard NLL.

**Strengths:**

1. The paper clearly identifies and empirically demonstrates (Figure 1, contribution ratio analysis) a specific optimization challenge in DAG learning under HNMs that hasn't been explicitly addressed before.
2. The paper includes extensive experiments with ablation studies (Section 4.2), hyperparameter analysis (Appendix D), trajectory visualizations (Figures 10-11), and comparisons across multiple baselines.
3. The proposed method achieves substantial improvements over baselines, particularly for larger and denser graphs, demonstrating practical value for the causal discovery community.

**Weaknesses:**

1. Lemma 3.1 only provides a sufficient condition without proving it's satisfied during training.
2. Claim 1 relies on multiple strong assumptions (uniform convergence, PL condition, "slow transition," "reconstruction-dominant phase") that are neither formally defined nor empirically verified.
3. The "proof" in Appendix E.6 is informal with undefined constants (C₁, C₂, C', κ).
4. Paper suggests ranges (λ_reg(0)∈[100,130], t*∈[1000,1200]) but no automatic selection procedure.
5. No guidance on adapting these parameters to different problem characteristics (graph density, sample size, etc.).
6. The main contribution is the scheduling heuristic, but similar ideas exist in the graduated non-convexity literature.
7. Section I.7 compares with methods (DARING, ReScore) designed for different problem settings (heterogeneous data vs. heteroscedastic noise). The comparison appears somewhat unfair, as these methods solve related but distinct problems, and there are missing comparisons with some recent gradient-based DAG learning methods.
8. The contribution score metric (Eq. 5) needs a more intuitive explanation.
9. For Claim 1, can you make the constants (C₁, C₂) explicit and empirically verify that the stated assumptions (uniform convergence bound, PL condition, etc.) actually hold in your experimental settings?
10. How does the computational cost (wall-clock time, number of gradient evaluations) compare to baselines, especially ICDH, which also uses alternating optimization?
11. Why does ICDH fail catastrophically at d=1000 (SHD=5158) while your method still performs reasonably (SHD=1130)? This difference seems surprisingly large and warrants investigation.
12. The contribution ratio (Eq. 5) is a novel diagnostic. Have you considered using it as a stopping criterion or to adjust scheduling parameters during training adaptively?
13. You mention connections to homotopy optimization and GNC, but the analysis is informal. Can you formalize this connection and leverage existing homotopy theory for convergence guarantees?

**Questions:**

Please see Weaknesses.

---

> ### Author Response · Authors · 2025-11-21
> **Rebuttal by Authors**
>
> Dear Reviewer fhFt, thank you so much for your time and effort in reviewing our paper.
>
> ## **Response to W1:**
> Lemma 3.1 formalizes the condition under which the early training phase becomes reconstruction-dominant and provides a theoretical handle for interpreting how our optimization strategy operates. Achieving such a reconstruction-dominant phase in the initial stage of training can directly influence the quality of the final solution.
>
> We provide empirical evidence that this condition is in fact satisfied in practice in Appendix D (Figures 6–10), where we visualize how the contribution ratio, the acyclicity constraint, and the optimization trajectory change as we adjust our scheduling.
>
> These experiments demonstrate that $\lambda_{\mathrm{reg}}(t)$ indeed creates an early training phase in which the reconstruction loss dominates the parameter updates during training, and that the transition from the surrogate loss to the NLL loss is sufficiently smooth in practice.
>
> In Section 3.2 (Line 351-357) of the revised version, we have added this clarification.
>
> ## **Response to W2:**
> First, the goal of Claim 1 is *not* to provide a fully general, non-asymptotic guarantee that applies to arbitrary network architectures, optimization algorithms, or data distributions. Rather than serving as a universal guarantee, it should be understood as an idealized result that offers intuition under standard assumptions.
>
> Regarding the assumptions mentioned in W2, Assumptions (iii) and (iv) are formalized using Assumption (i) and Lemma 3.1 (Eq. (14)–(15)), and they are empirically supported in Appendix D (Figures 6–10). These experiments show that, **when the scheduling coefficient decays sufficiently gradually, reconstruction-dominant phase can be obtained in the early stage of training**. Assumptions (i) and (ii) follow standard practice related to PL-type convergence in homotopy optimization [1] and graduated non-convexity methods [13].
>
> In our work, we use these assumptions as idealized conditions to relate the empirical risk and the population risk and to derive a sample-dependent bound. Verifying global uniform convergence and PL-type conditions for nonlinear neural networks is generally intractable, so we adopt relaxed assumptions in the spirit of [1, 13] and complement Claim 1 with empirical evidence for the behavior of our scheduling scheme and optimization trajectory.
>
> We have added this clarification in Section 3.2 (Lines 359–367) and Appendix E.6 (Lines 1435–1457) of the revised version.
>
>
> ## **Response to W3:**
> In Appendix E.6 (Line 1404, 1412, 1422, and 1430) of **revised version**, **we have explicitly defined the constants**  $C_1, C_2, C’, \kappa$.
>
> In particular, $\kappa$ denotes the condition number of the Hessian as defined in Eq. (34). The constant $C′$ arises from the gradient decomposition in Eq. (46), and  $C_2 = C′/(2\mu)$ follows directly from Eq. (47). Finally, $C_1 = 2$ is the constant that appears in the standard empirical-to-population risk decomposition.
>
> ## **Response to W4:**
> We intentionally did not propose an automatic selection procedure or graph-specific rules for $\lambda_{\mathrm{reg}}(0)$ and $t^ *$. The goal of our work is not to maximize performance tailored to specific graph characteristics, but rather to address the failure mode we identify under HNM. In all experiments, we fixed $\lambda_{\mathrm{reg}}(0)=100$, $t^ * =1000$ and $\tau=5$ without any specific tuning. In Section 4 (Line 372-373) of the **revised version, we have added this clarification**.
>
> The ranges $\lambda_{\mathrm{reg}}(0) \in \[100,130\], t^* \in \[1000,1200\]$ reported in Appendix D are not the result of a search for optimal performance. Instead, they specify **the range of scheduling settings under which the early reconstruction-dominant phase defined in Lemma 3.1 is satisfied and the smoothness of the scheduling coefficient in Assumption 1 (Appendix C.1) holds**. These experiments are intended to provide empirical evidence that our assumptions are approximately satisfied in practice, thereby supporting the plausibility of Claim 1.
>
> Regarding performance, the fixed hyperparameter choice used in our experiments is just one reasonable setting, and, as shown in Appendix I.6, the performance variation within the above range is small and stable (around SHD $\pm 1.5$). Moreover, using settings outside this range, where the assumptions are no longer satisfied, we observe that the failure mode we identify indeed appears and the performance degrades, further validating the role of these assumptions.

---

> ### Author Response · Authors · 2025-11-21
> **Rebuttal by Authors**
>
> ## **Response to W5:**
> Across different graph density settings (ER graphs with $k \in \\{3, 4, 5\\}$) in Appendix I.1) and graph size settings ($d \in \\{10, 20, 50, 100, 1000\\}$, in Appendix I.2 and J), our method consistently achieved the best or near-best performance among the compared baselines when using the same fixed hyperparameters.
>
> This suggests that the failure mode we identify under HNM is not a problem that needs to be addressed by adapting the hyperparameters to graph characteristics. Rather, **our optimization strategy alone can robustly resolve this issue across a wide range of graph characteristics**.
>
> ## **Response to W6:**
> Our contribution is not centered on the scheduling heuristic itself.
> Rather, our contributions are:
> (i) We empirically demonstrate and **formalize a failure mode specific to DAG learning under HNMs**, in which, during the early stages of training, the acyclicity constraint dominates the updates of $W$. We show this empirically via the gradient contribution ratio analysis and provide a formal characterization of this phenomenon.
> (ii) Targeting this failure mode, we design a surrogate loss that decomposes the standard NLL into an easier-to-optimize form, together with an optimization strategy in which this **surrogate loss drives the updates in early training and then gradually transitions to the standard NLL, so that the optimization objective under HNM can be effectively achieved**.
>
> Existing graduated non-convexity [13] and homotopy-based methods [2] mainly address relatively simple settings such as standard regression or bivariate linear DAGs [2]. In contrast, we extend these ideas from simple linear and bivariate settings to a **nonlinear multivariate HNM** regime, which is more realistic and challenging. In this regime, global optimality is no longer guaranteed, but we derive sample-dependent conditions under which the proposed optimization strategy is sufficient to improve DAG recovery, and we verify these conditions empirically.
>
> The specific design of the surrogate loss that targets the identified failure mode comprising $L_{\mathrm{MSE}}$, $L_{\mathrm{VarReg}}$, and $L_{\mathrm{stopNLL}}$, and the explicit goal of transitioning back to the standard NLL have not been studied in prior work. The schedule with surrogate loss **serves as an auxiliary optimization mechanism** introduced to improve the initial optimization trajectory.
>
> In Section 1 (Lines 67–91) of the **revised version**, we have revised the text to more clearly **emphasize our contributions and the overall objective of the proposed method**.
>
>
> ## **Response to W7:**
>
> ### **(1) Comparison with heterogeneous data methods.**
>
> In **Sections B.4 and I.7**, we explicitly acknowledge that the heterogeneous data methods (DARING [3], ReScore [4]) are designed for a **different problem setting from ours**. We do not use these results as our primary baselines. Instead, they are included only as **auxiliary out-of-setting experiments** to illustrate, at a conceptual level, **how our approach differs from heterogeneous data methods; the performance differences observed here should not be interpreted as evidence of superiority**. (This is analogous to the position taken in ICDH [5], which separately reports heterogeneous data experiments under a different problem setting in Appendix H.2.)
>
> In Appendix I.7 (Lines 1979–1981) of the **revised version**, we have added a clarification that these experiments are separated from our main results and contribution.
>
>
> ### **(2) More comparisons with recent gradient-based DAG learning methods.**
>
> In **Appendix I.9 (Tables 13–16) of the revised version**, we have added evaluation results for the recent gradient-based DAG learning methods NOTEARS-ENT [6], NPVAR [7], GraN-DAG [8], DAGMA [9], and MCP [10].
>
> We note that these new baselines are not designed under the HNM assumption; in particular, they do not estimate input-dependent variance, unlike our method. All methods are run under exactly the same protocol as in Section 4.1 (the same ER1 and ER2 graph, $d \in \\{10, 20, 50, 100\\}$, $N = 1000$) samples, and the same HNM setting using an identical MLP architecture when applicable). Our method is competitive with or outperforms these baselines, consistently achieving **the lowest SHD and the highest F1** across all settings, while **maintaining comparable or higher TPR and substantially lower FDR**.
>
>
> (Continue in next comment)

---

> ### Author Response · Authors · 2025-11-21
> **Rebuttal by Authors**
>
> The results below are part of our response to W7.
>
> - **Comparison of SHD (↓) results**
> | Graph | Methods      | d = 10   | d = 20    | d = 50     | d = 100        |
> |-------|--------------|-------------------|--------------------|---------------------|-------------------------|
> | ER1   | GraN-DAG     | 22.7 ± 8.179      | 91.6 ± 26.563      | 572.5 ± 74.911      | 2319.2 ± 316.701        |
> | ER1   | NPVAR        | 5.9 ± 3.635       | 14.2 ± 6.143       | 32.3 ± 3.592        | 66.1 ± 5.859            |
> | ER1   | NOTEARS-ENT  | 7.5 ± 3.240       | 17.5 ± 7.153       | 41.4 ± 6.415        | 103.5 ± 6.721           |
> | ER1   | DAGMA        | 5.0 ± 2.309       | 11.0 ± 4.619       | 31.7 ± 7.088        | 81.4 ± 19.478           |
> | ER1   | MCP          | 4.4 ± 2.591       | 12.3 ± 4.644       | 42.1 ± 10.671       | 139.9 ± 61.952          |
> | ER1   | **Ours**     | **3.8 ± 2.272**   | **7.4 ± 3.262**    | **22.6 ± 4.609**    | **46.7 ± 7.988**        |
> | ER2   | GraN-DAG     | 20.4 ± 6.346      | 77.6 ± 20.866      | 438.3 ± 77.379      | 1459.5 ± 215.857        |
> | ER2   | NPVAR        | 14.2 ± 4.158      | 32.5 ± 5.642       | 80.8 ± 8.025        | 169.6 ± 8.444           |
> | ER2   | NOTEARS-ENT  | 12.0 ± 3.590      | 29.1 ± 7.909       | 83.9 ± 6.540        | 200.2 ± 14.266          |
> | ER2   | DAGMA        | 11.5 ± 2.799      | 22.1 ± 6.967       | 63.5 ± 9.301        | 146.9 ± 64.851          |
> | ER2   | MCP          | 9.8 ± 3.882       | 23.5 ± 5.855       | 72.1 ± 17.785       | 167.7 ± 23.386          |
> | ER2   | **Ours**     | **7.0 ± 2.236**   | **17.5 ± 5.408**   | **48.5 ± 16.585**   | **132.38 ± 50.838**     |
>
>
> - **Comparison of F1-score (↑) results**
> | Graph | Methods      | d = 10 | d = 20 | d = 50 | d = 100 |
> |-------|--------------|----------------------|----------------------|----------------------|-----------------------|
> | ER1   | GraN-DAG     | 0.37 ± 0.133         | 0.23 ± 0.093         | 0.11 ± 0.023         | 0.06 ± 0.010          |
> | ER1   | NPVAR        | 0.62 ± 0.229         | 0.55 ± 0.181         | 0.57 ± 0.082         | 0.55 ± 0.050          |
> | ER1   | NOTEARS-ENT  | 0.65 ± 0.156         | 0.55 ± 0.146         | 0.48 ± 0.079         | 0.19 ± 0.052          |
> | ER1   | DAGMA        | 0.67 ± 0.192         | 0.66 ± 0.168         | 0.63 ± 0.080         | 0.60 ± 0.072          |
> | ER1   | MCP          | 0.73 ± 0.201         | 0.66 ± 0.136         | 0.59 ± 0.057         | 0.51 ± 0.094          |
> | ER1   | **Ours**     | **0.79 ± 0.132**     | **0.79 ± 0.111**     | **0.74 ± 0.053**     | **0.73 ± 0.052**      |
> | ER2   | GraN-DAG     | 0.54 ± 0.115         | 0.38 ± 0.067         | 0.24 ± 0.046         | 0.16 ± 0.022          |
> | ER2   | NPVAR        | 0.47 ± 0.171         | 0.48 ± 0.105         | 0.45 ± 0.071         | 0.41 ± 0.029          |
> | ER2   | NOTEARS-ENT  | 0.65 ± 0.111         | 0.59 ± 0.132         | 0.40 ± 0.073         | 0.18 ± 0.068          |
> | ER2   | DAGMA        | 0.59 ± 0.133         | 0.64 ± 0.164         | 0.58 ± 0.075         | 0.60 ± 0.043          |
> | ER2   | MCP          | 0.68 ± 0.143         | 0.67 ± 0.111         | 0.59 ± 0.070         | 0.56 ± 0.036          |
> | ER2   | **Ours**     | **0.79 ± 0.071**     | **0.76 ± 0.087**     | **0.71 ± 0.100**     | **0.62 ± 0.080**      |

---

> ### Author Response · Authors · 2025-11-21
> **Rebuttal by Authors**
>
> ## **Response to W8:**
> Eq. (5) is based on the **metric** proposed by Lan et al. (2019) [11] (which we refer to as the **contribution score** in the paper), adapted to our setting in order to examine how much each loss term influences the update of $W$. Intuitively, at each training step $s$, the dot product $\langle -\Delta W^{(s)},\,g_{\rm rec}^{(s)}\rangle$ measures how much the actual parameter update $\Delta W^{(s)}$ moves in the direction of decreasing the **reconstruction loss**, whereas $\langle -\Delta W^{(s)},\,g_h^{(s)}\rangle$ measures how much it moves in the direction of satisfying the **acyclicity constraint**.
>
> The cumulative quantities $\bar C_{\rm rec}(t)$ and $\bar C_{h}(t)$ in Eq. (5) are the averages of the contribution scores over training steps. Thus, the ratio $\bar C_h(t) / \bar C_{\rm rec}(t)>1$ means that, on average up to current step $t$, the updates of $W$ have been more strongly driven by the **acyclicity constraint** than by the **data reconstruction signal**.
>
> In Section 2.2 (Lines 193–195) of the **revised version**, we have **added a clearer explanation of this contribution score**, together with an explicit citation to Lan et al. (2019) [11].
>
> ## **Response to W9:**
> Since W9 raises the same points as W2 and W3, we address it by referring to our **responses to W2 and W3**. We would appreciate it if you could refer to those responses.
>
> ## **Response to W10:**
> Since our method uses the **same optimization framework and network architecture as NOTEARS-MLP** [14], its time complexity scales in the same way with respect to the number of samples and variables (i.e., $O(d^{3})$ with respect to the number of variables $d$). Our approach **only adds pointwise loss terms in the surrogate loss and a scalar schedule** $\lambda_{\mathrm{reg}}(t)$, which does not increase the time complexity. In contrast, ICDH [5] introduces an additional factor $S$, the number of alternating optimization steps, leading to a time complexity of $O(S d^{3})$.
>
> Also, in **Appendix I.8 and Table 12 of the revised version**, we have added empirical results on **computational complexity**. We report wall-clock training time (in seconds), the total number of optimization iterations (in units of $10^3$), and peak GPU memory usage (in MB) for each method.
>
> Our method exhibits about **1.7× longer wall-clock time** than ICDH due to the **joint optimization of the mean and variance networks**. However, when we directly use the **standard NLL**, the training takes **3.7× longer** than ICDH. This indicates that simultaneously computing gradients with respect to both the mean and variance and updating the corresponding parameters is heavy and converges slowly. By introducing the **surrogate loss**, which alleviates this issue, **we substantially reduce the training time**.
>
> Our method also uses **4.7× fewer iterations** than ICDH (ICDH requires more iterations due to the factor $S$), and compared to NOTEARS-MLP, the number of iterations increases only slightly. We interpret this slight increase as the additional iterations **required in the early training phase** to let the reconstruction loss dominate the updates up to $t ^ * $; this is precisely the mechanism by which our optimization strategy leads to improved performance in DAG learning.
>
>
> - **Computational cost on synthetic ER2 graph (20 nodes)**
> | Method       | Time (s) | Iterations (×10³) | Memory (MB) |
> |-------------|---------:|------------------:|-----------:|
> | NOTEARS-MLP |   19.67  |            25.07  |     540.93 |
> | ICDH        |  389.85  |           131.88  |     542.87 |
> | NLL         | 1463.85  |           163.07  |     550.05 |
> | Ours        |  651.47  |            28.03  |     549.11 |

---

> ### Author Response · Authors · 2025-11-21
> **Rebuttal by Authors**
>
> ## **Response to W11:**
> As can be seen in Appendix J.1 (Tables 17–20), ICDH [5] **attained the highest TPR**, but this is because ICDH produced more edges than the ground-truth DAG. Consequently, as (d) increases, ICDH shows, among the baselines, both a high TPR and at the same time the largest increase in FDR and the largest degradation in performance (SHD, F1-score); this tendency also appears at $d = 1000$. SHD is the number of edge additions, deletions, and reversals needed to transform the estimated DAG into the ground-truth DAG. Hence, the more false positive edges are included in the estimated DAG, the larger the SHD becomes.
>
> In **Appendix I.2 (Table 5) of the revised version**, we have **added the results for** $d = 1000$ **(1,000 nodes) on TPR, FDR, and the total number of predicted edges**. For a ground-truth DAG with **2,000 true edges**, ICDH generates **6,404 edges**, attaining the **highest TPR (0.64)** but also a **high SHD (5158.3)** and a **high FDR (0.95)**. These results are consistent with our analysis. The following table summarizes all metrics at ($d = 1000$) to present these results together.
>
> - **Comparison of SHD, F1-score, FDR, TPR, and the number of edges (1,000 nodes)**
> | Methods      | SHD (↓)            | F1-score (↑)       | FDR (↓)            | TPR (↑)           | # of edges      |
> |-------------|--------------------|--------------------|--------------------|-------------------|-----------------|
> | NOTEARS-MLP | 1592.3 ± 219.8     | 0.18 ± 0.127       | 0.89 ± 0.085       | 0.43 ± 0.142      | 2150 ± 237.61   |
> | ICDH        | 5158.3 ± 598.1     | 0.09 ± 0.066       | 0.95 ± 0.033       | **0.64 ± 0.128**  | 6404 ± 672.31   |
> | Ours        | **1130.5 ± 153.5** | **0.54 ± 0.093**   | **0.42 ± 0.087**   | 0.51 ± 0.109      | 1552 ± 75.66    |
>
> ## **Response to W12:**
> ### **(1) On the “novel diagnostic”.**
>
> Eq. (5) is based on the metric introduced by Lan et al. (2019) [11], which was originally proposed as a tool to measure how much each loss term contributes to parameter updates. In our work, we simply compute this metric at each step, accumulate it, and then take an average. Thus, we do not present it as a new diagnostic tool per se. However, to the best of our knowledge, it is new to use this metric in DAG learning to quantify how much the reconstruction loss and the acyclicity constraint respectively influence the updates of $W$.
>
> ### **(2) On using it for adaptive training.**
> We did not introduce such dynamic optimization mechanisms because **our optimization strategy is designed, under several assumptions and principles, to address the specific failure mode we identify**. In an augmented Lagrangian setting, if the scheduling were adjusted based on data statistics (or quantities like the contribution ratio), the optimization trajectory could become sensitive to noise in the early training phase and to the updates of the Lagrangian multipliers, potentially harming stability and reproducibility.
>
> Moreover, **computing the contribution ratio online during training** would require evaluating and accumulating the inner products between gradients and parameters at every step, which can **introduce substantial overhead as the number of variables grows**. For these reasons, in this paper we deliberately restrict the use of the contribution ratio to a diagnostic tool that verifies which component of the objective dominates the updates in the early stages of training, rather than using it to adapt the training dynamics.

---

> ### Author Response · Authors · 2025-11-21
> **Rebuttal by Authors**
>
> ## **Response to W13:**
> ### **(1) Connections with prior work and convergence guarantees.**
> For nonlinear neural network–based multivariate HNM DAG learning, directly importing the strong global guarantees from the existing homotopy theory literature [1, 2]—such as **global optimality** or “escaping all bad local minima”—**would require very strong additional assumptions**. To the best of our knowledge, this remains an open problem even in related fields. In those works, convergence guarantees are typically formalized for linear models with nonconvex objective functions, and their discussion of extensions to nonlinear neural networks remains at a more informal level.
>
> In our setting, HNM is intrinsically **nonlinear**, so we do **not** claim such global guarantees. Instead, we focus on (i) formalizing and analyzing the **early training phase failure mode that appears under HNM**, and (ii) showing that the optimization strategy designed to mitigate this issue **stably converges to a stationary point under reasonable assumptions**.
>
> ### **(2) Convergence analysis.**
> In Appendix C.1, we apply the general convergence theorem of Wu (1983) [12] in a way similar to Yin et al. (2024) [5], and show that the joint parameter sequence ($\Theta_t, W_t$) converges to the set of stationary points. This convergence analysis starts from Assumption 1 in Appendix C.1, namely the “smoothness of the scheduling coefficient”. This type of condition is analogous to the “stationary trajectory / locally linear tracking” assumptions that arise in homotopy-SGD [1] and in homotopy approaches for bivariate linear DAG models [2].
>
> In Section 3.1 (Line 310) of the **revised version**, we have added an explicit pointer in the main paper indicating that **Appendix C.1 contains the convergence analysis**, and in Appendix E.6 we have clarified that the assumptions for Claim 1 are derived from these prior works, in order to make the connections more explicit.
>
>
> ## References for responses
>
> [1] Gargiani, Matilde, et al. "Convergence analysis of homotopy-sgd for non-convex optimization." arXiv preprint arXiv:2011.10298 (2020).
>
> [2] Deng, Chang, et al. "Global optimality in bivariate gradient-based DAG learning." Advances in Neural Information Processing Systems 36 (2023): 17929-17968.
>
> [3] He, Yue, et al. "Daring: Differentiable causal discovery with residual independence." Proceedings of the 27th ACM SIGKDD conference on knowledge discovery & data mining. 2021.
>
> [4] Zhang, An, et al. "Boosting differentiable causal discovery via adaptive sample reweighting." arXiv preprint arXiv:2303.03187 (2023).
>
> [5] Yin, Naiyu, et al. "Effective causal discovery under identifiable heteroscedastic noise model." Proceedings of the AAAI Conference on Artificial Intelligence. Vol. 38. No. 15. 2024.
>
> [6] Chen, Weilin, et al. "On the role of entropy-based loss for learning causal structure with continuous optimization." IEEE Transactions on Neural Networks and Learning Systems (2023).
>
> [7] Gao M, Ding Y, Aragam B. A polynomial-time algorithm for learning nonparametric causal graphs[J]. Advances in Neural Information Processing Systems, 2020, 33: 11599-11611.
>
> [8] Lachapelle S, Brouillard P, Deleu T, et al. Gradient-Based Neural DAG Learning[C]//International Conference on Learning Representations.
>
> [9] Bello, Kevin, Bryon Aragam, and Pradeep Ravikumar. "Dagma: Learning dags via m-matrices and a log-determinant acyclicity characterization." Advances in Neural Information Processing Systems 35 (2022): 8226-8239.
>
> [10] Deng, Chang, et al. "Markov equivalence and consistency in differentiable structure learning." Advances in Neural Information Processing Systems 37 (2024): 91756-91797.
>
> [11] Lan, Janice, et al. "Lca: Loss change allocation for neural network training." Advances in Neural Information Processing Systems 32 (2019).
>
> [12] Wu, C.F. Jeff. “On the convergence properties of the EM algorithm.” The Annals of Statistics 11.1 (1983): 95–103.
>
> [13] Hazan, Elad, Kfir Yehuda Levy, and Shai Shalev-Shwartz. "On graduated optimization for stochastic non-convex problems." International conference on machine learning. PMLR, 2016.
>
> [14] Zheng, Xun, et al. "Learning sparse nonparametric dags." International conference on artificial intelligence and statistics. Pmlr, 2020.

---

### Official Review · Reviewer_1251 · 2025-10-30

**Soundness:** 2
**Presentation:** 3
**Contribution:** 3
**Rating:** 6
**Confidence:** 4

**Summary:**

This paper addresses an optimization challenge in learning causal structure on Heteroscedastic Noise Models (HNMs). The paper identifies that during training with a negative log-likelihood (NLL) loss, the gradient from the reconstruction term is attenuated by the predicted variance. This allows the acyclicity constraint to dominate early optimization, hindering effective structure learning. To mitigate this, this paper proposes a novel graduated optimization technique that uses a weighted loss schedule. This method starts with a relaxed loss to focus on learning the mean and variance functions before gradually transitioning to the standard NLL and enforcing the acyclicity constraint. The proposed method is empirically shown to outperform existing baselines on synthetic and real-world datasets.

**Strengths:**

Strengths

- The analysis of how the variance term in NLL influences the reconstruction gradient and the acyclicity constraint is insightful.

- The proposed graduated optimization technique is a straightforward yet well-motivated solution to the identified problem.

- The paper provides comprehensive empirical evidence of the proposed method.

**Weaknesses:**

Weaknesses

- The contribution of this work is somewhat incremental. The work builds directly upon the continuous optimization framework and the recent HNM-based DAG learning approach of Yin et al. (2024). The core idea of a loss schedule or curriculum learning, while novel in this specific context, is a known technique in optimization. The advance is therefore more of an important and effective improvement to an existing line of research rather than a novel breakthrough.

- The proposed solution is largely heuristic. The choice of the scheduling function and the specific rate at which the loss transitions are presented as design choices validated empirically, but they lack a rigorous theoretical justification. Moreover, it is unclear whether the change of the objective function to a surrogate will affect the identifiability of the HNM model.

**Questions:**

See the weaknesses above.

---

> ### Author Response · Authors · 2025-11-21
> **Rebuttal by Authors**
>
> Dear Reviewer 1251, we greatly appreciate your thorough and thoughtful review of our paper.
>
> ## **Response to W1:**
> ### **(1) Our contribution.**
>
> Our contribution is not limited to scheduling. Our contribution is two-fold:
>
> - (i) in DAG learning under HNM, we empirically demonstrate (e.g., through an analysis based on the gradient contribution ratio) and formalize a **failure mode** in which, at the early stage of training, the **acyclicity constraint dominates** the update of $W$;
> - (ii) to address the failure mode, we propose a **graduated optimization strategy** based on weighted loss scheduling to start from a **surrogate loss and smoothly transition to the standard NLL**, and gradually activate the acyclicity constraint.
>
> Please note that the scheduling function ($\lambda_{\mathrm{reg}}(t)$) is **only one component** of the proposed optimization strategy designed to resolve the problem we identified. Our optimization strategy was designed **in the spirit** of homotopy optimization [2,3] and graduated non-convexity (GNC) [6] techniques, but it is specifically tailored to DAG learning under HNM and to addressing the optimization issue we discovered.
>
> In Section 1 (Line 67–91) of the revised version, we have further clarified our contribution.
>
> ### **(2) Comparison with ICDH Yin et al. (2024).**
>
> Yin et al. (2024) [5] and our method differ in clear ways; what is shared is the HNM assumption and the network architecture. This network follows the backbone of Zheng et al. (2020) [4]. ICDH, adopting an alternating training scheme similar to [1], trains the mean and variance networks in a two-phase manner: it first trains the mean network and $W$ until convergence, then freezes $W$ and updates only the variance network, and repeats this cycle.
>
> By contrast, our method jointly optimizes the mean, variance, and $W$ throughout the entire training process using the surrogate loss in Eq. (7) together with the scheduling coefficient $\lambda_{\mathrm{reg}}(t)$ in Eq. (8).
>
> In Appendix B.3 (Lines 743–755) of the revised version, we have added a detailed description of the differences between Yin et al. (2024) and our method.

---

> ### Author Response · Authors · 2025-11-21
> **Rebuttal by Authors**
>
> ## **Response to W2:**
> ### **(1) On our homotopy-inspired scheduling strategy.**
> The schedule is not a heuristic chosen by empirically searching over schedules and picking the one that gives the best performance.
> Our chosen schedule is based on homotopy optimization theory [2, 3]: to stably maintain a stationary point, the scheduling coefficient must decrease monotonically. Prior works [2, 3] rigorously justify their scheduling when solving non-convex problems in linear models, but explicitly acknowledge that this justification does not directly extend to nonlinear models. They therefore define a class of admissible schedules under weaker assumptions.
> In the same spirit, for DAG learning under HNM, our scheduling must satisfy the following:
> - (i) it should be sufficiently large at the beginning to induce an early **reconstruction-dominant** phase (Lemma 3.1), and
> - (ii) it should decrease slowly and smoothly so that the objective changes gradually and the continuous path remains stable (Appendix C.1 and E.5).
>
> Given fixed hyperparameters ($\lambda_{\mathrm{reg}}(0), t ^ *, \tau$), our schedule in Eq. (8) is the simplest function that satisfies these conditions; it is not tuned per dataset. Appendix I.6 shows that the performance is robust over a wide range of $\lambda_{\mathrm{reg}}(0), t ^ *, \tau$, and even under a linear schedule. This indicates that the performance gain does not come from heuristically fine-tuned schedules, but from sufficient conditions derived from homotopy theory and the failure mode we identified.
>
> Additionally, in Appendix C.2 we provide a theoretical justification of the **surrogate loss** and show, via a Hessian analysis, that this loss is easier to optimize than the standard NLL. Furthermore, Appendix C.1 shows that our **optimization strategy** converges to a **stationary point**, and Appendix E.6 provides a bound showing that, under standard assumptions, our method can **achieve a smaller risk than the standard NLL**. We also present complementary empirical evidence in Appendix D through additional analysis.
>
> Our focus is therefore on (i) formalizing and analyzing the early training phase **failure mode** that appears under HNM, and (ii) showing that the optimization strategy that mitigates this issue stably converges to a stationary point under reasonable assumptions.
>
> ### **(2) Our optimization strategy does not alter identifiability.**
> Our method does **not** change the model class (we assume the same HNM) nor the **population objective** itself. The schedule with surrogate loss **serves as an auxiliary optimization mechanism** introduced to improve the initial optimization trajectory: They guide the parameters toward a reconstruction-friendly regime and then, in the later stages of training, converge to the standard NLL with the acyclicity constraint.
>
> The discussion of **identifiability** under ideal conditions is based exactly on this final NLL-based objective, and our method uses this final objective as is. Therefore, we would like to emphasize that the proposed **optimization strategy** does **not** alter the identifiability of the HNM model itself; it only improves the early optimization trajectory during training.
>
> In Appendix F (Lines 1540–1547) of the revised version, we have added a clarification that our approach does not affect **identifiability**.
>
>
> ## References for responses
>
> [1] Skafte, Nicki, Martin Jørgensen, and Søren Hauberg. "Reliable training and estimation of variance networks." Advances in Neural Information Processing Systems 32 (2019).
>
> [2] Gargiani, Matilde, et al. "Convergence analysis of homotopy-sgd for non-convex optimization." arXiv preprint arXiv:2011.10298 (2020).
>
>
>
> [3] Deng, Chang, et al. "Global optimality in bivariate gradient-based DAG learning." Advances in Neural Information Processing Systems 36 (2023): 17929-17968.
>
>
> [4] Zheng, Xun, et al. "Learning sparse nonparametric dags." International conference on artificial intelligence and statistics. Pmlr, 2020.
>
>
> [5] Yin, Naiyu, et al. "Effective causal discovery under identifiable heteroscedastic noise model." Proceedings of the AAAI Conference on Artificial Intelligence. Vol. 38. No. 15. 2024.
>
> [6] Hazan, Elad, Kfir Yehuda Levy, and Shai Shalev-Shwartz. "On graduated optimization for stochastic non-convex problems." International conference on machine learning. PMLR, 2016.

---

### Official Review · Reviewer_tzGf · 2025-10-31

**Soundness:** 3
**Presentation:** 4
**Contribution:** 2
**Rating:** 6
**Confidence:** 3

**Summary:**

This paper identify a challenge of learning Directed Acyclic Graphs (DAGs) under a Heteroscedastic Noise Model (HNM), where noise variance depends on causal variables. The authors identify that during early training, the Negative Log-Likelihood (NLL) loss gradient is scaled by the predicted variance, weakening the reconstruction signal and allowing the acyclicity constraint to dominate prematurely, leading to suboptimal DAG structures. To address this, this paper propose a graduated optimization strategy with a weighted loss schedule. Initially, a surrogate loss (combining MSE, variance regularization, and a stop-gradient NLL) is used to stabilize mean and variance learning. And then it  transition to the standard NLL objective and progressively enforcing acyclicity. Experiments on synthetic and real-world data demonstrate improved structure learning accuracy over existing methods.

**Strengths:**

1. The paper is very well written and easy to follow.
1. The paper clearly identifies a previously unexplored optimization challenge specific to DAG learning under HNMs
1. The paper introduces a new optimization strategies for nonlinear heteroscedastic noise model  via continuous optimization.
1. The experimental results well support the claims.

**Weaknesses:**

1. The proposed scheduling mechanism introduces additional hyperparameters (e.g., initial weight $\lambda_{reg}(0)$, transition points $t$ and $t^*$), which require tuning and may affect reproducibility.
2. The surrogate loss and scheduling strategy likely increase the computational cost and training complexity. It would be helpful to include an analysis of computational complexity and report runtime comparisons in the experiments.
3. Including more baseline methods, such as [1, 2, 3], would strengthen the empirical evaluation.
4. Beyond SHD and F1, additional metrics (e.g., FDR and TPR) could provide a more comprehensive evaluation of performance.



[1] Chen, Weilin, et al. "On the role of entropy-based loss for learning causal structure with continuous optimization." *IEEE Transactions on Neural Networks and Learning Systems* (2023).

[2] Gao M, Ding Y, Aragam B. A polynomial-time algorithm for learning nonparametric causal graphs[J]. Advances in Neural Information Processing Systems, 2020, 33: 11599-11611.

[3] Lachapelle S, Brouillard P, Deleu T, et al. Gradient-Based Neural DAG Learning[C]//International Conference on Learning Representations.

**Questions:**

see weaknesses

---

> ### Author Response · Authors · 2025-11-21
> **Rebuttal by Authors**
>
> Dear Reviewer tzGf, thank you so much for your time and effort in reviewing our paper.
>
> ## **Response to W1:**
> In our scheduling mechanism, the three scalar hyperparameters $\lambda_{\mathrm{reg}}(0),\tau $, and $ t^* $ are **fixed to the same values across all experimental settings**. They **do not require any dataset or graph specific tuning and therefore do not pose an obstacle to reproducibility**. Specifically, we set $\lambda_{\mathrm{reg}}(0)=100, t^* =1000, \tau=5$ in all experiments. These values are not finely tuned optima, but rather generic settings that ensure the surrogate loss is trained for a sufficiently long period before transitioning to the standard objective.
>
> Moreover, as shown in Appendix I.6 (Figure 12), SHD remains stable within $\pm 1.5$ for $100 \le \lambda_{\mathrm{reg}}(0) \le 130$ and $1000 \le t^* \le 1200$, confirming that our method is not sensitive to these hyperparameter choices. All reported evaluation results are averaged over 10 random seeds under this fixed hyperparameter configuration, and the released code specifies these values explicitly, ensuring practical reproducibility.
>
> ## **Response to W2:**
> ### **(1) Time complexity.**
> **Our method** uses the same optimization framework and network architecture as **NOTEARS-MLP** [5], so time complexity scales in the same way with respect to the number of samples and variables ($O(d^{3})$ with respect to the number of variables $d$). Ours only adds pointwise loss terms surrogate loss and a scalar schedule $\lambda_{\mathrm{reg}}(t)$, which does not increase the time complexity. In contrast, **ICDH** [4] introduces an additional number of iterations $S$ due to alternating optimization, leading to a time complexity of $O(S d^{3})$.
>
>
> ### **(2) Empirical analysis.**
> In **Appendix I.8 and Table 12 of the revised version**, we have added an analysis of computational complexity and runtime. We report wall-clock training time (in seconds), the total number of optimization iterations (in units of $10^3$), and peak GPU memory usage (in MB) for each method.
>
> **Ours incurs about 1.7 times longer wall-clock time than ICDH** due to joint optimization of the mean and variance networks. However, when **standard NLL is used as-is, it takes 3.7 times longer than ICDH**. This indicates that simultaneously computing gradient with respect to both mean and variance and updating parameters accordingly is heavy and leads to slow convergence. **By introducing surrogate loss to mitigate this issue, we significantly reduced the training time**.
>
> **Ours uses 4.7 times fewer iterations than ICDH** (ICDH requires more iterations due to the effect of $S$), and the number of iterations increases only slightly compared to NOTEARS-MLP. This can be interpreted as a small additional number of iterations incurred in the early training phase, where reconstruction loss is enforced to dominate updates up to $t^*$. We believe this is the reason why our optimization strategy leads to improved performance in DAG learning.
>
>
> - **Computational cost on synthetic ER2 graph (20 nodes)**
> | Method       | Time (s) | Iterations (×10³) | Memory (MB) |
> |-------------|---------:|------------------:|-----------:|
> | NOTEARS-MLP |   19.67  |            25.07  |     540.93 |
> | ICDH        |  389.85  |           131.88  |     542.87 |
> | NLL         | 1463.85  |           163.07  |     550.05 |
> | Ours        |  651.47  |            28.03  |     549.11 |

---

> ### Author Response · Authors · 2025-11-21
> **Rebuttal by Authors**
>
> ## **Response to W3:**
> In Appendix I.9 and Tables 13–16 of the revised version, **we have added experimental results comparing with the three requested baselines**, NOTEARS-ENT [1], NPVAR [2], and GraN-DAG [3].
>
> All methods are run under exactly the same protocol as in Section 4.1 (the same ER1 and ER2 graphs, $d \in \\{10, 20, 50, 100\\}$, $N = 1000$ samples, and the HNM setting using the same MLP architecture whenever applicable).**Our method** is competitive with or outperforms these baselines, **consistently achieving the lowest SHD and the highest F1 across all settings**, while maintaining comparable or higher TPR and substantially lower FDR.
>
> We would like to emphasize that these **new baselines are not designed under the HNM assumption, i.e., they do not aim to estimate input-dependent variance**. The issue we identified can also arise in these methods: due to unstable variance estimation, the reconstruction loss becomes fragile in its interaction with the acyclicity constraint, and this problem becomes more severe as the graph grows larger and denser. The fact that ours still outperforms these baselines in SHD, F1, and FDR provides additional evidence that the proposed optimization strategy is effective for DAG learning, particularly by alleviating the issue we identified under HNM.
>
>
>  - **Comparison of SHD (↓) results across variance estimation methods**
> | Graph | Methods      | d = 10   | d = 20    | d = 50     | d = 100        |
> |-------|--------------|-------------------|--------------------|---------------------|-------------------------|
> | ER1   | GraN-DAG     | 22.7 ± 8.179      | 91.6 ± 26.563      | 572.5 ± 74.911      | 2319.2 ± 316.701        |
> | ER1   | NPVAR        | 5.9 ± 3.635       | 14.2 ± 6.143       | 32.3 ± 3.592        | 66.1 ± 5.859            |
> | ER1   | NOTEARS-ENT  | 7.5 ± 3.240       | 17.5 ± 7.153       | 41.4 ± 6.415        | 103.5 ± 6.721           |
> | ER1   | **Ours**     | **3.8 ± 2.272**   | **7.4 ± 3.262**    | **22.6 ± 4.609**    | **46.7 ± 7.988**        |
> | ER2   | GraN-DAG     | 20.4 ± 6.346      | 77.6 ± 20.866      | 438.3 ± 77.379      | 1459.5 ± 215.857        |
> | ER2   | NPVAR        | 14.2 ± 4.158      | 32.5 ± 5.642       | 80.8 ± 8.025        | 169.6 ± 8.444           |
> | ER2   | NOTEARS-ENT  | 12.0 ± 3.590      | 29.1 ± 7.909       | 83.9 ± 6.540        | 200.2 ± 14.266          |
> | ER2   | **Ours**     | **7.0 ± 2.236**   | **17.5 ± 5.408**   | **48.5 ± 16.585**   | **132.38 ± 50.838**     |
>
>
> (Continue in next comment)

---

> ### Author Response · Authors · 2025-11-21
> **Rebuttal by Authors**
>
> The results below are part of our response to W3.
>
> - **Comparison of F1-score (↑) results across variance estimation methods**
> | Graph | Methods      | d = 10 | d = 20 | d = 50 | d = 100 |
> |-------|--------------|----------------------|----------------------|----------------------|-----------------------|
> | ER1   | GraN-DAG     | 0.37 ± 0.133         | 0.23 ± 0.093         | 0.11 ± 0.023         | 0.06 ± 0.010          |
> | ER1   | NPVAR        | 0.62 ± 0.229         | 0.55 ± 0.181         | 0.57 ± 0.082         | 0.55 ± 0.050          |
> | ER1   | NOTEARS-ENT  | 0.65 ± 0.156         | 0.55 ± 0.146         | 0.48 ± 0.079         | 0.19 ± 0.052          |
> | ER1   | **Ours**     | **0.79 ± 0.132**     | **0.79 ± 0.111**     | **0.74 ± 0.053**     | **0.73 ± 0.052**      |
> | ER2   | GraN-DAG     | 0.54 ± 0.115         | 0.38 ± 0.067         | 0.24 ± 0.046         | 0.16 ± 0.022          |
> | ER2   | NPVAR        | 0.47 ± 0.171         | 0.48 ± 0.105         | 0.45 ± 0.071         | 0.41 ± 0.029          |
> | ER2   | NOTEARS-ENT  | 0.65 ± 0.111         | 0.59 ± 0.132         | 0.40 ± 0.073         | 0.18 ± 0.068          |
> | ER2   | **Ours**     | **0.79 ± 0.071**     | **0.76 ± 0.087**     | **0.71 ± 0.100**     | **0.62 ± 0.080**      |
>
>
> - **Comparison of TPR (↑) results across variance estimation methods**
> | Graph | Methods      | d = 10   | d = 20   | d = 50   | d = 100   |
> |-------|--------------|-------------------|-------------------|-------------------|--------------------|
> | ER1   | GraN-DAG     | 0.70 ± 0.200      | 0.67 ± 0.103      | 0.69 ± 0.076      | **0.69 ± 0.072**   |
> | ER1   | NPVAR        | 0.56 ± 0.196      | 0.51 ± 0.158      | 0.53 ± 0.102      | 0.49 ± 0.046       |
> | ER1   | NOTEARS-ENT  | 0.76 ± 0.196      | 0.59 ± 0.160      | 0.41 ± 0.086      | 0.13 ± 0.041       |
> | ER1   | **Ours**     | **0.77 ± 0.127**  | **0.76 ± 0.157**  | **0.71 ± 0.069**  | 0.66 ± 0.073       |
> | ER2   | GraN-DAG     | 0.69 ± 0.116      | 0.63 ± 0.071      | **0.69 ± 0.078**  | **0.69 ± 0.031**   |
> | ER2   | NPVAR        | 0.41 ± 0.178      | 0.46 ± 0.134      | 0.40 ± 0.071      | 0.35 ± 0.028       |
> | ER2   | NOTEARS-ENT  | 0.67 ± 0.092      | 0.59 ± 0.159      | 0.30 ± 0.071      | 0.11 ± 0.047       |
> | ER2   | **Ours**     | **0.71 ± 0.092**  | **0.72 ± 0.109**  | 0.63 ± 0.092      | 0.54 ± 0.033       |
>
>
> - **Comparison of FDR (↓) results across variance estimation methods**
> | Graph | Methods      | d = 10   | d = 20   | d = 50   | d = 100   |
> |-------|--------------|-------------------|-------------------|-------------------|--------------------|
> | ER1   | GraN-DAG     | 0.74 ± 0.106      | 0.86 ± 0.070      | 0.94 ± 0.013      | 0.97 ± 0.005       |
> | ER1   | NPVAR        | 0.29 ± 0.308      | 0.38 ± 0.234      | 0.37 ± 0.071      | 0.35 ± 0.071       |
> | ER1   | NOTEARS-ENT  | 0.43 ± 0.143      | 0.47 ± 0.149      | 0.43 ± 0.094      | 0.63 ± 0.071       |
> | ER1   | **Ours**     | 0.18 ± 0.154      | **0.16 ± 0.092**  | **0.22 ± 0.043**  | **0.18 ± 0.070**   |
> | ER2   | GraN-DAG     | 0.54 ± 0.128      | 0.72 ± 0.064      | 0.85 ± 0.030      | 0.91 ± 0.014       |
> | ER2   | NPVAR        | 0.42 ± 0.204      | 0.48 ± 0.117      | 0.48 ± 0.075      | 0.50 ± 0.039       |
> | ER2   | NOTEARS-ENT  | 0.36 ± 0.143      | 0.41 ± 0.124      | 0.39 ± 0.072      | 0.58 ± 0.132       |
> | ER2   | **Ours**     | **0.12 ± 0.080**  | **0.20 ± 0.070**  | **0.19 ± 0.125**  | 0.29 ± 0.190       |

---

> ### Author Response · Authors · 2025-11-21
> **Rebuttal by Authors**
>
> ## **Response to W4:**
>
> **Our paper already includes TPR and FDR for all synthetic experiments in Appendices J.1–J.2**. Appendix J.1 (Tables 11–14) reports SHD, F1, TPR, and FDR for the heteroscedastic-noise synthetic datasets, and Appendix J.2 (Tables 15–18) reports the same set of metrics for the homoscedastic-noise synthetic datasets. These experiments match the setting of Figure 2 in the main paper, and can be viewed as an extension of the evaluation metrics from (SHD, F1) to (SHD, F1, TPR, FDR).
>
> Our method maintains a competitive TPR among the baselines while consistently achieving the lowest FDR, thereby obtaining the highest F1 score overall. In contrast, ICDH attains the highest TPR but at the cost of a high FDR, indicating that many of the additionally recovered edges are false positives.
>
> **Due to space limitations in the main paper, Figure 2 focuses on SHD together with F1**, which compactly summarizes the joint behaviour of TPR and FDR for each method.
>
> In Appendix K and Table 25 of revised version, **we have added TPR and FDR results for the real-world dataset**.
>
>
> - **FDR and TPR results on the real-world datasets**
> | Method       | Sachs FDR (↓)     | Sachs TPR (↑)     | SynTReN FDR (↓)    | SynTReN TPR (↑)      | CausalAssembly FDR (↓) | CausalAssembly TPR (↑) |
> |-------------|-------------------|-------------------|--------------------|----------------------|------------------------|------------------------|
> | NOTEARS-MLP | 0.72 ± 0.027      | 0.25 ± 0.017      | 0.92 ± 0.043       | 0.37 ± 0.191         | 0.92 ± 0.049           | 0.06 ± 0.046           |
> | HOST        | 0.68 ± 0.024      | 0.35 ± 0.015      | 0.91 ± 0.043       | 0.39 ± 0.176         | **0.76 ± 0.028**       | **0.13 ± 0.042**       |
> | ICDH        | 0.71 ± 0.028      | 0.38 ± 0.018      | 0.92 ± 0.045       | 0.38 ± 0.166         | 0.91 ± 0.029           | **0.13 ± 0.050**       |
> | **Ours**    | **0.55 ± 0.031**  | **0.41 ± 0.015**  | **0.88 ± 0.062**   | **0.40 ± 0.209**     | 0.87 ± 0.065           | 0.11 ± 0.036           |
>
>
> ## References for responses
>
> [1] Chen, Weilin, et al. "On the role of entropy-based loss for learning causal structure with continuous optimization." IEEE Transactions on Neural Networks and Learning Systems (2023).
>
>
> [2] Gao M, Ding Y, Aragam B. A polynomial-time algorithm for learning nonparametric causal graphs[J]. Advances in Neural Information Processing Systems, 2020, 33: 11599-11611.
>
>
> [3] Lachapelle S, Brouillard P, Deleu T, et al. Gradient-Based Neural DAG Learning[C]//International Conference on Learning Representations.
>
>
> [4] Yin, Naiyu, et al. "Effective causal discovery under identifiable heteroscedastic noise model." Proceedings of the AAAI Conference on Artificial Intelligence. Vol. 38. No. 15. 2024.
>
>
> [5] Zheng, Xun, et al. "Learning sparse nonparametric dags." International conference on artificial intelligence and statistics. Pmlr, 2020.

---

> > ### Comment · Reviewer_tzGf · 2025-11-25
> >
> > Thanks for the detailed responses, which addressed most of my concerns well. The added experiments are satisfactory, and I will keep my positive score.
> >
> > Additionally, regarding hyperparameter sensitivity experiments, I would suggest that the authors could explore wider ranges for the evaluated hyperparameters, on the order of 10$\times$. The current intervals of $[100,130]$ and $[1000,1200]$ appear somewhat arbitrary and odd.

---

> > > ### Author Response · Authors · 2025-12-03
> > >
> > > Dear Reviewer tzGf, thanks for your time and effort devoted to carefully evaluating our manuscript and rebuttal! And we are glad that our responses have addressed most of your concerns.
> > >
> > > Actually, the ranges we described ($\lambda_{\mathrm{reg}}(0) \in [100,130]$ and $t^* \in [1000,1200]$) were obtained after exploring a much wider range, $\lambda_{\mathrm{reg}}(0) \in {1, 10, 20, \dots, 190, 200}$ and $t^* \in {50, 100, 200, \dots, 1900, 2000}$. As explained in Appendix I.6, we performed extensive analysis to verify when Assumptions 1–4 hold in the training process and how DAG learning performance changes within those hyperparameter regions.
> > >
> > > Specifically, in Appendix D.1, we set $\lambda_{\mathrm{reg}}(0) \in {1, 10, 20, \dots, 190, 200}$ and $t^* \in {50, 100, 200, \dots, 1900, 2000}$ and monitored how the contribution ratio and $h(W)$ change throughout training. As a result, we observed that in the actual DAG learning phase, when $100 < \lambda_{\mathrm{reg}}(0) < 130$ (Line 1079) and $1000 < t^* < 1200$ (Line 1112), the Assumptions are well satisfied and the conditions of Lemma 3.1 and Claim 1 hold (Lines 1115–1118). Subsequently, in Appendix I.6, we measured SHD over a wide range $t^* \in [100, 200, \dots, 2100]$ and $\lambda_{\mathrm{reg}}(0) \in [10, 20, \dots, 210]$ (Lines 1922–1924), and confirmed that the best performance is again attained in exactly the same region identified in Appendix D.1. As a result, we have already explored a wider range (as above), and the current intervals of $[100,130]$ and $[1000,1200]$ summarize the stable region where both the theoretical assumptions and empirical performance are simultaneously best satisfied.
> > >
> > > We hope this clarification addresses your concern that the explored range might be odd or somewhat arbitrary.

---

### Official Review · Reviewer_9YYo · 2025-11-11

**Soundness:** 2
**Presentation:** 2
**Contribution:** 1
**Rating:** 4
**Confidence:** 4

**Summary:**

This paper addresses the challenging optimization problem of performing continuous DAG learning under a heteroscedastic noise model. To mitigate the issue of the reconstruction loss constraint being overly attenuated, the paper proposes a graduated optimization strategy that gradually decreases the corresponding coefficients to better enforce both the likelihood objective and the acyclicity constraint. The method demonstrates strong performance on both synthetic and real-world datasets.

**Strengths:**

-	This paper tackles a practical and existing challenge in continuous DAG learning under heteroscedastic noise.
-	The paper provides a thorough discussion of relevant related works and compares the proposed method against appropriate baselines on a reasonable set of benchmark datasets.

**Weaknesses:**

- **Overstated novelty and unclear theoretical distinctions:** Many of the proposed formulations and objectives closely resemble existing methods, raising concerns about the true conceptual contribution. Please see the question 6 and 8 in **Questions** section.
- **Overclaims and inaccurate statements:** The paper contains several overclaims or incorrect assertions, such as implying that using MSE assumes equal noise variance, or misrepresenting how Yin et al. (2024) establishes identifiability and performs DAG recovery. These inaccuracies undermine the technical rigor and weaken the overall argumentation. Please see my specific questions below.
- **Insufficient empirical and theoretical validation:** The paper lacks convergence guarantees or complexity analysis for the proposed algorithm, and some experimental results appear counterintuitive (e.g., heteroscedastic methods underperforming simpler baselines). These issues weaken the empirical credibility of the claimed advantages in both performance and efficiency.

**Questions:**

1.	In lines 39–41, I do not think it is rigorous to claim that using MSE as the reconstruction loss implies an equal noise variance assumption. In particular, Zheng et al. (2020) considers various types of noise, not limited to Gaussian noise.

2.	In lines 50–52, Yin et al. (2024) establishes identifiability by first providing sufficient conditions for the bivariate case and then extending them to the multivariate case using standard procedures from Peters et al. (2014). Algorithmically, it performs gradient-based continuous optimization to directly recover the DAG. In either case, it does not “first learn a causal ordering of the variables and then reconstruct the DAG via conditional independence tests.”

3.	It seems somewhat overstated to claim that one of this paper’s contributions is identifying a new challenge specific to structure learning with HNMs. This optimization issue has been discussed previously, such as in Yin et al. (2024), though perhaps not as clearly formulated or presented as in this work. Moreover, the problem of variance explosion hindering accurate mean estimation under the NLL objective has been well-documented in prior studies. The paper’s novelty lies in proposing a new approach to address this issue within the context of DAG learning, rather than introducing it as a new challenge.

4.	What do the weights in the weighted matrix $W$ represent? In linear SEMs, these weights correspond to causal mechanisms, but since this paper focuses on the nonlinear setting, their interpretation is less clear.

5.	**[Very Important]** What is the difference between the **Theorem 2.1** and **Theorem 1** in Yin et al. (2024)?


6.	What are the specific choices of the functions $\mu_j$ and $\sigma_j^2$ in Eq. (2)? Also, when you mention that $W_j$ represents the directed connectivity from $X_{\text{pa}_j}$ to $X_j$, does this imply that $W$ is a binary vector of 0s and 1s? If so, why is $W$ referred to as a weight matrix?

7.	I wonder if the authors could include a comparison between the proposed algorithm and that of Yin et al. (2024). My concern about novelty arises because, in essence, Eqs. (8) and (9) appear quite similar to the two-phase training objectives in Yin et al. (2024). In particular, the supposed novelty—the stop-gradient NLL—does not seem to contribute to the actual training process, but rather serves as an evaluation metric for model selection.

8.	Since the contributions are merely algorithmic, is there any convergence guarantee for the proposed algorithm? Does the algorithm guarantee to learn the stationary solution as all the existing DAG learning algorithm, or it can go beyond this limitation? What is the complexity analysis of proposed algorithm against its major competitor, ICDH?

9.	The results in Figure 2 appear counterintuitive. Methods specifically designed for heteroscedastic data, such as HOST and ICDH, perform worse than NOTEARS-MLP, which completely ignores potential heteroscedasticity. These results are inconsistent with the original findings reported for HOST and ICDH. It is possible that both methods, being relatively complex, are highly sensitive to initialization and hyperparameter settings, which may have hindered their performance in this evaluation. Therefore, it may not be entirely fair to claim superiority in DAG learning accuracy based on these results.

---

> ### Author Response · Authors · 2025-11-21
>
> Dear Reviewer 9YYo, thank you so much for your time and effort in reviewing our paper.
>
> ## **Response to W1:**
> We do not intend to claim that our network architecture or DAG constraint is fundamentally different from existing methods. As stated in Section 2.1 and Appendix H.5, the continuous weighted adjacency matrix $W$ and the networks $\mu_j$ and $\sigma_j$ we use, follow the same implementation as in NOTEARS-MLP [7] and ICDH [4].
>
> **The contribution of of our work** are:
> (i) We formally **characterize a new failure mode of optimization using NLL with DAG constraint under HNMs via a failure certificate**, and quantitatively identify the early training regime in which the DAG constraint dominates the reconstruction signal;
> (ii) To mitigate this failure mode, we design a **time-varying surrogate objective** together with a scheduling coefficient ($\lambda_{\mathrm{reg}}(t)$) that **gradually transitions the objective to the standard NLL**. We show that the joint parameter sequence converges to a stationary point of the final objective as shown in Appendix C.1.
>
> Related theoretical and empirical details are further elaborated in our responses to **Q3, Q6, and Q8**.
>
>
> ## **Response to W2:**
> In the revised version, we clarify that using an MSE reconstruction loss does not in itself imply an equal-variance Gaussian noise assumption. We have softened the relevant statement in the Introduction. We also corrected a misleading expression in Yin et al. (2024). These corrections are addressed in **Q1 and Q2**.
>
>
> ## **Response to W3:**
> Our proposed method is not an empirical heuristic but is accompanied by a theoretical analysis. In **Appendix C.1, Assumption 1** specifies the **convergence conditions** under which the joint parameter sequence $(\Theta_t, W_t)$ converges to the set of stationary points of the final surrogate objective. Building on the simple settings in which global optimality convergence guarantees can be established in the existing literature [3], we extend the analysis to a **nonlinear multivariate HNM regime with a DAG constraint**. The details of this **convergence analysis**, together with the **complexity analysis** newly added in the revised version, are presented and discussed in detail in our response to **Q8**.
>
>
> Regarding the observation that heteroscedastic methods underperform simpler baselines, we analyze this behavior in **Section 4.1 and Appendix I.2**.  To further support our analysis, we **additionally report TPR, FDR, and the total number of predicted edges for a large graph** with $d=1000$; these new results are included in the revised Appendix I.5 (Table 5). Moreover, to ensure fairness, we use the official implementations and recommended hyperparameters released by the authors for all competing methods. **These experimental clarifications and additional results** are explained in detail in our response to **Q9**.
>
>
> We believe these additions and clarifications strengthen both the theoretical and empirical support for the claimed advantages in performance and efficiency.

---

> ### Author Response · Authors · 2025-11-21
>
> ## **Response to Q1:**
> As the reviewer correctly points out, using an MSE reconstruction loss does not, in a rigorous sense, automatically entail an equal noise variance assumption, and in particular Zheng et al. (2020) [7] considers more general noise models that are not restricted to Gaussian noise. Accordingly, we have refined the wording in that part to be more precise.
>
> What we intended to emphasize is that, in the specific SEM class instantiated by these approaches, the noise term associated with each variable is modeled as **having constant variance across observations for that variable, regardless of how the noise distributions may differ across variables**. We have revised the wording to make this point more precise.
>
> In Section 1 (Line 40-43) of revised version, we have replaced the original sentence “This practice implies an equal noise variance assumption, or so-called homoscedastic noise.” with the following:
> * “These approaches are typically instantiated as SEMs with additive noise. For each variable, the associated noise is modeled as independent and identically distributed across observations with constant variance, which corresponds to homoscedastic noise.”
>
> Furthermore, in Appendix B.3 of the revised version, we explicitly **clarify that the description “equal noise variance across variables and observations” refers specifically to the case of ANMs with Gaussian noise**.
>
> Through these revisions, we have addressed the reviewer’s concern about rigor and properly reflected the more general noise setting considered in Zheng et al. (2020) [7].
>
>
> ## **Response to Q2:**
> We thank the reviewer for pointing out our original description of Yin et al. (2024) [4]. In that passage, our intention was to emphasize that both **Duong & Nguyen (2023) (HOST)** [8] and **Yin et al. (2024) (ICDH)** [4] establish identifiability of HNMs and to contrast their algorithmic approaches;however, our wording mistakenly described ICDH as following the same procedure as HOST.
>
> The phrase “first learn a causal ordering of the variables and then reconstruct the DAG via conditional independence tests” accurately characterizes HOST but, as the reviewer correctly notes, it does not describe ICDH. **ICDH instead establishes identifiability by first deriving sufficient conditions in the bivariate case and then extending these conditions to the multivariate case using standard arguments from Peters et al. (2014), and algorithmically recovers the DAG via gradient-based continuous optimization with a differentiable acyclicity constraint**.
>
> In Section 1 (Lines 52–56) of the revised version, **we have revised the text accordingly to clarify the respective identifiability results and algorithmic procedures of HOST and ICDH**.
>
> We again thank the reviewer for carefully reading our work and for bringing this issue to our attention.

---

> ### Author Response · Authors · 2025-11-21
>
> ## **Response to Q3:**
> ### **(1) Our contribution.**
>
> The optimization issue that variance estimation interferes with mean estimation is already well known in prior work. We review this line of work in the first paragraph of Section 2.2.
>
> Our contribution lies in analyzing and explicitly characterizing **the failure mode of optimization using NLL with DAG constraint under HNMs via a failure certificate**. When we refer to a **“new challenge”** (second paragraph of Section 2.2), we mean that, from the perspective of DAG learning, we pose as a separate challenge the phenomenon whereby, due to the NLL optimization issue, **the reconstruction gradient is not sufficiently reflected and the structural updates become dominated by the DAG constraint**.
>
> The existing optimization issue alone does not adequately explain, in the context of DAG learning, the behaviour where **acyclicity is well satisfied but better structures are not discovered**, nor does it directly suggest a corresponding **design principle** to address this, as illustrated in Figure 1(c).
>
> ### **(2) Difference in the problem addressed compared to Yin et al. (2024).**
> Yin et al. (2024) [4] adopt an **alternative training strategy that separates variance from mean estimation**, similar to [1], in order to mitigate the well-known optimization issue (first paragraph of Section 2.2). This can be effective in alleviating the problem of variance overestimation.
>
> In contrast, following the variance overestimation issue, we introduce **a new perspective that the acyclicity constraint drives the updates of $W$ in the early stage of training** (second paragraph of Section 2.2), and we design **an optimization strategy that directly targets this phenomenon**. In this sense, as the reviewer pointed out, we provide a **“new approach”** to address this issue, and this new approach forms a unified contribution together with the formulation of the underlying problem.
>
> In Section 1 (Lines 67–91) of the revised version, we have explicitly stated that the challenge we raise is that, due to the well-known optimization issue, **the reconstruction gradient is not sufficiently reflected and the structural updates are dominated by the DAG constraint**.
>
>
> ## **Response to Q4:**
> In linear SEM, each entry of the weight matrix can be directly interpreted as the linear causal mechanism between variables, but we agree that this interpretation does not straightforwardly carry over to the nonlinear setting considered in this paper.
>
> However, **the weight matrix $W$ we use is the same form of continuous weighted adjacency matrix as in NOTEARS-MLP [7] and ICDH [4]**, and is introduced for the purpose of parameterizing the DAG. Conceptually, $W_j$ represents the directed connectivity from $X_{pa(j)}$ to $X_j$. Based on this $W$, we define the acyclicity constraint, and after training we apply thresholding to $W$, as in existing methods, to obtain the final binary DAG structure. **These details are described in Appendix H.5.**
>
> In Section 2.1 (Lines 143–144) of the revised version, we have added a sentence that guides the reader to Appendix H.5.

---

> ### Author Response · Authors · 2025-11-21
>
> ## **Response to Q5:**
> Theorem 2.1 is essentially Theorem 1 of Yin et al. (2024) [4] restated in the context of our paper. However, **the theoretical goal of this work** is, under an identifiable HNM condition, **to define the failure certificate we discovered and to design an optimization strategy that resolves it**, as mentioned in Line 121–125. For this reason, Theorem 2.1 is placed in the Background section.
>
> Specifically, Yin et al. (2024) [4] propose the HNM identifiability and an algorithm that addresses the NLL optimization issue. **On the same identifiability basis, we formalize why actual training fails in DAG learning with acyclicity constraint as a failure certificate and propose a new optimization strategy that addresses the problem**. From this perspective, Theorem 2.1 is a background result introduced to clarify the premise of our work, and we would like to emphasize that **the main contributions lie in the failure analysis and the proposed method built upon this premise**.
>
> In the revised version, in Section 2.1 (Line 112), we have added the label “**Theorem 2.1 (Yin et al., 2024, restated)**” at the point where the theorem is stated.
>
>
>
> ## **Response to Q6:**
> In all experiments, we use **the same network as ICDH** [4].
>
> ### **(1) $\mu_j$ and $\sigma_j$.**
> $\mu_j$ is implemented as a **two-layer MLP with a nonlinear activation (sigmoid), as in NOTEARS-MLP** [7].
>
> $\sigma_j$ is implemented as a **linear layer** that takes the intermediate hidden representations of this MLP as input. We then apply a **piecewise nonlinearity (softplus) to its output to ensure non-negativity**.
>
> These choices are made **to satisfy the identifiability conditions in Theorem 2.1**, that $\mu_j$ is **nonlinear** and $\sigma_j$ is a **piecewise function**.
>
>
> ### **(2) $W$.**
> In the main text, we represent $W$ as an element of $\mathbb{R}^{d \times d}$, so its entries are real-valued. The statement that it represents “**directed connectivity**” **is an abstract notion**, used as a symbol to express the DAG and parent variables.
>
> In the actual NOTEARS-MLP [7] implementation, **$W$ is a tensor in $\mathbb{R}^{d \times d \times 2m}$, where $m$ is the hidden dimension of the MLP**. After training, **$W$ is reduced via Eq. (60) to a matrix in $\mathbb{R}^{d \times d}$, and we then apply thresholding to obtain a binary adjacency matrix. This structure is the same as in both NOTEARS-MLP and ICDH**, and we describe it in Appendix H.5.

---

> ### Author Response · Authors · 2025-11-21
>
> ## **Response to Q7**:
> ### **(1) Detailed comparison with ICDH (Yin et al. (2024))**.
> In response to the reviewer’s comment on the similarity to Yin et al. (2024) (ICDH) [4], we first clarify that **our optimization strategy is fundamentally different from that of ICDH**. ICDH, adopting an **alternating training scheme** similar to [1], trains the mean and variance networks in a **two-phase manner**: it first trains the mean network and $W$ until convergence, then freezes $W$ and updates only the variance network, and repeats this cycle. As a result, during the variance updates, the graph $W$ is kept fixed at the value learned in the previous phase, so the parameter updates of the variance function do not directly feed back into the choice of parent variables.
>
> By contrast, **our method jointly optimizes the mean, variance, and $W$ throughout the entire training process** using the surrogate loss in Eq. (7) together with the scheduling coefficient $\lambda_{\mathrm{reg}}(t)$ in Eq. (8). Since $W$ is shared by both the mean and variance networks, parent selection is influenced by both mean and variance estimation, which is consistent with the HNM formulation where the two networks share the same parent set. In particular, to mitigate the harmful interaction between the standard NLL loss and the acyclicity constraint in the early training phase under HNM, the surrogate loss is designed as a decomposition of the NLL into mean and variance terms. This design enables stable joint optimization without significantly increasing computational cost.
>
> In Appendix B.3 (Line 743-755) of the revised version, **we have added detailed comparison with ICDH**.
>
>
> ### **(2) Role of $L_{\mathrm{StopNLL}}$.**
>
> $L_{\mathrm{StopNLL}}$ in Eq. (11) **plays a clearly defined role as a component of the surrogate loss**. It aligns the variance, which has been fitted to the residuals by $L_{\mathrm{varreg}}$, with the NLL criterion before we finally transition to the standard NLL, while blocking gradients with respect to the mean and $W$. This **prevents unstable NLL gradients** in the early training phase from excessively distorting the $W$. As shown in the ablation study in Section 4.2 (Figure 3(a)), removing $L_{\mathrm{stopNLL}}$ consistently worsens SHD, which confirms that the stop-gradient NLL is a key component that substantially contributes to the training dynamics and the performance.

---

> ### Author Response · Authors · 2025-11-21
>
> ## **Response to Q8:**
>
> Our contribution is not purely algorithmic, and the method we propose does not merely resolve an already-known issue. **We identify a new problem** that arises in DAG learning under HNM and **propose a tailored optimization strategy** to address our problem.
>
> In Section 1 (Line 67–91) of the revised version, **we have further clarified our contribution**.
>
>
> ### **(1) Convergence guarantee and stationary point.**
>
> In **Appendix C.1**, by applying the general convergence theorem of Wu (1983) [5] in a manner similar to Yin et al. (2024) [4], we show that the joint parameter sequence $(\Theta_t, W_t)$ **converges to the set of stationary points**. This convergence analysis begins with Assumption 1 of Appendix C.1, “**smoothness of scheduling coefficient**”, which is analogous to the “stationary trajectory / locally linear tracking” type of conditions obtained in homotopy-SGD [2] and in homotopy approaches for bivariate linear DAG [3].
>
> For nonlinear neural network-based multivariate HNM DAG learning, directly transplanting **strong global optimality guarantees** or “avoidance of all bad local minima” from the existing homotopy theory literature would require **very strong additional assumptions**, and to the best of our knowledge this remains an **open problem** even in related areas. The convergence guarantees in that literature are mostly formalized for **linear model and nonconvex objective function**. Under HNM, nonlinearity is inherent in the condition, so we do not claim such global optimality guarantees. Instead, **we focus on (i) formalizing and analyzing the early-stage failure mode that appears under HNM, and (ii) showing that the optimization strategy that mitigates this issue converges stably to a stationary point under reasonable assumptions.**
>
> In Section 3.1 (Line 310) of the revised version, we have added an explicit statement in the main paper that Appendix C.1 contains the convergence analysis.
>
>
> ### **(2) Complexity analysis.**
> Since **our method** uses the same optimization framework and network architecture as **NOTEARS-MLP** [7], its time complexity scales in the same way with respect to the number of samples and variables ($\boldsymbol{O(d^{3})}$ with respect to the number of variables $d$). Ours only adds pointwise loss terms as a surrogate loss and a scalar schedule $\lambda_{\mathrm{reg}}(t)$, which do not increase the time complexity. In contrast, **ICDH** [4] introduces an additional factor $S$ due to alternating optimization, leading to a time complexity of $\boldsymbol{O(S d^{3})}$.
>
> In Appendix I.8 and Table 12 of the revised version, we have added an analysis of computational complexity and runtime. We report wall-clock training time (in seconds), the total number of optimization iterations (in units of $10^3$), and peak GPU memory usage (in MB) for each method.
>
> **Ours** incurs about **1.7 times longer** wall-clock time than ICDH due to the joint optimization of the mean and variance networks. However, when we used **the standard NLL** as-is, the training took **3.7 times longer** than ICDH. This indicates that computing gradients for both the mean and variance simultaneously and updating parameters accordingly is heavy and leads to slow convergence. By introducing **the surrogate loss that can alleviate this**, we were able to save a substantial amount of time.
> **Ours** uses **4.7 times fewer iterations than ICDH** (ICDH requires more iterations due to the factor $S$), and the number of iterations increases only slightly compared to NOTEARS-MLP. We interpret this as a small increase in iterations caused by the process in the early training phase where the reconstruction loss is made to dominate the updates up to $t^*$; this is precisely why our optimization strategy yields performance improvements in DAG learning.
>
>
> - **Computational cost on synthetic ER2 graph (20 nodes)**
> | Method       | Time (s) | Iterations (×10³) | Memory (MB) |
> |-------------|---------:|------------------:|-----------:|
> | NOTEARS-MLP |   19.67  |            25.07  |     540.93 |
> | ICDH        |  389.85  |           131.88  |     542.87 |
> | NLL         | 1463.85  |           163.07  |     550.05 |
> | Ours        |  651.47  |            28.03  |     549.11 |

---

> ### Author Response · Authors · 2025-11-21
>
> ## **Response to Q9:**
>
> ### **(1) On the experimental setup and fairness of the comparison.**
> In all experiments, **we fixed the hyperparameters of our method across all datasets**, rather than tuning them per dataset. For all competing methods, **we used the official code released by the authors together with the default hyperparameters** specified in their papers. In addition, we extended the range of the number of variables and graph densities to be much broader than in prior work, and for each configuration we report the mean and standard deviation over 10 different random seeds. Therefore, the comparison in Figure 2 does not reflect a setting where only our method is favorably tuned; rather, each method is evaluated under its recommended settings and under the same experimental conditions.
>
> ### **(2) Why NOTEARS-MLP outperforms ICDH/HOST on structural metrics.**
>
> As explained in Section 4.1 and Appendix I.2, under the HNM setting, we interpret that **ICDH** [4] and **HOST** [8] tend to overestimate the variance in order to compensate for mean prediction errors, which in turn leads to **many spurious edges**. And it leads to higher SHD and FDR and a lower F1-score.
>
> By contrast, **NOTEARS-MLP** does not model the variance at all, and therefore does not suffer from such variance-related failures. As a result, it produces **fewer false positive edges**, and as the number of variables increases, it tends to achieve better SHD (lower) and F1-score (higher) than ICDH and HOST. However, NOTEARS-MLP has the limitation that, in terms of data reconstruction, it cannot properly capture heteroscedastic noise. In other words, rather than being due to randomness or implementation issues, these results are consistent with our analysis in Section 4.1 and Appendix I.2 of how directly optimizing input-dependent variance under HNM affects DAG learning.
>
>
> In Appendix I.5 (Table 5) of the revised version, **we added results on a larger graph ($d = 1000$), reporting TPR, FDR, and the total number of predicted edges** to support this analysis. **For a ground-truth DAG with 2,000 edges**, ICDH predicts **6,404** edges, achieving the highest **TPR (0.64)** but also a very large **SHD (5,158.3)**. This behavior is fully consistent with our explanation above.
>
>
> - **Comparison of SHD, F1-score, FDR, TPR, and the number of edges (1,000 nodes)**
> | Methods      | SHD (↓)            | F1-score (↑)       | FDR (↓)            | TPR (↑)           | # of edges      |
> |-------------|--------------------|--------------------|--------------------|-------------------|-----------------|
> | NOTEARS-MLP | 1592.3 ± 219.8     | 0.18 ± 0.127       | 0.89 ± 0.085       | 0.43 ± 0.142      | 2150 ± 237.61   |
> | ICDH        | 5158.3 ± 598.1     | 0.09 ± 0.066       | 0.95 ± 0.033       | **0.64 ± 0.128**  | 6404 ± 672.31   |
> | Ours        | **1130.5 ± 153.5** | **0.54 ± 0.093**   | **0.42 ± 0.087**   | 0.51 ± 0.109      | 1552 ± 75.66    |
>
> ## References for responses
>
> [1] Skafte, Nicki, Martin Jørgensen, and Søren Hauberg. "Reliable training and estimation of variance networks." Advances in Neural Information Processing Systems 32 (2019).
>
> [2] Gargiani, Matilde, et al. "Convergence analysis of homotopy-sgd for non-convex optimization." arXiv preprint arXiv:2011.10298 (2020).
>
> [3] Deng, Chang, et al. "Global optimality in bivariate gradient-based DAG learning." Advances in Neural Information Processing Systems 36 (2023): 17929-17968.
>
> [4] Yin, Naiyu, et al. "Effective causal discovery under identifiable heteroscedastic noise model." Proceedings of the AAAI Conference on Artificial Intelligence. Vol. 38. No. 15. 2024.
>
> [5] Wu, C.F. Jeff. “On the convergence properties of the EM algorithm.” The Annals of Statistics 11.1 (1983): 95–103.
>
> [6] Hazan, Elad, Kfir Yehuda Levy, and Shai Shalev-Shwartz. "On graduated optimization for stochastic non-convex problems." International conference on machine learning. PMLR, 2016.
>
> [7] Zheng, Xun, et al. "Learning sparse nonparametric dags." International conference on artificial intelligence and statistics. Pmlr, 2020.
>
> [8] Duong, Bao, and Thin Nguyen. "Heteroscedastic Causal Structure Learning." ECAI. 2023.

---

### Author Response · Authors · 2025-11-25
**The revised paper has been uploaded**

Dear Reviewers,

We are again sincerely grateful for your detailed questions and criticisms, which helped us significantly improve the quality of our paper. In response to the reviewers’ comments, we have carefully revised the paper and have uploaded it to OpenReview. All major additions and revisions in the manuscript are highlighted in blue.

A brief summary of the changes is as follows; upon your suggestion, we have:
- Clarified our contributions more explicitly,
- Explained the role of Lemma 3.1 and explicitly stated the assumptions and scope required for Claim 1,
- Refined several ambiguous or potentially confusing explanations,
- Performed additional experiments on (1) computational cost on a synthetic ER2 graph with 20 nodes; (2)
comparison with recent gradient-based DAG learning methods; (3) comparison with variance–estimation–based methods,
- Employed additional evaluation metrics to aid interpretation of the experimental results: (1) SHD, F1-score, FDR, TPR, and the number of predicted edges for the synthetic graphs with 1,000 nodes; (2) FDR and TPR for the real-world datasets.

These revisions mainly strengthen and clarify the exposition; there are no substantive changes to the core method, experimental results, or conclusions. We believe that this revision substantially improved the quality of the paper and addressed all of your concerns. If any concerns remain, please let us know. We would be more than happy to discuss them further.

---

### Author Response · Authors · 2025-12-03
**A Recap of Our Contributions and How We Addressed the Reviewers’ Concerns**

We sincerely thank the reviewers for their careful reading and constructive feedback, which have significantly improved the clarity and strength of the paper. We provide a recap of our contributions and summarizes how we addressed all concerns raised by the reviewers.

## Our Contributions
We formalize a failure mode that arises in continuous DAG learning under heteroscedastic noise models (HNMs), which has not been examined in prior work, and propose an optimization strategy tailored to this setting.
1. By analyzing the gradients of the reconstruction loss and the acyclicity constraint, we formalize a failure mode in which, in the early training phase of DAG learning, the structure updates are dominated by the acyclicity constraint.
2. To address this failure mode, we introduce an optimization strategy that transitions from a surrogate loss to the standard NLL while progressively enforcing the acyclicity constraint.
3. We show that, in a finite sample setting, our optimization strategy can attain lower risk than directly optimizing the HNM objective (Eq. 3), and we extend homotopy-based convergence conditions, previously established for the linear case, to the nonlinear HNM setting.

## Summary of Rebuttals
We address the main concerns raised by the reviewers as follows.
### (1) On the contributions and heuristic nature of the method
- Our novelty and contribution lie in formally characterizing a failure mode in DAG learning under HNM where the acyclicity constraint drives structure updates and designing an optimization strategy that directly targets this failure mode.
- The schedule function is not an arbitrary choice but is derived within a family of schedules that satisfy relaxed conditions [1] for nonlinear networks (e.g., sufficiently large initial weights and slow, gradual decrease). The schedule is derived so as to maintain the optimization path toward a stationary point, and we empirically validate this via optimization trajectory analysis.
- The homotopy-based method [2] focuses on simple bivariate linear DAGs, whereas we extend this idea to a nonlinear multivariate HNM regime. Here, global optimality is no longer guaranteed, but we provide sample-dependent conditions under which our method improves DAG recovery and support them empirically.
- Whereas the homotopy-based method [2] schedules the weights of loss terms, we design a strategy that avoids early failure mode in HNM and transitions from easier objectives to the final NLL. We decouple NLL into mean and variance terms and jointly optimize them to restore the reconstruction signal, then align only the variance parameters to the NLL criterion, and smoothly converge to the final objective. All surrogate components are designed to ultimately reach the final NLL, and ablations show their importance.
- We have added a detailed comparison with ICDH [3] in Appendix B.3.
### (2) On the strong assumptions and justification
- We have clearly positioned our theory not as claiming global convergence for general nonlinear neural networks, but as providing an interpretation of the proposed failure mode and the working mechanism of the homotopy-based strategy.
- We explicitly state that we follow the relaxed conditions for nonlinear functions provided by homotopy optimization theory [1,2]. We then confirm through empirical analyses that these conditions tend to hold in practice and lead to stable optimization trajectories.
- We clarify that, instead of strong assumptions such as a global PL-type condition, we adopt relaxed assumptions such as local PL-type conditions that are commonly used in analyses of nonlinear networks [1].
- We have explicitly defined all constants in the proof of Claim 1.
### (3) On additional experiments and schedule hyperparameter tuning
- We have added evaluation results for the recent gradient-based DAG learning methods and complexity analysis.
- We use fixed schedule hyperparameters across all experiments. Our method addresses the failure mode and achieves competitive performance without dataset-specific tuning, which suggests low sensitivity to these choices. The schedule parameters were chosen within reasonable ranges that satisfy relaxed conditions [1] to mitigate the failure mode.

We believe that the revised version of the paper has been substantially strengthened by the reviewers’ feedback and provides a meaningful contribution to the area of DAG learning under HNM. We hope that the Area Chair will find that the revisions adequately address all the reviewers’ concerns and support a positive decision on the paper.

[1] Gargiani, Matilde, et al. "Convergence analysis of homotopy-sgd for non-convex optimization." arXiv preprint arXiv:2011.10298 (2020).

[2] Deng, Chang, et al. "Global optimality in bivariate gradient-based DAG learning." in NeurIPS 36 (2023): 17929-17968.

[3] Yin, Naiyu, et al. "Effective causal discovery under identifiable heteroscedastic noise model." in AAAI 2024.

---

### Meta-Review · Area_Chair_HvUH · 2026-01-06

**Summary:**

This paper proposes a graduated optimization strategy for learning Directed Acyclic Graphs under a Heteroscedastic Noise Model.

**Reviewer Concerns:**

Although the paper has some merits, such as clearly identifying a specific optimization challenge and providing comprehensive empirical evaluations, the issues raised by the reviews are critical. For instance, the overstated novelty and inaccurate statements regarding prior work (Reviewer 9YYo), the lack of theoretical guarantees and convergence analysis for the proposed heuristic scheduling mechanism (Reviewer fhFt), and the somewhat limited contribution (Reviewer 1251).

**Reviewer Scores:**

9YYo would remain the score as it raises concerns about novel theoretical distinctions, which are unlikely to be resolved through discussion alone.

tzGf and 1251 would remain the score as they found the paper is well-written and clearly identified a challenge.

fhFt would remain the score as 4 as there is still lack of detailed theoretical analysis.

---

### Decision · Program_Chairs · 2026-01-26

Reject